# Theory-Driven Label-Specific Representation for Incomplete Multi-View Multi-Label Learning

**Quanjiang Li**[1†]
liquanjiang@nudt.edu.cn

**Tianxiang Xu**[2†]
xtx09@hotmail.com

**Tingjin Luo**[1*]
tingjinluo@hotmail.com

**Yan Zhong**[2]
zhongyan@stu.pku.edu.cn

**Yang Li**[1]
liyang_albert@foxmail.com

**Yiyun Zhou**[3]
yiyunzhou@zju.edu.cn

**Chenping Hou**[1]
hcpnudt@hotmail.com

[1]National University of Defense Technology,  [2]Peking University,  [3]Zhejiang University

## Abstract

Multi-view multi-label learning typically suffers from dual data incompleteness due to limitations in feature storage and annotation costs. The interplay of heterogeneous features, numerous labels, and missing information significantly degrades model performance. To tackle the complex yet highly practical challenges, we propose a Theory-Driven Label-Specific Representation (TDLSR) framework. Through constructing the view-specific sample topology and prototype association graph, we develop the proximity-aware imputation mechanism, while deriving class representatives that capture the label correlation semantics. To obtain semantically distinct view representations, we introduce principles of information shift, interaction and orthogonality, which promotes the disentanglement of representation information, and mitigates message distortion and redundancy. Besides, label-semantic-guided feature learning is employed to identify the discriminative shared and specific representations and refine the label preference across views. Moreover, we theoretically investigate the characteristics of representation learning and the generalization performance. Finally, extensive experiments on public datasets and real-world applications validate the effectiveness of TDLSR.

## 1 Introduction

The popularity of multi-view learning stems from its ability to provide comprehensive representations of samples [29, 9]. Integrating multi-source information effectively reveals latent semantic associations in multimodal data, thereby enriching the feature space and improving model accuracy and generalization [10]. With the advancement of information technology, single-label proves inadequate in meeting the object labeling and recognition demands [3]. In practice, objects frequently fall under multiple categories simultaneously, with a document in text classification [40] being assigned labels like topic, intended audience, sentiment and so on. Benefiting from the holistic characterization offered by multi-view approaches [41] and the capacity of multi-label methods [31] to capture all sample attributes, diverse views and labels are emerging as the dominant form of training data. Consequently, multi-view multi-label classification (MvMLC) has attracted significant research attention, as it delivers a refined understanding of the complexity and diversity inherent in real-world situations.

---

[†]Equal contribution
[*]Corresponding Author

39th Conference on Neural Information Processing Systems (NeurIPS 2025).

Existing MvMLC methods are capable of implementing both multivariate feature fusion and multi-objective joint discrimination. For example, LVSL [43] performed view-specific label learning and leveraged low-rank label structures to enhance performance. E$F^2$FS [12] proposed an embedded feature selection model that integrated feature aggregation and enhancement. However, these ideal methods assume the presence of both features and labels, whereas limitations in feature collection techniques and annotation complexity result in the unavailability of partial views and tags [37], motivating researchers to explore the incomplete multi-view multi-label classification (iMvMLC) approach. Early efforts like iMVWL [33] employed a joint learning strategy to refine the shared subspace and enhance the robustness of the weak label classifier. NAIM3L [21] combined matrix factorization with global high-rank and local low-rank constraints to obtain a shared label space. Given the ability to capture complex semantic information, deep learning based methods have demonstrated promising performance. The pioneering work DICNet [25] introduced an incomplete instance-level contrastive learning method for improved consensus representation, while LMVCAT [26] designed two transformer-based modules for cross-view feature aggregation and category awareness.

Despite the emergence of various effective iMvMLC methods, they remain deficient in feature information reconstruction, semantic distinctness of extracted representations and the coupling between label correlation semantics with feature connotation. (i) The absence of views severely undermines both the capture of cross-view dependencies and the stability of downstream modules. Approaches [4] that only mask missing samples tend to circumvent the challenge, rather than engaging in robust information recovery. AIMNet [23] attempted to perform missing imputation via cross-view global attention computation. However, the integration of a global weighting scheme may further exacerbate reconstruction noise since not all samples are strongly correlated. (ii) The essence of multi-view learning lies in extracting the complementarity and consistency across diverse views [36]. Additionally, increased attention should be devoted to feature separation in multi-label scenarios [42], as distinct labels exhibit varying sensitivities towards particular features. DIMC [38], TSIEN [34] and SIP [27] primarily focused on shared subspaces while overlooking the unique characteristics of individual views. MTD [24] attempted to obtain shared and specific representations through geometric distance constraints. However, this linear interaction pattern struggled to reflect the intricate inter-view relationships. (iii) Label relevance is fundamental to multi-label learning and distinguishes it from multi-class problem [44], which makes multiple independent binary classifications insufficient as done in some methods [24]. Besides, instead of being considered in isolation, mutually dependent label semantics should interact seamlessly with feature information to facilitate both the perception of label-specific features and the selection of the preferred labels corresponding to those features.

To tackle these issues, we propose a Theory-Driven Label-Specific Representation framework named TDLSR. The motivation behind TDLSR is to minimize view reconstruction errors, improve the semantic discriminability of extracted representations and strengthen the interaction between label-relevant semantics and feature information. We first construct view-specific instance relation graphs using the attention mechanism integrated with neighborhood-aware selection. Without any discrepancy arising from network parameter updates, the reconstruction risk is minimized by propagating highly relevant sample information across views. By introducing the principles of information shift, interaction and orthogonality, we develop a mutual information optimization model that aligns the core constituents of representations with their respective views, facilitates interaction among shared representations, suppresses cross-talk between private components and enforces orthogonality of shared and specific information from the same view. Label correlation information is transmitted via graph-induced relational network modeling, which results in the emergence of interdependent category representatives. Through discrete engagements between each class prototype and semantically distinct features, the most sensitive feature ensemble for each label across both shared and specific feature pools can be identified. We further establish the generalization error bound via error decomposition, showing that feature disentanglement maximizes the mutual information between representations and objects and reduces generalization error. The main contributions of our work are summarized as follows:

- We propose a general multi-view representation extraction model inspired by information theory. This model guarantees unbiased representations through the constraint of information shift, separates shared and specific semantic via the regulation of information interaction, and eliminates representation redundancy by strengthening information orthogonality.

- TDLSR enhances the propagation of feature dependencies and label correlation semantics by constructing relational graphs. Besides, it introduces label-specific shared and private feature

learning for the first time. Theoretically, we prove the discriminability and effectiveness of feature extraction and derive the generalization error bound.

- Extensive experimental results across diverse public available datasets, along with applications on real-world NBA data, validate the effectiveness and robustness of our method.

## 2   Method

In this section, we present the following critical components of our TDLSR as shown in Fig. 1: proximity-aware graph attention recovery mechanism, universal view extraction framework under mutual information constraints and multi-label semantic and label-specific representation learning.

### 2.1   Problem definition

We define $\{\boldsymbol{X}^{(v)}\}_{v=1}^{V}$ as original multi-view setting, where $\boldsymbol{X}^{(v)} = \{\boldsymbol{x}_i^{(v)}\}_{i=1}^{N} \in \mathbb{R}^{N \times d_v}$ represents $d_v$ dimensional feature matrix of the $v$-th view. The label matrix $\boldsymbol{Y} \in \{0,1\}^{N \times C}$ corresponds to $C$ categories, with $\boldsymbol{Y}_{i,j} = 1$ if sample $i$ is tagged as class $j$. Besides, we let $\boldsymbol{W} \in \{0,1\}^{N \times V}$ and $\boldsymbol{G} \in \{0,1\}^{N \times C}$ denote the missing indicator for views and labels, respectively. Specifically, $\boldsymbol{W}_{i,j} = 1$ if the $j$-th view of the $i$-th sample is available, otherwise $\boldsymbol{W}_{i,j} = 0$. Similarly, $\boldsymbol{G}_{i,j} = 1$ or $0$ reflects the certainty of the corresponding label. Our goal is to train an end-to-end neural network capable of performing classification inference on incomplete multi-view weak multi-label data.

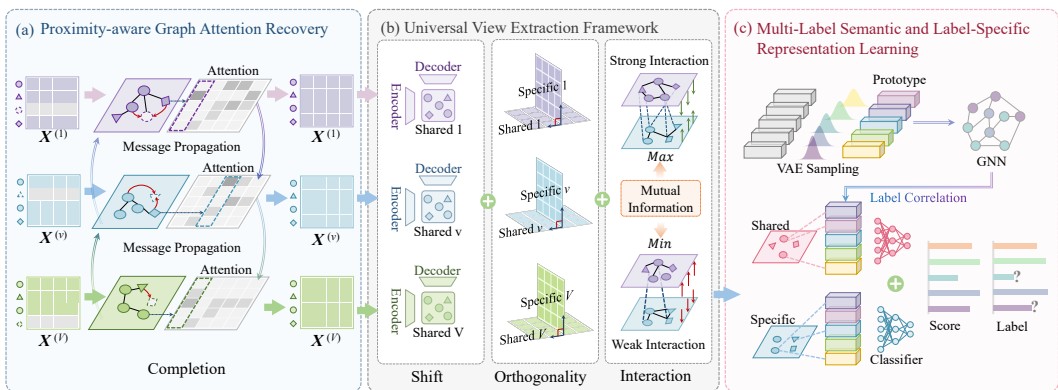

Figure 1: The main framework of our proposed TDLSR. Different shapes signify different samples.

### 2.2   Proximity-aware Graph Attention Recovery Mechanism

The lack of diverse views limits the capacity of deep neural models to extract high-level representations [20]. Consequently, data augmentation through reconstruction techniques is essential for improving model accuracy and stability. To mitigate information deficiency arising from incompleteness, we propose an attention-based relational graph construction strategy to propagate similarity signals across samples for missing imputation. For any view $v$, the attention score for an instance pair is computed by $\boldsymbol{B}_{i,j}^{(v)} = e^{h(x_i^{(v)})h(x_j^{(v)T})/\tau}$, where $\tau$ is the temperature parameter and $h(\cdot)$ denotes the normalization function. Each row of the matrix $\boldsymbol{B}^{(v)}$ quantifies the degree of similarity between the corresponding sample with all other samples [23], which facilitates the identification of the $k$-nearest neighbors for each instance. Therefore, we construct the view-specific graph $\hat{\boldsymbol{S}}^{(v)} \in \mathbb{R}^{N \times N}$ through attention-induced proximity awareness, where $\hat{\boldsymbol{S}}_{i,j}^{(v)} = 1$ means $\boldsymbol{W}_{i,v}\boldsymbol{W}_{j,v} = 1$ and $\boldsymbol{B}_{i,j}^{(v)}$ is one of the top-$k$ largest elements in the $i$-th row, i.e., $\boldsymbol{x}_j^{(v)}$ is the neighbor of $\boldsymbol{x}_i^{(v)}$. Considering similarity relations between instances in existing views are applicable to the missing views, the transferred graph for finding the available instances related to the missing ones can be obtained:

$$\boldsymbol{K}^{(v)} = \sum_{k=1, k \neq v}^{V} \hat{\boldsymbol{S}}^{(k)} \operatorname{diag}\left(\boldsymbol{W}_{:,v}\right), \tag{1}$$

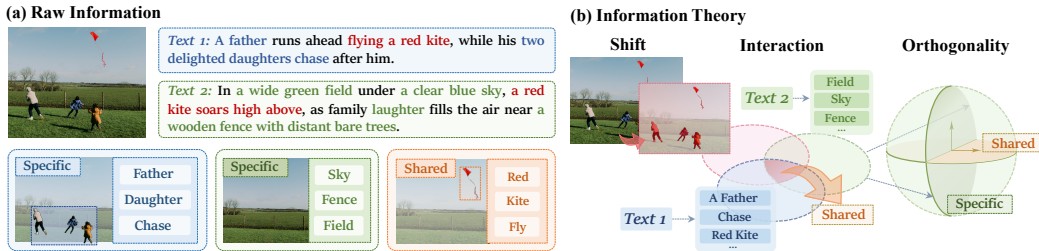

Figure 2: Depiction of information shift, interaction, and orthogonality.

where $\mathrm{diag}(\cdot)$ creates a diagonal matrix and $\boldsymbol{W}_{:,v}$ is the $v$-th column. Besides, the contribution of available samples to reconstruct missing ones is governed by the maximal computable correlation from alternative views:

$$\bar{\boldsymbol{B}}_{i,j} = \max\left(\boldsymbol{B}_{i,j}^{(1)}\hat{\boldsymbol{S}}_{i,j}^{(1)}, \boldsymbol{B}_{i,j}^{(2)}\hat{\boldsymbol{S}}_{i,j}^{(2)}, \cdots, \boldsymbol{B}_{i,j}^{(V)}\hat{\boldsymbol{S}}_{i,j}^{(V)}\right), \tag{2}$$

where $\bar{\boldsymbol{B}}_{i,j}$ encapsulates the influence of the $j$-th sample in the recovery process of the $i$-th sample. By treating $\boldsymbol{K}$ as the adjacency matrix and $\bar{\boldsymbol{B}}$ as the edge weight, we can get the reconstructed data after message propagation:

$$\hat{\boldsymbol{x}}_i^{(v)} = \frac{\sum_{\boldsymbol{K}_{i,j}^{(v)} \geq 1} \boldsymbol{K}_{i,j}^{(v)} \bar{\boldsymbol{B}}_{i,j} \boldsymbol{x}_j^{(v)}}{\sum_{\boldsymbol{K}_{i,j}^{(v)} \geq 1} \bar{\boldsymbol{B}}_{i,j}}. \tag{3}$$

Since $\hat{\boldsymbol{X}}^{(v)}$ serves as an approximate proxy of missing instances, we combine it with the original view to generate the final recovery matrix for downstream tasks:

$$\boldsymbol{Z}_{i,:}^{(v)} = \hat{\boldsymbol{X}}_{i,:}^{(v)}\left(1 - \boldsymbol{W}_{i,v}\right) + \boldsymbol{X}_{i,:}^{(v)}\boldsymbol{W}_{i,v}. \tag{4}$$

### 2.3 Universal View Extraction Framework under Mutual Information Constraints

Previous studies [27] have primarily concentrated on enabling networks to extract shared information, while overlooking the contributions of each unique view. Moreover, the reliance on linear geometric constraints in early view-specific representation learning [24] falls short of capturing complex feature interactions. To effectively assess both the consistency and the complementarity of distinct features, we propose a general model that directly constrains the mutual information between representations for precisely measuring their interaction degree. Two groups of multi-layer perceptrons $\{M_v^S\}_{v=1}^V$ and $\{M_v^O\}_{v=1}^V$ are employed as shared and view-private channel to extract the corresponding representations, i.e., $\{M_v^S : \boldsymbol{Z}^{(v)} \rightarrow \boldsymbol{S}^{(v)}\}_{v=1}^V$ and $\{M_v^O : \boldsymbol{Z}^{(v)} \rightarrow \boldsymbol{O}^{(v)}\}_{v=1}^V$. Taking the image-text retrieval task [32] for explanation, as illustrated in the Fig. 2, an image is typically paired with multiple textual descriptions that emphasize on individual relationships and environmental context, respectively. It requires the model to simultaneously extract the consistent shared representations (central actions) and unique modality-specific elements (e.g., distinct individual identities and background details) for holistic multimodal perception. To achieve this, maximizing $I(\boldsymbol{s}^{(v)}; \boldsymbol{z}^{(v)})$ is critical for guaranteeing that the representations stay consistent with the core semantics of raw modalities, such as people, kit, background, etc. Besides, comprehensive feature disentanglement is facilitated by maximizing the mutual information $I(\boldsymbol{s}^{(v^*)}|\boldsymbol{z}^{(v^*)}; \boldsymbol{s}^{(v)}|\boldsymbol{z}^{(v)})$ and minimizing $I(\boldsymbol{o}^{(v^*)}|\boldsymbol{z}^{(v^*)}; \boldsymbol{o}^{(v)}|\boldsymbol{z}^{(v)})$ between representations from any pair of views $v$ and $v^*$ $(1 \leq v \neq v^* \leq V)$, which focuses on capturing the central characteristics of the actions and preserving unique information embedded in each modality, respectively. Furthermore, establishing complete representation orthogonality from the same modality, as indicated by $I(\boldsymbol{s}^{(v)}|\boldsymbol{z}^{(v)}; \boldsymbol{o}^{(v)}|\boldsymbol{z}^{(v)}) = 0$, effectively minimizes semantic redundancy, such as the identification of the individual and their actions.

As mentioned above, feature separation is driven by the following three criteria: (i) prevent information shift, (ii) optimize information interaction and (iii) promote information orthogonality. Under

the condition of mutual information constraints, the universal model can be expressed as

$$
\begin{cases}
\max \underbrace{\sum_{v=1}^{V} I(\boldsymbol{s}^{(v)}; \boldsymbol{z}^{(v)})}_{\text{information shift}} + \underbrace{\sum_{v=1}^{V} \sum_{v^* \neq v}^{V} (I(\boldsymbol{s}^{(v^*)}|\boldsymbol{z}^{(v^*)}; \boldsymbol{s}^{(v)}|\boldsymbol{z}^{(v)}) - I(\boldsymbol{o}^{(v^*)}|\boldsymbol{z}^{(v^*)}; \boldsymbol{o}^{(v)}|\boldsymbol{z}^{(v)}))}_{\text{information interaction}} \\
s.t. \quad \underbrace{\sum_{v=1}^{V} I(\boldsymbol{s}^{(v)}|\boldsymbol{z}^{(v)}; \boldsymbol{o}^{(v)}|\boldsymbol{z}^{(v)}) = 0}_{\text{information orthogonality}}.
\end{cases} \tag{5}
$$

By employing the Lagrange multiplier method, the equality constraint can be appropriately scaled. Direct computation of mutual information is impractical due to the challenges of distribution inference and the complexity of high-dimensional integrals. Thus, stability and feasibility are typically ensured by refining estimable mutual information bounds [35]. Regarding the information shift term, its lower bound is commonly represented through a reconstruction loss[15], where the representation $\boldsymbol{s}^{(v)}$ is decoded via $q(\boldsymbol{z}^{(v)}|\boldsymbol{s}^{(v)})$ to faithfully preserve the original view:

$$
I(\boldsymbol{s}^{(v)}; \boldsymbol{z}^{(v)}) \geq \mathbb{E}_{p(\boldsymbol{z}^{(v)}, \boldsymbol{s}^{(v)})} \left[ \log q \left( \boldsymbol{z}^{(v)}|\boldsymbol{s}^{(v)} \right) \right]. \tag{6}
$$

For the second term, based on the definition expansion and the non-negativity of entropy, we can derive the following lower bound by introducing another variational distribution $q(\boldsymbol{s}^{(v)}|\boldsymbol{s}^{v^*})$:

$$
I(\boldsymbol{s}^{(v^*)}|\boldsymbol{z}^{(v^*)}; \boldsymbol{s}^{(v)}|\boldsymbol{z}^{(v)}) \geq \int \int p(\boldsymbol{s}^{(v)}|\boldsymbol{z}^{(v)}/\boldsymbol{s}^{(v^*)}|\boldsymbol{z}^{(v^*)}) p(\boldsymbol{s}^{(v^*)}|\boldsymbol{z}^{(v^*)}) \log q(\boldsymbol{s}^{(v)}|\boldsymbol{s}^{v^*}) d\boldsymbol{s}^{(v^*)} d\boldsymbol{s}^{(v)}
$$
$$
+ \int \int p(\boldsymbol{s}^{(v^*)}|\boldsymbol{z}^{(v^*)}) \underbrace{p(\boldsymbol{s}^{(v)}|\boldsymbol{z}^{(v)}/\boldsymbol{s}^{(v^*)}|\boldsymbol{z}^{(v^*)}) \log \frac{p(\boldsymbol{s}^{(v)}|\boldsymbol{z}^{(v)}/\boldsymbol{s}^{(v^*)}|\boldsymbol{z}^{(v^*)})}{q(\boldsymbol{s}^{(v)}|\boldsymbol{s}^{v^*})} d\boldsymbol{s}^{(v)}}_{\text{Kullback-Leibler divergence}} d\boldsymbol{s}^{(v^*)},
$$

$$ \tag{7} $$

where $p(\boldsymbol{s}^{(v)}|\boldsymbol{z}^{(v)}/\boldsymbol{s}^{(v^*)}|\boldsymbol{z}^{(v^*)})$ refers to the conditional distribution between the shared representations from $\boldsymbol{z}^{(v)}$ and $\boldsymbol{z}^{(v^*)}$. Leveraging the non-negativity of Kullback–Leibler divergence between the distributions $p(\boldsymbol{s}^{(v)}|\boldsymbol{z}^{(v)}/\boldsymbol{s}^{(v^*)}|\boldsymbol{z}^{(v^*)})$ and $q(\boldsymbol{s}^{(v)}|\boldsymbol{s}^{(v^*)})$, the tighter lower bound is further obtained. Similarly, upper bounds for the remaining two negative terms can be derived following the same rationale. Therefore, the problem (16) is transformed into optimizing the following objective:

$$
\mathcal{L}_{IB} = - \underbrace{\mathbb{E}_{p(\boldsymbol{z}^{(v)}, \boldsymbol{s}^{(v)})} \left[ \log q \left( \boldsymbol{z}^{(u)}|\boldsymbol{s}^{(v)} \right) \right]}_{\text{lower bound for information shift}} - \underbrace{\int \int p(\boldsymbol{s}^{(v)}|\boldsymbol{z}^{(v)}, \boldsymbol{s}^{(v^*)}|\boldsymbol{z}^{(v^*)}) \log q(\boldsymbol{s}^{(v)}|\boldsymbol{s}^{v^*}) d\boldsymbol{s}^{(v^*)} d\boldsymbol{s}^{(v)}}_{\text{lower bound for shard information interaction}}
$$
$$
+ \underbrace{D_{KL}(p(\boldsymbol{o}^{(v^*)}|\boldsymbol{z}^{(v^*)}; \boldsymbol{o}^{(v)}|\boldsymbol{z}^{(v)})\|p(\boldsymbol{o}^{(v^*)}|\boldsymbol{z}^{(v^*)})q(\boldsymbol{o}^{(v)}|\boldsymbol{o}^{(v^*)}))}_{\text{upper bound for specific information interaction}}
$$
$$
+ \beta \underbrace{D_{KL}(p(\boldsymbol{s}^{(v)}|\boldsymbol{z}^{(v)}; \boldsymbol{o}^{(v)}|\boldsymbol{z}^{(v)})\|p(\boldsymbol{o}^{(v)}|\boldsymbol{z}^{(v)})q(\boldsymbol{s}^{(v)}|\boldsymbol{o}^{(v)}))}_{\text{upper bound for information orthogonality}},
$$

$$ \tag{8} $$

where $\beta$ is the Lagrange multiplier. Minimizing $\mathcal{L}_{IB}$ promotes effective distinction between shared and specific feature semantic, with implementation details provided in the Appendix.

## 2.4 Multi-Label Semantic and Label-Specific Representation Learning

In multi-label classification tasks, binary encoding struggles to capture the underlying label semantics [19]. To overcome this limitation, we employ a data-driven approach to learn label prototypes [27], which provides a clear insight into label structures and enhances effective semantic associations with features. After initializing the one-hot vector $\boldsymbol{b}_i \in \mathbb{R}^C$ for each category, we utilize two stochastic encoders to model the prototype distribution i.e., $\boldsymbol{h}_i \sim \mathcal{N}\left(\mu_i, \sigma_i^2 \boldsymbol{I}\right)$, where the mean $\mu_i$ and variance $\sigma_i$ are determined by the encoders $g_\mu\left(\boldsymbol{b}_i\right)$ and $g_{\sigma^2}\left(\boldsymbol{b}_i\right)$. Then, we sample from the distribution with the reparameterization trick to obtain the prototype representation $\boldsymbol{h}_i^0 = \frac{1}{s} \sum_{d=1}^{s}\left(\mu_i + \sigma_i \odot \delta^d\right)$ [7], where $\delta^d$ means the $d$-th individual sampling and $\odot$ denotes the element-wise product.

The prototype representations are independently learned for each label, yet investigating intrinsic label correlation remains a core challenge in multi-label learning [3]. For bridging this gap, graph neural networks are leveraged to propagate prior correlation information and refine the label representations to encapsulate the inherent relationships of label semantics. Prototypes are regraded as a set of nodes positioned on the label relation graph, with edge weights reflecting the correlation between corresponding label pairs. Besides, the label correlation matrix $\boldsymbol{A}$ quantitatively characterized on the training data serves as the appropriate substitute for the adjacency matrix. Rather than utilizing co-occurrence frequency to evaluate correlation degree, we use the Jaccard distance [30] calculated over positive labels, as we are only concerned with the categories assigned to each instance. By computing the intersection and union of two classes regarding positive values, we can obtain

$$A_{i,j} = \frac{\langle \boldsymbol{Y}_{:,i} \cdot \boldsymbol{Y}_{:,j} \rangle}{\sum_{k=1}^{N} (\boldsymbol{Y}_{k,i} + \boldsymbol{Y}_{k,j}) - \langle \boldsymbol{Y}_{:,i} \cdot \boldsymbol{Y}_{:,j} \rangle}, \tag{9}$$

where $A_{ii}$ is set to 0 to eliminate self-dependency. Given an aggregated matrix $\boldsymbol{H} \in \mathbb{R}^{C \times d}$, the label embeddings corresponding to each row can be updated by passing through the GIN layer with propagated correlation information [11], i.e., $\boldsymbol{E} = f\left[(1 + \epsilon)\boldsymbol{H} + \boldsymbol{A}\boldsymbol{H}\right]$, where $f(\cdot)$ denotes a fully-connected layer followed by Batch Normalization and Leaky ReLU activation, and $\epsilon$ is a learnable scalar that controls the influence of node's own features. To reinforce cohesion between relevant prototype representations and distinguish unrelated ones, we employ the following objective that aligns representation similarity with label correlation:

$$\mathcal{L}_{le} = -\frac{1}{C^2} \sum_{i=1}^{C} \sum_{j=1}^{C} \hat{\boldsymbol{A}}_{ij} \log\left(\cos\left(\boldsymbol{E}_i, \boldsymbol{E}_j\right)\right) + (1 - \hat{\boldsymbol{A}}_{ij}) \log\left(1 - \cos\left(\boldsymbol{E}_i, \boldsymbol{E}_j\right)\right), \tag{10}$$

where $\cos\left(\boldsymbol{E}_i, \boldsymbol{E}_j\right)$ is the cosine similarity and $\hat{\boldsymbol{A}} = \boldsymbol{A} + \boldsymbol{I}$ with $\boldsymbol{I}$ denoting an identity matrix. Multi-label classification tasks often involve labels with varying sensitivities to different feature subsets [14]. Consequently, label-specific feature learning has become a widely adopted technique to select the most relevant features tailored to classifying each label. However, label-specific disentangled feature learning remains underexplored, despite its potential to boost model performance. For instance, in image recognition, shared information capture general visual cues, while private features highlight other distinctive traits tied to each label, such as breed for "dog" and texture for "cat". Thus, we treat activated label prototypes as feature importance scores and engage them with both shared and private representations to discern the label-specific view embeddings:

$$\hat{\boldsymbol{P}}_i^{(v)} = \left[\sigma_S\left(\boldsymbol{E}_1\right) \odot x_i^{(v)}; \sigma_S\left(\boldsymbol{E}_2\right) \odot x_i^{(v)}; \dots; \sigma_S\left(\boldsymbol{E}_C\right) \odot x_i^{(v)}\right], \tag{11}$$

where $\sigma_S$ is the Sigmoid function. According to the Eq. (11), we can obtain the label-specific shared and private features, i.e., $\{\boldsymbol{S}^{(v)} \to \hat{\boldsymbol{U}}^{(v)} \in \mathbb{R}^{N \times C \times d}\}_{v=1}^{V}$ and $\{\boldsymbol{O}^{(v)} \to \hat{\boldsymbol{V}}^{(v)} \in \mathbb{R}^{N \times C \times d}\}_{v=1}^{V}$. The interaction between label semantics and view representations supports a bidirectional selection mechanism, in which discriminative views are assigned to specific labels, while information-related label subsets are uncovered associated with distinct views. Processing through the linear classifiers, view-specific predictions $\boldsymbol{U}^{(v)} \in \mathbb{R}^{N \times C}$ and $\boldsymbol{V}^{(v)} \in \mathbb{R}^{N \times C}$ are generated. Given the variability in reconstruction quality, views with higher recovery accuracy should be emphasized in the fusion process. For this aspect, certain reconstructed views characterized by low attention scores in relation to their associated samples naturally exhibit reduced confidence [23]. Thus, we calculate the maximum original attention for each instance with respect to other instances as its confidence score:

$$\boldsymbol{Q}_{i,v} = \max\left(\left\{\tau \log \bar{\boldsymbol{B}}_{i,j} \boldsymbol{W}_{j,v}\right\}_{j=1}^{n}\right), \tag{12}$$

where $\bar{\boldsymbol{B}}$ is computed by Eq. (2) and $\boldsymbol{Q} \in \mathbb{R}^{N \times V}$ stores the confidence score for individual instances. Since $\boldsymbol{Q}$ is tailor-made for the missing samples, the confidence matrix is updated as $\boldsymbol{Q}' = (1 - \boldsymbol{W}) \odot \boldsymbol{Q} + \boldsymbol{W}$. During the late fusion, we combine the feature reconstruction efficiency to obtain the final prediction:

$$\overline{\boldsymbol{P}}_{i,:} = \sigma_S\left(\frac{\sum_{v=1}^{V} \boldsymbol{U}_{i,:}^{(v)} \boldsymbol{Q}_{i,v}'}{\sum_{v=1}^{V} \boldsymbol{Q}_{i,v}'} + \frac{\sum_{v=1}^{V} \boldsymbol{V}_{i,:}^{(v)} \boldsymbol{Q}_{i,v}'}{\sum_{v=1}^{V} \boldsymbol{Q}_{i,v}'}\right). \tag{13}$$

Then, we employ the weighted cross-entropy loss to mitigate the impact of missing labels:

$$\mathcal{L}_{bce} = -\frac{1}{\sum_{i,j} \boldsymbol{G}_{i,j}} \sum_{i=1}^{N} \sum_{j=1}^{C} \left(\boldsymbol{Y}_{i,j} \log\left(\overline{\boldsymbol{P}}_{i,j}\right) + (1 - \boldsymbol{Y}_{i,j}) \log\left(1 - \overline{\boldsymbol{P}}_{i,j}\right)\right) \boldsymbol{G}_{i,j}. \tag{14}$$

By incorporating $\lambda_1$ and $\lambda_2$ to balance the loss effects, our total training loss can be expressed as

$$\mathcal{L} = \mathcal{L}_{bce} + \lambda_1 \mathcal{L}_{IB} + \lambda_2 \mathcal{L}_{le}. \tag{15}$$

## 2.5 Theoretical Results

In this subsection, we aim to theoretically explore the fundamental mechanisms that contribute to the model performance and the generalization capability of TDLSR. Through rigorous derivations (proofs in the Appendix), we obtain the following theorems:

**Theorem 1.** (*Discriminability of Label-specific Representation.*) *For label prototypes $E_j$ and $E_k$ such that $k \neq j$ for all $k$, and view representations $X^{(v)}$ and $X^{(v^*)}$ such that $v^* \neq v$ for all $v^*$, the discriminability of $\hat{P}^{(v)}$ for class $j$ necessitates that either of the following conditions be satisfied:*

$$
\begin{cases}
\mathbb{E}\left[X^{(v)}\sigma_S(E_j)^\top\right] = \dfrac{1}{N}\sum_{i=1}^{N} x_i^{(v)}\sigma_S(E_j)^\top > \mathbb{E}\left[X^{(v)}\sigma_S(E_k)^\top\right] = \dfrac{1}{N}\sum_{i=1}^{N} x_i^{(v)}\sigma_S(E_k)^\top \\[4mm]
\mathbb{E}\left[X^{(v)}\sigma_S(E_j)^\top\right] = \dfrac{1}{N}\sum_{i=1}^{N} x_i^{(v)}\sigma_S(E_j)^\top > \mathbb{E}\left[X^{(v^*)}\sigma_S(E_j)^\top\right] = \dfrac{1}{N}\sum_{i=1}^{N} x_i^{(v^*)}\sigma_S(E_j)^\top.
\end{cases}
$$

**Theorem 2.** (*Effectiveness of Disentangled Representation.*) *Let the disentangled representation be denoted as $R = (S^{(1)}, \ldots, S^{(V)}, O^{(1)}, \ldots, O^{(V)})$, where the information entropy of each representation is assumed to be fixed, i.e., $H(S^{(v)}) = H(O^{(v)}) = H^0$ ($1 \leq v \leq V$). Then, in the case where each shared and specific representation is indispensable for prediction, $I(R; Y)$ will attain its maximum when $R = R_*$, with $R_*$ being the optimal solution of the problem (16).*

**Theorem 3.** (*Generalization Error Bound.*) *Our model is designed to learn a vector-valued function $f = (f_1, \ldots, f_C) : \mathcal{X} \mapsto \mathbb{R}^C$. The expected risk and empirical risk w.r.t. the training dataset $D$ are denoted as $R(f) = \mathbb{E}_{(X,Y) \sim \mathcal{X} \times \mathcal{Y}}[\ell(f^{av}(X, Q), Y)]$ and $\hat{R}_D(f) = \frac{1}{NC}\sum_{i=1}^{N}\sum_{c=1}^{C}\ell(\sum_{v=1}^{V}(Q_{i,v}f_c(x_i^{(v)})), Y_{i,c})$, where $f^{av}(\cdot)$ refers to the late fusion of multiple views. With probability at least $1 - \delta$, we have the following generalization error bound:*

$$
R(f) - \hat{R}_D(f) \leq \frac{\widetilde{\mathcal{K}}_1}{\sqrt{NV}} + \frac{\widetilde{\mathcal{K}}_2}{N^{3/4}V^{1/4}} + \overline{\text{gen}}_{rec}(Q, X, Y)
$$
$$
+ \widetilde{\mathcal{K}}_3 \sqrt{\frac{\sum_{c=1}^{C}\left(\sum_{v=1}^{V} I(X^{(v)}; S^{(v)}, O^{(v)}|Y^c) + \widetilde{\mathcal{K}}_4\right)}{NC}},
$$

*where $\widetilde{\mathcal{K}}_1 = \widetilde{\mathcal{K}}_3 = \mathcal{O}(C)$, $\widetilde{\mathcal{K}}_2 = \mathcal{O}(\sqrt{C})$, $\widetilde{\mathcal{K}}_4$ is constant of order $\widetilde{\mathcal{O}}(1)$ as $N, V \to \infty$, and $\overline{\text{gen}}_{rec}(Q, X, Y)$ is the generalization error related to the view reconstruction quality. Moreover, the generalization error bound becomes increasingly tighter during the optimization of the problem (16).*

# 3 Experiments

## 3.1 Datasets and metrics

In our experiments, we utilize six popular multi-view multi-label datasets to validate the performance of our TDLSR, i.e., Corel 5k [5], ESPGame [1], IAPRTC12 [8], Mirflickr [16], Pascal07 [6], OBJECT [13]. In accordance with [25, 38], we select six metrics to construct a comprehensive evaluation system, i.e., Hamming Loss (HL), Ranking Loss (RL), OneError (OE), Coverage (Cov), Average Precision (AP), and Area Under Curve (AUC). To facilitate comparison, we present 1-HL, 1-OE, 1-Cov, and 1-RL values in the report, where higher values correspond to better performance.

## 3.2 Comparison methods

To measure the advancement of our TDLSR, nine state-of-the-art methods are selected for comparison experiments, i.e., AIMNet [23], DICNet [25], DIMC [38], iMVWL [33], LMVCAT [26], MTD [24], SIP [27], LVSL [43], DM2L [28]. Specifically, the first seven methods can simultaneously address the issues of missing views and labels. LVSL is a MvMLC method unable to handle missing data.

Thus, we use the mean of available instances to complete the missing views and fill the unknown labels with "0". DM2L is a kernel-based nonlinear method for incomplete multi-label learning. Then, we concatenate all views into a single-view representation for the execution of DM2L. All parameters of compared methods are configured as the recommended values in their original codes.

### 3.3 Implementation details

Each dataset is divided into training, validation and test sets in the ratio of 7:1:2. To simulate the partial view setting, a specified proportion of instances based on the Partial Example Ratio (PER), are randomly marked as unavailable in each view. Additionally, we ensure that each sample contains at least one complete view to avoid invalid cases. For weak supervision, we introduce label omissions for both positive and negative tags in each category applying the same proportion determined by the Label Missing Ratio (LMR). The process of constructing incomplete data is repeated multiple times to mitigate the impact of experimental randomness. Our model is implemented by PyTorch on one NVIDIA GeForce RTX 4090 GPU of 24GB memory.

### 3.4 Experimental results and analysis

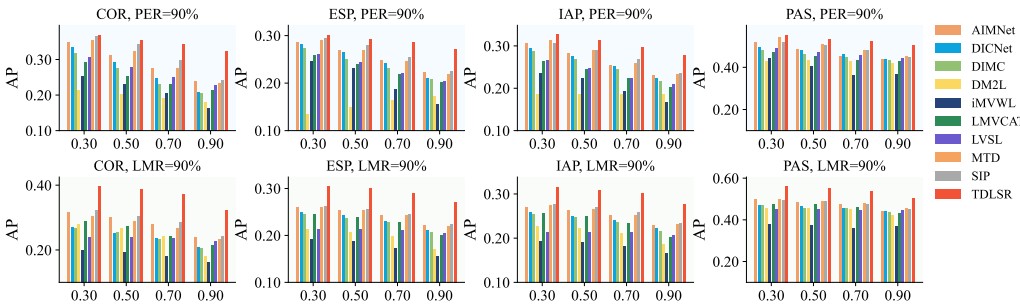

Figure 3: Experimental results on four datasets with PER and LMR changing from $30\%$ to $90\%$.

To evaluate the effectiveness of our TDLSR in handling absent views and labels, we benchmark it against nine closely related algorithms across six datasets with varying levels of data sparsity. The proportion of missing views (PER) and labels (LMR) encompasses values of $\{30\%, 50\%, 70\%, 90\%\}$. The mean and standard deviation of the results with PER and LMR fixed at $50\%$ are reported in Table 1. Besides, the average ranking of each algorithm based on the six metrics is calculated to perform a thorough assessment. Fig. 3 illustrates the variation in AP as PER and LMR changes from $30\%$ to $90\%$. The other relevant results will be presented in the Appendix.

Drawing from the comparison results, we have the following observations: (i) Our method exhibits outstanding performance on almost all metrics across all six datasets. As shown in Table 1, despite the fluctuating rankings of other methods, TDLSR consistently holds the top position. Therefore, our method effectively addresses the iMvMLC problem and maintains stable outcomes. (ii) SIP and MTD are top-performing methods that always appear among the top three. The reason our method surpasses these leading approaches lies in the mutual information optimization, which constrains complex interactions between representations that are insufficiently addressed by MTD. It also overcomes the limitations of SIP by accounting for the impact of private features and transmitting label correlation information to refine label prototypes. Compared to AIMNet that similarly engages in view recovery, our method achieves an improvement in AP from 0.40 to 0.45 on Corel 5k, which demonstrates that our proximity-aware strategy can greatly suppress reconstruction noise. Achieving over a $10\%$ performance gain against traditional multi-label methods like DM2L and deep learning frameworks such as DIMC and DICNet that disregard label correlation, our approach highlights its strength in capturing high-level view representations and leveraging label dependencies to enhance overall performance. (iii) As depicted in Fig. 3, our method exhibits remarkable performance and strong robustness across a wide range of missing ratios. Moreover, our method is particularly well-suited for highly incomplete settings. For instance, when PER reaches 90% on Corel5k and ESPGame, all baseline methods collapse, while our approach continues to deliver commendable results.

Table 1: Experimental results of nine methods on the six datasets with 50% PER and 50% LMR. 'Ave.R' refers to the mean ranking of the corresponding method across all six metrics.

| DATA | METRIC | AIMNet | DICNet | DIMC | DM2L | iMVWL | LMVCAT | LVSL | MTD | SIP | TDLSR |
|---|---|---|---|---|---|---|---|---|---|---|---|
| COR | 1-HL | $0.988_{0.000}$ | $0.987_{0.000}$ | $0.987_{0.000}$ | $0.987_{0.000}$ | $0.978_{0.000}$ | $0.986_{0.000}$ | $0.987_{0.000}$ | $0.988_{0.000}$ | $0.988_{0.000}$ | $0.988_{0.000}$ |
| | 1-OE | $0.478_{0.011}$ | $0.460_{0.012}$ | $0.446_{0.009}$ | $0.378_{0.014}$ | $0.308_{0.017}$ | $0.448_{0.011}$ | $0.353_{0.017}$ | $0.492_{0.011}$ | $0.492_{0.014}$ | $0.541_{0.014}$ |
| | 1-Cov | $0.766_{0.004}$ | $0.726_{0.007}$ | $0.709_{0.008}$ | $0.640_{0.007}$ | $0.701_{0.003}$ | $0.720_{0.006}$ | $0.720_{0.005}$ | $0.754_{0.005}$ | $0.780_{0.004}$ | $0.801_{0.009}$ |
| | 1-RL | $0.900_{0.002}$ | $0.881_{0.004}$ | $0.874_{0.004}$ | $0.843_{0.004}$ | $0.864_{0.002}$ | $0.876_{0.004}$ | $0.879_{0.004}$ | $0.893_{0.004}$ | $0.908_{0.003}$ | $0.917_{0.004}$ |
| | AP | $0.404_{0.005}$ | $0.381_{0.006}$ | $0.370_{0.005}$ | $0.318_{0.005}$ | $0.281_{0.005}$ | $0.379_{0.006}$ | $0.311_{0.005}$ | $0.410_{0.007}$ | $0.414_{0.006}$ | $0.450_{0.006}$ |
| | AUC | $0.903_{0.002}$ | $0.883_{0.004}$ | $0.877_{0.004}$ | $0.846_{0.004}$ | $0.867_{0.002}$ | $0.879_{0.002}$ | $0.882_{0.002}$ | $0.895_{0.002}$ | $0.910_{0.002}$ | $0.919_{0.004}$ |
| | AVE | 3.5 | 5 | 7.333 | 9 | 9.5 | 6.833 | 7.333 | 3.167 | 2.333 | 1 |
| ESP | 1-HL | $0.983_{0.000}$ | $0.983_{0.000}$ | $0.983_{0.000}$ | $0.983_{0.000}$ | $0.972_{0.000}$ | $0.982_{0.000}$ | $0.983_{0.000}$ | $0.983_{0.000}$ | $0.983_{0.000}$ | $0.983_{0.000}$ |
| | 1-OE | $0.442_{0.006}$ | $0.440_{0.009}$ | $0.431_{0.009}$ | $0.302_{0.008}$ | $0.343_{0.010}$ | $0.431_{0.006}$ | $0.365_{0.006}$ | $0.452_{0.007}$ | $0.450_{0.006}$ | $0.477_{0.007}$ |
| | 1-Cov | $0.621_{0.003}$ | $0.601_{0.003}$ | $0.586_{0.004}$ | $0.532_{0.003}$ | $0.548_{0.004}$ | $0.587_{0.003}$ | $0.578_{0.002}$ | $0.617_{0.004}$ | $0.622_{0.004}$ | $0.646_{0.003}$ |
| | 1-RL | $0.845_{0.002}$ | $0.836_{0.002}$ | $0.830_{0.002}$ | $0.804_{0.002}$ | $0.807_{0.002}$ | $0.827_{0.002}$ | $0.829_{0.001}$ | $0.843_{0.002}$ | $0.847_{0.002}$ | $0.859_{0.002}$ |
| | AP | $0.305_{0.003}$ | $0.300_{0.003}$ | $0.294_{0.003}$ | $0.229_{0.003}$ | $0.243_{0.004}$ | $0.293_{0.003}$ | $0.266_{0.003}$ | $0.309_{0.003}$ | $0.309_{0.004}$ | $0.328_{0.004}$ |
| | AUC | $0.850_{0.001}$ | $0.841_{0.002}$ | $0.835_{0.002}$ | $0.808_{0.001}$ | $0.813_{0.002}$ | $0.832_{0.001}$ | $0.834_{0.001}$ | $0.847_{0.002}$ | $0.851_{0.002}$ | $0.863_{0.002}$ |
| | AVE | 3.833 | 4.5 | 5.833 | 9.667 | 9.167 | 7.333 | 7.167 | 3.833 | 2.667 | 1 |
| IAP | 1-HL | $0.981_{0.000}$ | $0.981_{0.000}$ | $0.981_{0.000}$ | $0.980_{0.000}$ | $0.969_{0.000}$ | $0.980_{0.000}$ | $0.981_{0.000}$ | $0.981_{0.000}$ | $0.981_{0.000}$ | $0.981_{0.000}$ |
| | 1-OE | $0.457_{0.007}$ | $0.464_{0.008}$ | $0.454_{0.006}$ | $0.378_{0.008}$ | $0.351_{0.008}$ | $0.433_{0.009}$ | $0.377_{0.007}$ | $0.479_{0.007}$ | $0.459_{0.005}$ | $0.491_{0.008}$ |
| | 1-Cov | $0.675_{0.004}$ | $0.649_{0.005}$ | $0.630_{0.005}$ | $0.555_{0.004}$ | $0.565_{0.004}$ | $0.646_{0.004}$ | $0.605_{0.004}$ | $0.670_{0.004}$ | $0.678_{0.003}$ | $0.706_{0.005}$ |
| | 1-RL | $0.884_{0.001}$ | $0.874_{0.002}$ | $0.868_{0.002}$ | $0.837_{0.002}$ | $0.833_{0.002}$ | $0.868_{0.002}$ | $0.857_{0.002}$ | $0.882_{0.002}$ | $0.886_{0.001}$ | $0.899_{0.002}$ |
| | AP | $0.329_{0.003}$ | $0.326_{0.003}$ | $0.318_{0.002}$ | $0.254_{0.002}$ | $0.236_{0.002}$ | $0.313_{0.004}$ | $0.262_{0.002}$ | $0.340_{0.002}$ | $0.331_{0.003}$ | $0.358_{0.004}$ |
| | AUC | $0.885_{0.001}$ | $0.876_{0.002}$ | $0.870_{0.001}$ | $0.838_{0.001}$ | $0.835_{0.001}$ | $0.870_{0.002}$ | $0.859_{0.001}$ | $0.883_{0.002}$ | $0.887_{0.001}$ | $0.899_{0.002}$ |
| | AVE | 4 | 4.333 | 6 | 8.833 | 9.833 | 6.833 | 8 | 2.833 | 2.833 | 1 |
| MIR | 1-HL | $0.890_{0.001}$ | $0.890_{0.001}$ | $0.890_{0.001}$ | $0.876_{0.001}$ | $0.840_{0.004}$ | $0.880_{0.004}$ | $0.877_{0.001}$ | $0.893_{0.001}$ | $0.890_{0.001}$ | $0.896_{0.001}$ |
| | 1-OE | $0.646_{0.009}$ | $0.647_{0.010}$ | $0.645_{0.008}$ | $0.533_{0.008}$ | $0.511_{0.016}$ | $0.639_{0.009}$ | $0.609_{0.007}$ | $0.667_{0.006}$ | $0.654_{0.007}$ | $0.690_{0.009}$ |
| | 1-Cov | $0.673_{0.003}$ | $0.661_{0.004}$ | $0.657_{0.003}$ | $0.615_{0.002}$ | $0.588_{0.013}$ | $0.665_{0.002}$ | $0.624_{0.002}$ | $0.681_{0.002}$ | $0.675_{0.003}$ | $0.694_{0.003}$ |
| | 1-RL | $0.874_{0.002}$ | $0.869_{0.003}$ | $0.867_{0.003}$ | $0.835_{0.004}$ | $0.809_{0.014}$ | $0.862_{0.003}$ | $0.847_{0.001}$ | $0.878_{0.001}$ | $0.873_{0.002}$ | $0.888_{0.002}$ |
| | AP | $0.599_{0.003}$ | $0.595_{0.007}$ | $0.592_{0.006}$ | $0.519_{0.003}$ | $0.494_{0.017}$ | $0.589_{0.004}$ | $0.548_{0.003}$ | $0.614_{0.004}$ | $0.603_{0.005}$ | $0.631_{0.004}$ |
| | AUC | $0.861_{0.001}$ | $0.855_{0.002}$ | $0.854_{0.002}$ | $0.828_{0.001}$ | $0.801_{0.017}$ | $0.852_{0.002}$ | $0.839_{0.002}$ | $0.864_{0.001}$ | $0.859_{0.002}$ | $0.875_{0.001}$ |
| | AVE | 3.833 | 4.667 | 6.167 | 9 | 10 | 6.667 | 8 | 2 | 3.667 | 1 |
| OBJ | 1-HL | $0.948_{0.001}$ | $0.948_{0.001}$ | $0.947_{0.001}$ | $0.935_{0.000}$ | $0.899_{0.002}$ | $0.940_{0.003}$ | $0.935_{0.001}$ | $0.949_{0.001}$ | $0.948_{0.001}$ | $0.953_{0.001}$ |
| | 1-OE | $0.619_{0.015}$ | $0.601_{0.011}$ | $0.594_{0.012}$ | $0.537_{0.011}$ | $0.465_{0.018}$ | $0.604_{0.016}$ | $0.450_{0.008}$ | $0.627_{0.011}$ | $0.626_{0.009}$ | $0.685_{0.011}$ |
| | 1-Cov | $0.806_{0.006}$ | $0.794_{0.006}$ | $0.793_{0.006}$ | $0.768_{0.005}$ | $0.744_{0.008}$ | $0.796_{0.008}$ | $0.759_{0.006}$ | $0.812_{0.006}$ | $0.809_{0.006}$ | $0.834_{0.007}$ |
| | 1-RL | $0.888_{0.005}$ | $0.876_{0.004}$ | $0.875_{0.004}$ | $0.860_{0.004}$ | $0.833_{0.006}$ | $0.878_{0.006}$ | $0.850_{0.004}$ | $0.890_{0.005}$ | $0.889_{0.004}$ | $0.910_{0.004}$ |
| | AP | $0.639_{0.010}$ | $0.627_{0.009}$ | $0.623_{0.010}$ | $0.577_{0.009}$ | $0.512_{0.014}$ | $0.630_{0.012}$ | $0.537_{0.008}$ | $0.649_{0.009}$ | $0.649_{0.006}$ | $0.692_{0.009}$ |
| | AUC | $0.897_{0.004}$ | $0.886_{0.004}$ | $0.885_{0.004}$ | $0.872_{0.004}$ | $0.846_{0.004}$ | $0.888_{0.004}$ | $0.864_{0.004}$ | $0.900_{0.004}$ | $0.898_{0.004}$ | $0.918_{0.004}$ |
| | AVE | 4 | 5.833 | 6.833 | 8.167 | 9.833 | 5.333 | 9 | 2 | 3 | 1 |
| PAS | 1-HL | $0.931_{0.001}$ | $0.931_{0.000}$ | $0.931_{0.001}$ | $0.927_{0.001}$ | $0.882_{0.004}$ | $0.915_{0.005}$ | $0.928_{0.001}$ | $0.933_{0.001}$ | $0.932_{0.001}$ | $0.933_{0.001}$ |
| | 1-OE | $0.462_{0.009}$ | $0.443_{0.007}$ | $0.435_{0.010}$ | $0.419_{0.006}$ | $0.366_{0.039}$ | $0.433_{0.017}$ | $0.418_{0.008}$ | $0.473_{0.008}$ | $0.468_{0.006}$ | $0.495_{0.013}$ |
| | 1-Cov | $0.781_{0.007}$ | $0.749_{0.003}$ | $0.738_{0.010}$ | $0.720_{0.004}$ | $0.674_{0.011}$ | $0.759_{0.006}$ | $0.738_{0.003}$ | $0.790_{0.006}$ | $0.778_{0.004}$ | $0.817_{0.004}$ |
| | 1-RL | $0.830_{0.006}$ | $0.803_{0.002}$ | $0.792_{0.008}$ | $0.778_{0.003}$ | $0.736_{0.011}$ | $0.808_{0.006}$ | $0.797_{0.002}$ | $0.836_{0.005}$ | $0.828_{0.004}$ | $0.862_{0.004}$ |
| | AP | $0.548_{0.007}$ | $0.517_{0.004}$ | $0.510_{0.008}$ | $0.482_{0.005}$ | $0.438_{0.022}$ | $0.524_{0.009}$ | $0.486_{0.005}$ | $0.562_{0.005}$ | $0.552_{0.006}$ | $0.590_{0.008}$ |
| | AUC | $0.851_{0.005}$ | $0.827_{0.002}$ | $0.817_{0.008}$ | $0.806_{0.003}$ | $0.767_{0.011}$ | $0.830_{0.006}$ | $0.823_{0.002}$ | $0.855_{0.005}$ | $0.848_{0.005}$ | $0.880_{0.003}$ |
| | AVE | 3.5 | 5.667 | 7.167 | 8.667 | 10 | 6 | 7.5 | 2 | 3.5 | 1 |

## 3.5 Ablation Study

The ablation experiments are conducted to deeply investigate the effect of the three crucial modules of TDLSR, i.e., proximity-aware graph attention recovery mechanism ($S_1$), information theory-driven representation extraction framework ($S_2$), multi-label semantic and label-specific representation learning ($S_3$). After individually removing $S_1$, $S_2$ and $S_3$, we use mean imputation for missing samples, rely solely on a single multilayer perceptron (MLP) for feature extraction while discarding $\mathcal{L}_{IB}$, and directly employ a classifier based on fully connected layers without exploring label semantics, respectively. Based on the ablation results provided in Table 2, we have the following observations: (i) When either module is removed, the performance declines, which indicates the effectiveness and thoughtful design of our TDLSR. (ii) The recovery mechanism is crucial for enhancing performance, as it provides downstream modules with rich feature information. Moreover, feature separation outperforms the single-channel representations and incorporating category semantic learning enhances performance beyond that of classifier-only approaches. It demonstrates our thorough consideration of feature extraction and label associations.

## 4 Application to Comprehensive Potential Prediction of Players

To validate the practical applicability of our TDLSR, we evaluate its ability to predict multiple attributes of NBA players under partial data missingness. The NBA dataset was collected from Basketball-Reference [2], which contains 16,992 player-season records from the 2002-2022 seasons. Each sample is structured across six principal statistical views including scoring efficiency, rebounding and physical metrics, technical statistics, advanced efficiency metrics, player background, and season context. The prediction tasks comprise career stage classification, positional identification and awards prediction. Career stages are partitioned into early (first 25%), peak (middle 50%), and late (final

Table 2: Ablation study on Pascal07, OBJECT and Mirflickr with PER=50% and LMR=50%. '✓' and '✗' represent the used and not used corresponding item, respectively.

| $S_1$ | $S_2$ | $S_3$ | Pascal07 | | | | OBJECT | | | | Mirflickr | | | |
|---|---|---|---|---|---|---|---|---|---|---|---|---|---|---|
| | | | AP | AUC | 1-RL | 1-OE | AP | AUC | 1-RL | 1-OE | AP | AUC | 1-RL | 1-OE |
| ✗ | ✓ | ✓ | 0.546 | 0.852 | 0.830 | 0.455 | 0.650 | 0.903 | 0.894 | 0.633 | 0.594 | 0.859 | 0.872 | 0.649 |
| ✓ | ✗ | ✓ | 0.576 | 0.874 | 0.853 | 0.478 | 0.687 | 0.914 | 0.906 | 0.678 | 0.614 | 0.872 | 0.881 | 0.652 |
| ✓ | ✓ | ✗ | 0.582 | 0.874 | 0.857 | 0.486 | 0.690 | 0.912 | 0.904 | 0.680 | 0.616 | 0.870 | 0.882 | 0.659 |
| ✓ | ✓ | ✓ | **0.599** | **0.882** | **0.864** | **0.519** | **0.702** | **0.924** | **0.916** | **0.688** | **0.631** | **0.875** | **0.889** | **0.687** |

25%) phases according to each player's professional timeline. Player positions (PG, SG, SF, PF, C) are represented using one-hot encoding, and multiple binary indicators corresponding to honors such as MVP awards and Defensive Player of the Year, are included to provide multi-task objectives and comprehensive modeling of player achievements.

Across varying levels of data incompleteness, with PER and LMR ranging from 50% to 90%, our method consistently surpasses baseline approaches, demonstrating superior robustness and reliability in attribute prediction. Moreover, all comparison methods fail to surpass an AP of 0.6 at 90% missing ratio, whereas our TDLSR achieves 0.668. Despite incomplete technical statistics and constrained annotation resources, our method remains effective in predicting player potential, including career development and honor attainment, which offers considerable promise for real-world applications.

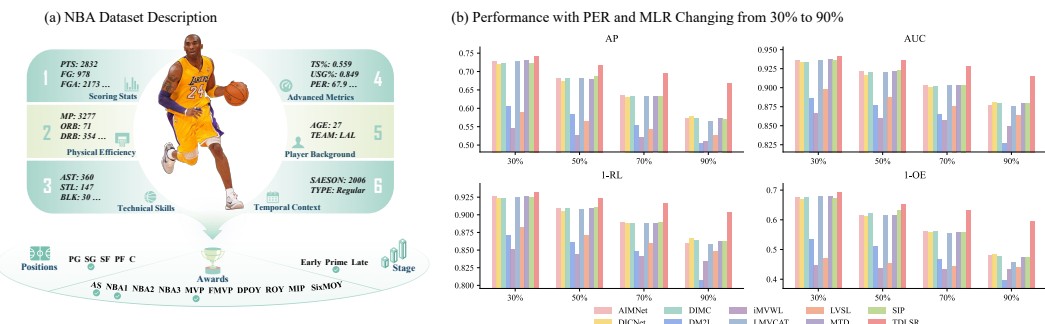

Figure 4: Overview of the NBA dataset and performance with missing views and labels.

# 5   Conclusion

In this paper, we propose a theory-driven label-specific representation (TDLSR) framework for addressing the iMvMLC problem. Specifically, structural dependencies are modeled through graph attention mechanism inside each view for recovery, with the reconstruction fidelity adaptively tailored to enhance classification efficacy. Meanwhile, we construct a universal feature extraction model, where mutual information optimization serves to regulate information shift, interaction and orthogonality between representations. On the basis of complete feature semantic separation, we independently interact each representation with label prototypes that encode correlation semantic, aiming to extract label-specific discriminative features and uncover representation-sensitive label subsets. Moreover, we theoretically validate the effectiveness of representation learning and its influence on the generalization performance. Finally, the superiority of TDLSR are validated through extensive experiments and application to the NBA dataset. In the future, we will explore leveraging the prior knowledge embedded in LLM to facilitate label semantic learning.

# Acknowledgments

This work was supported by the National Natural Science Foundation of China under Grant No. 62376281, the NSF for Distinguished Young Scholars under Grant No. 62425607, and the Key NSF of China under Grant No. 62136005.

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

# A Derivation and Implementation of the Loss Function for the Universal View Extraction Model

## A.1 Derivation of the Loss Function

The universal view extraction model under mutual information constraints is

$$
\begin{cases}
\max \underbrace{\sum_{v=1}^{V} I(\boldsymbol{s}^{(v)}; \boldsymbol{z}^{(v)})}_{\text{information shift}} + \underbrace{\sum_{v=1}^{V}\sum_{v^* \neq v}^{V} (I(\boldsymbol{s}^{(v^*)}|\boldsymbol{z}^{(v^*)}; \boldsymbol{s}^{(v)}|\boldsymbol{z}^{(v)}) - I(\boldsymbol{o}^{(v^*)}|\boldsymbol{z}^{(v^*)}; \boldsymbol{o}^{(v)}|\boldsymbol{z}^{(v)}))}_{\text{information interaction}} \\[2mm]
s.t. \quad \underbrace{\sum_{v=1}^{V} I(\boldsymbol{s}^{(v)}|\boldsymbol{z}^{(v)}; \boldsymbol{o}^{(v)}|\boldsymbol{z}^{(v)}) = 0}_{\text{information orthogonality}} \,.
\end{cases}
\tag{16}
$$

By incorporating variational derivations from the information theory, we arrive at the following objective function for optimization:

$$
\mathcal{L}_{IB} = - \underbrace{\mathbb{E}_{p\left(\boldsymbol{z}^{(v)}, \boldsymbol{s}^{(v)}\right)}\left[\log q\left(\boldsymbol{s}^{(u)}|\boldsymbol{z}^{(v)}\right)\right]}_{\text{lower bound for information shift}} - \underbrace{\int\int p(\boldsymbol{s}^{(v)}|\boldsymbol{z}^{(v)}, \boldsymbol{s}^{(v^*)}|\boldsymbol{z}^{(v^*)}) \log q(\boldsymbol{s}^{(v)}|\boldsymbol{s}^{v^*}) d\boldsymbol{s}^{(v^*)} d\boldsymbol{s}^{(v)}}_{\text{lower bound for shard information interaction}}
$$

$$
+ \underbrace{D_{KL}(p(\boldsymbol{o}^{(v^*)}|\boldsymbol{z}^{(v^*)}; \boldsymbol{o}^{(v)}|\boldsymbol{z}^{(v)}) \| p(\boldsymbol{o}^{(v^*)}|\boldsymbol{z}^{(v^*)}) q(\boldsymbol{o}^{(v)}|\boldsymbol{o}^{(v^*)}))}_{\text{upper bound for specific information interaction}}
$$

$$
+ \beta \underbrace{D_{KL}(p(\boldsymbol{s}^{(v)}|\boldsymbol{z}^{(v)}; \boldsymbol{o}^{(v)}|\boldsymbol{z}^{(v)}) \| p(\boldsymbol{o}^{(v)}|\boldsymbol{z}^{(v)}) q(\boldsymbol{s}^{(v)}|\boldsymbol{o}^{(v)}))}_{\text{upper bound for information orthogonality}} \,.
\tag{17}
$$

*Proof.* Due to the challenge of distribution inference and the complexity of high-dimensional integration, directly computing mutual information is impractical. Therefore, it is common to ensure stability and feasibility by optimizing an estimable bound of mutual information. For the information shift term, we have the following transformation based on the definitions of mutual information and information entropy:

$$
I(\boldsymbol{s}^{(v)}; \boldsymbol{z}^{(v)})
$$
$$
= \int\int p(\boldsymbol{s}^{(v)}, \boldsymbol{z}^{(v)}) \log \frac{p(\boldsymbol{z}^{(v)}|\boldsymbol{s}^{(v)})}{p(\boldsymbol{z}^{(v)})} d\boldsymbol{s}^{(v)} d\boldsymbol{z}^{(v)}
$$
$$
= \int\int p(\boldsymbol{s}^{(v)}, \boldsymbol{z}^{(v)}) \log p(\boldsymbol{z}^{(v)}|\boldsymbol{s}^{(v)}) d\boldsymbol{s}^{(v)} d\boldsymbol{z}^{(v)} + \int p(\boldsymbol{z}^{(v)}|\boldsymbol{s}^{(v)}) \int p(\boldsymbol{z}^{(v)}) \log \frac{1}{p(\boldsymbol{z}^{(v)})} d\boldsymbol{s}^{(v)} d\boldsymbol{z}^{(v)}
$$
$$
= \int\int p(\boldsymbol{s}^{(v)}, \boldsymbol{z}^{(v)}) \log p(\boldsymbol{z}^{(v)}|\boldsymbol{s}^{(v)}) d\boldsymbol{s}^{(v)} d\boldsymbol{z}^{(v)} + H(\boldsymbol{z}^{(v)}).
\tag{18}
$$

Since Eq. (18) is intractable, we approximate $p(\boldsymbol{z}^{(v)}|\boldsymbol{s}^{(v)})$ using a stochastic variational distribution $q(\boldsymbol{z}^{(v)}|\boldsymbol{s}^{(v)})$, which can be reasonably estimated. Combing the fact that entropy $H(\boldsymbol{z}^{(v)}) \geq 0$, we can further obtain the lower bound:

$$
I(\boldsymbol{s}^{(v)}; \boldsymbol{z}^{(v)})
\tag{19}
$$
$$
\geq \int\int p(\boldsymbol{z}^{(v)}, \boldsymbol{s}^{(v)}) \log p(\boldsymbol{s}^{(v)}|\boldsymbol{z}^{(v)}) d\boldsymbol{s}^{(v)} d\boldsymbol{z}^{(v)}
$$
$$
= \int p(\boldsymbol{z}^{(v)}) \int p(\boldsymbol{s}^{(v)}|\boldsymbol{z}^{(v)}) \log q(\boldsymbol{z}^{(v)}|\boldsymbol{s}^{(v)}) d\boldsymbol{s}^{(v)} d\boldsymbol{z}^{(v)}
$$
$$
+ \int p(\boldsymbol{s}^{(v)}) \int p(\boldsymbol{z}^{(v)}|\boldsymbol{s}^{(v)}) \log \frac{p(\boldsymbol{z}^{(v)}|\boldsymbol{s}^{(v)})}{q(\boldsymbol{z}^{(v)}|\boldsymbol{s}^{(v)})} d\boldsymbol{s}^{(v)} d\boldsymbol{z}^{(v)}.
$$

The Kullback-Leibler divergence is denoted as

$$D_{KL}(p(\boldsymbol{z}^{(v)}|\boldsymbol{s}^{(v)})\|q(\boldsymbol{z}^{(v)}|\boldsymbol{s}^{(v)})) = \int p(\boldsymbol{z}^{(v)}|\boldsymbol{s}^{(v)})\log\frac{p(\boldsymbol{z}^{(v)}|\boldsymbol{s}^{(v)})}{q(\boldsymbol{z}^{(v)}|\boldsymbol{s}^{(v)})}dz^{(v)} \geq 0 \qquad (20)$$

Thus, we have

$$I(\boldsymbol{s}^{(v)};\boldsymbol{z}^{(v)})$$

$$= \int p(\boldsymbol{z}^{(v)})\int p(\boldsymbol{s}^{(v)}|\boldsymbol{z}^{(v)})\log q(\boldsymbol{z}^{(v)}|\boldsymbol{s}^{(v)})ds^{(v)}dz^{(v)}$$

$$+ \int p(\boldsymbol{s}^{(v)})D_{KL}(p(\boldsymbol{z}^{(v)}|\boldsymbol{s}^{(v)})\|q(\boldsymbol{z}^{(v)}|\boldsymbol{s}^{(v)}))ds^{(v)}dz^{(v)}$$

$$\geq \int p(\boldsymbol{z}^{(v)})\int p(\boldsymbol{s}^{(v)}|\boldsymbol{z}^{(v)})\log q(\boldsymbol{z}^{(v)}|\boldsymbol{s}^{(v)})ds^{(v)}dz^{(v)} \qquad (21)$$

$$= \int p(\boldsymbol{s}^{(v)},\boldsymbol{z}^{(v)})\log q(\boldsymbol{z}^{(v)}|\boldsymbol{s}^{(v)})ds^{(v)}dz^{(v)}$$

$$= \mathbb{E}_{p(\boldsymbol{z}^{(v)},\boldsymbol{s}^{(v)})}\left[\log q(\boldsymbol{z}^{(v)}|\boldsymbol{s}^{(v)})\right].$$

For the second term, based on the definition and non-negativity of entropy, a lower bound can be derived by introducing another estimable variational distribution $q(\boldsymbol{s}^{(v)}|\boldsymbol{s}^{(v^*)})$. By expanding the mutual information in its integral form, we have

$$I(\boldsymbol{s}^{(v^*)}|\boldsymbol{z}^{(v^*)};\boldsymbol{s}^{(v)}|\boldsymbol{z}^{(v)}) = \iint p(\boldsymbol{s}^{(v)}|\boldsymbol{z}^{(v)},\boldsymbol{s}^{(v^*)}|\boldsymbol{z}^{(v^*)})\log\frac{p(\boldsymbol{s}^{(v)}|\boldsymbol{z}^{(v)},\boldsymbol{s}^{(v^*)}|\boldsymbol{z}^{(v^*)})}{p(\boldsymbol{s}^{(v^*)}|\boldsymbol{z}^{(v^*)})p(\boldsymbol{s}^{(v)}|\boldsymbol{z}^{(v)})}ds^{(v^*)}ds^{(v)},$$
$$(22)$$

Considering $p(\boldsymbol{s}^{(v^*)}|\boldsymbol{z}^{(v^*)},\boldsymbol{s}^{(v)}|\boldsymbol{z}^{(v)}) = p(\boldsymbol{s}^{(v)}|\boldsymbol{z}^{(v)}/\boldsymbol{s}^{(v^*)}|\boldsymbol{z}^{(v^*)})p(\boldsymbol{s}^{(v^*)}|\boldsymbol{z}^{(v^*)})$, we can get

$$I(\boldsymbol{s}^{(v^*)}|\boldsymbol{z}^{(v^*)};\boldsymbol{s}^{(v)}|\boldsymbol{z}^{(v)}) \qquad (23)$$

$$= \iint p(\boldsymbol{s}^{(v)}|\boldsymbol{z}^{(v)}/\boldsymbol{s}^{(v^*)}|\boldsymbol{z}^{(v^*)})p(\boldsymbol{s}^{(v^*)}|\boldsymbol{z}^{(v^*)})\log\frac{p(\boldsymbol{s}^{(v)}|\boldsymbol{z}^{(v)}/\boldsymbol{s}^{(v^*)}|\boldsymbol{z}^{(v^*)})p(\boldsymbol{s}^{(v^*)}|\boldsymbol{z}^{(v^*)})}{p(\boldsymbol{s}^{(v^*)}|\boldsymbol{z}^{(v^*)})p(\boldsymbol{s}^{(v)}/\boldsymbol{z}^{(v)})}ds^{(v^*)}ds^{(v)}$$

$$= \iint p(\boldsymbol{s}^{(v^*)}|\boldsymbol{z}^{(v^*)}/\boldsymbol{s}^{(v)}|\boldsymbol{z}^{(v)})p(\boldsymbol{s}^{(v^*)}|\boldsymbol{z}^{(v^*)})\log p(\boldsymbol{s}^{(v)}|\boldsymbol{z}^{(v)}/\boldsymbol{s}^{(v^*)}|\boldsymbol{z}^{(v^*)})ds^{(v^*)}ds^{(v)}$$

$$+ \iint p(\boldsymbol{s}^{(v)}|\boldsymbol{z}^{(v)}/\boldsymbol{s}^{(v^*)}|\boldsymbol{z}^{(v^*)})p(\boldsymbol{s}^{(v)}|\boldsymbol{z}^{(v)})\log\frac{1}{p(\boldsymbol{s}^{(v)}|\boldsymbol{z}^{(v)})}ds^{(v^*)}ds^{(v)}.$$

Since $H(\boldsymbol{s}^{(v)}|\boldsymbol{z}^{(v)}) = -\int p(\boldsymbol{s}^{(v)}|\boldsymbol{z}^{(v)})\log p(\boldsymbol{s}^{(v)}|\boldsymbol{z}^{(v)})ds^{(v)} \geq 0$, we obtain

$$I(\boldsymbol{s}^{(v^*)}|\boldsymbol{z}^{(v^*)};\boldsymbol{s}^{(v)}|\boldsymbol{z}^{(v)})$$

$$\geq \iint p(\boldsymbol{s}^{(v)}|\boldsymbol{z}^{(v)}/\boldsymbol{s}^{(v^*)}|\boldsymbol{z}^{(v^*)})p(\boldsymbol{s}^{(v^*)}|\boldsymbol{z}^{(v^*)})\log p(\boldsymbol{s}^{(v)}|\boldsymbol{z}^{(v)}/\boldsymbol{s}^{(v^*)}|\boldsymbol{z}^{(v^*)})ds^{(v^*)}ds^{(v)}$$

$$= \iint p(\boldsymbol{s}^{(v)}|\boldsymbol{z}^{(v)}/\boldsymbol{s}^{(v^*)}|\boldsymbol{z}^{(v^*)})p(\boldsymbol{s}^{(v^*)}|\boldsymbol{z}^{(v^*)})\log q(\boldsymbol{s}^{(v)}|\boldsymbol{s}^{(v^*)})ds^{(v^*)}ds^{(v)}$$

$$+ \iint p(\boldsymbol{s}^{(v)}|\boldsymbol{z}^{(v)}/\boldsymbol{s}^{(v^*)}|\boldsymbol{z}^{(v^*)})p(\boldsymbol{s}^{(v^*)}|\boldsymbol{z}^{(v^*)})\log\frac{p(\boldsymbol{s}^{(v)}|\boldsymbol{z}^{(v)}/\boldsymbol{s}^{(v^*)}|\boldsymbol{z}^{(v^*)})}{q(\boldsymbol{s}^{(v)}|\boldsymbol{s}^{(v^*)})}ds^{(v^*)}ds^{(v)} \quad (24)$$

Based on the definition of Kullback-Leibler divergence, we have

$$D_{KL}(p(\boldsymbol{s}^{(v)}|\boldsymbol{z}^{(v)}/\boldsymbol{s}^{(v^*)}|\boldsymbol{z}^{(v^*)})\|q(\boldsymbol{s}^{(v)}|\boldsymbol{s}^{(v^*)}))$$

$$= \int p(\boldsymbol{s}^{(v)}|\boldsymbol{z}^{(v)}/\boldsymbol{s}^{(v^*)}|\boldsymbol{z}^{(v^*)})\log\frac{p(\boldsymbol{s}^{(v)}|\boldsymbol{z}^{(v)}/\boldsymbol{s}^{(v^*)}|\boldsymbol{z}^{(v^*)})}{q(\boldsymbol{s}^{(v)}|\boldsymbol{s}^{(v^*)})}ds^{(v)}. \qquad (25)$$

Since $D_{KL}(p(\boldsymbol{s}^{(v)}|\boldsymbol{z}^{(v)}/\boldsymbol{s}^{(v^*)}|\boldsymbol{z}^{(v^*)})\|q(\boldsymbol{s}^{(v)}|\boldsymbol{s}^{(v^*)})) \geq 0$, we have

$$I(\boldsymbol{s}^{(v^*)}|\boldsymbol{z}^{(v^*)};\boldsymbol{s}^{(v)}|\boldsymbol{z}^{(v)}) \geq \iint p(\boldsymbol{s}^{(v)}|\boldsymbol{z}^{(v)}/\boldsymbol{s}^{(v^*)}|\boldsymbol{z}^{(v^*)})p(\boldsymbol{s}^{(v^*)}|\boldsymbol{z}^{(v^*)})\log q(\boldsymbol{s}^{(v)}|\boldsymbol{s}^{(v^*)})ds^{(v^*)}ds^{(v)}$$

$$= \iint p(\boldsymbol{s}^{(v)}|\boldsymbol{z}^{(v)},\boldsymbol{s}^{(v^*)}|\boldsymbol{z}^{(v^*)})\log q(\boldsymbol{s}^{(v)}|\boldsymbol{s}^{(v^*)})ds^{(v^*)}ds^{(v)}.$$

$$(26)$$

For the third term $I(\boldsymbol{o}^{\boldsymbol{v}^*}|\boldsymbol{z}^{(v^*)}; \boldsymbol{o}^{(v)}|\boldsymbol{z}^{(v)})$, we derive the following upper bound:

$$I(\boldsymbol{o}^{\boldsymbol{v}^*}|\boldsymbol{z}^{(v^*)}; \boldsymbol{o}^{(v)}|\boldsymbol{z}^{(v)})$$

$$= \iint p(\boldsymbol{o}^{\boldsymbol{v}^*}|\boldsymbol{z}^{(v^*)}; \boldsymbol{o}^{(v)}|\boldsymbol{z}^{(v)}) \log \frac{p(\boldsymbol{o}^{\boldsymbol{v}^*}|\boldsymbol{z}^{(v^*)}; \boldsymbol{o}^{(v)}|\boldsymbol{z}^{(v)})}{p(\boldsymbol{o}^{(v^*)}|\boldsymbol{z}^{(v^*)})p(\boldsymbol{o}^{(v)}|\boldsymbol{z}^{(v)})} d\boldsymbol{o}^{(v^*)}d\boldsymbol{o}^{(v)}$$

$$= \iint p(\boldsymbol{o}^{(v^*)}|\boldsymbol{z}^{(v^*)}; \boldsymbol{o}^{(v)}|\boldsymbol{z}^{(v)}) \log \frac{p(\boldsymbol{o}^{(v)}|\boldsymbol{z}^{(v)}/\boldsymbol{o}^{(v^*)}|\boldsymbol{z}^{(v^*)})}{p(\boldsymbol{o}^{(v)}|\boldsymbol{z}^{(v)})} d\boldsymbol{o}^{(v^*)}d\boldsymbol{o}^{(v)}$$

$$= \iint p(\boldsymbol{o}^{\boldsymbol{v}^*}|\boldsymbol{z}^{(v^*)}; \boldsymbol{o}^{(v)}|\boldsymbol{z}^{(v)}) \log \frac{p(\boldsymbol{o}^{(v)}|\boldsymbol{z}^{(v)}/\boldsymbol{o}^{(v^*)}|\boldsymbol{z}^{(v^*)})}{q(\boldsymbol{o}^{(v)}|\boldsymbol{o}^{(v^*)})} d\boldsymbol{o}^{(v^*)}d\boldsymbol{o}^{(v)}$$

$$\quad + \iint p(\boldsymbol{o}^{\boldsymbol{v}^*}|\boldsymbol{z}^{(v^*)}; \boldsymbol{o}^{(v)}|\boldsymbol{z}^{(v)}) \log \frac{q(\boldsymbol{o}^{(v)}|\boldsymbol{o}^{(v^*)})}{p(\boldsymbol{o}^{(v)}|\boldsymbol{z}^{(v)})} d\boldsymbol{o}^{(v^*)}d\boldsymbol{o}^{(v)}$$

$$= \iint p(\boldsymbol{o}^{\boldsymbol{v}^*}|\boldsymbol{z}^{(v^*)}; \boldsymbol{o}^{(v)}|\boldsymbol{z}^{(v)}) \log \frac{p(\boldsymbol{o}^{(v)}|\boldsymbol{z}^{(v)}/\boldsymbol{o}^{(v^*)}|\boldsymbol{z}^{(v^*)})}{q(\boldsymbol{o}^{(v)}|\boldsymbol{o}^{(v^*)})} d\boldsymbol{o}^{(v^*)}d\boldsymbol{o}^{(v)}$$

$$\quad - \int p(\boldsymbol{o}^{(v^*)}|\boldsymbol{z}^{(v^*)}/\boldsymbol{o}^{(v)}|\boldsymbol{z}^{(v)})D_{KL}(p(\boldsymbol{o}^{(v)}|\boldsymbol{z}^{(v)})\|q(\boldsymbol{o}^{(v)}|\boldsymbol{o}^{(v^*)}))d\boldsymbol{o}^{(v^*)}$$

$$\leq \iint p(\boldsymbol{o}^{\boldsymbol{v}^*}|\boldsymbol{z}^{(v^*)}; \boldsymbol{o}^{(v)}|\boldsymbol{z}^{(v)}) \log \frac{p(\boldsymbol{o}^{(v)}|\boldsymbol{z}^{(v)}/\boldsymbol{o}^{(v^*)}|\boldsymbol{z}^{(v^*)})}{q(\boldsymbol{o}^{(v)}|\boldsymbol{o}^{(v^*)})} d\boldsymbol{o}^{(v^*)}d\boldsymbol{o}^{(v)}$$

$$= \iint p(\boldsymbol{o}^{\boldsymbol{v}^*}|\boldsymbol{z}^{(v^*)}; \boldsymbol{o}^{(v)}|\boldsymbol{z}^{(v)}) \log \frac{p(\boldsymbol{o}^{\boldsymbol{v}^*}|\boldsymbol{z}^{(v^*)}; \boldsymbol{o}^{(v)}|\boldsymbol{z}^{(v)})}{p(\boldsymbol{o}^{(v^*)}|\boldsymbol{z}^{(v^*)})q(\boldsymbol{o}^{(v)}|\boldsymbol{o}^{(v^*)})} d\boldsymbol{o}^{(v^*)}d\boldsymbol{o}^{(v)}$$

$$= D_{KL}(p(\boldsymbol{o}^{\boldsymbol{v}^*}|\boldsymbol{z}^{(v^*)}; \boldsymbol{o}^{(v)}|\boldsymbol{z}^{(v)})\|p(\boldsymbol{o}^{(v^*)}|\boldsymbol{z}^{(v^*)})q(\boldsymbol{o}^{(v)}|\boldsymbol{o}^{(v^*)})). \tag{27}$$

Similarly, we can derive the upper bound for the last term, which exhibit structure analogous to that of Eq. (27):

$$I(\boldsymbol{s}^{(v)}|\boldsymbol{z}^{(v)}; \boldsymbol{o}^{(v)}|\boldsymbol{z}^{(v)}) \leq D_{KL}(p(\boldsymbol{s}^{(v)}|\boldsymbol{z}^{(v)}; \boldsymbol{o}^{(v)}|\boldsymbol{z}^{(v)})\|p(\boldsymbol{o}^{(v)}|\boldsymbol{z}^{(v)})q(\boldsymbol{s}^{(v)}|\boldsymbol{o}^{(v)})). \tag{28}$$

Combining Eqs. (21), (25), (27) and (28), the objective is naturally transformed into minimizing its upper bound:

$$\mathcal{L}_{IB} = - \underbrace{\mathbb{E}_{p(\boldsymbol{z}^{(v)}, \boldsymbol{s}^{(v)})} \left[ \log q\left(\boldsymbol{z}^{(v)}|\boldsymbol{s}^{(v)}\right) \right]}_{\text{lower bound for information shift}} - \underbrace{\iint p(\boldsymbol{s}^{(v)}|\boldsymbol{z}^{(v)}, \boldsymbol{s}^{(v^*)}|\boldsymbol{z}^{(v^*)}) \log q(\boldsymbol{s}^{(v)}|\boldsymbol{s}^{\boldsymbol{v}^*}) d\boldsymbol{s}^{(v^*)}d\boldsymbol{s}^{(v)}}_{\text{lower bound for shard information interaction}}$$

$$+ \underbrace{D_{KL}(p(\boldsymbol{o}^{(v^*)}|\boldsymbol{z}^{(v^*)}; \boldsymbol{o}^{(v)}|\boldsymbol{z}^{(v)})\|p(\boldsymbol{o}^{(v^*)}|\boldsymbol{z}^{(v^*)})q(\boldsymbol{o}^{(v)}|\boldsymbol{o}^{(v^*)}))}_{\text{upper bound for specific information interaction}}$$

$$+ \beta \underbrace{D_{KL}(p(\boldsymbol{s}^{(v)}|\boldsymbol{z}^{(v)}; \boldsymbol{o}^{(v)}|\boldsymbol{z}^{(v)})\|p(\boldsymbol{o}^{(v)}|\boldsymbol{z}^{(v)})q(\boldsymbol{s}^{(v)}|\boldsymbol{o}^{(v)}))}_{\text{upper bound for information orthogonality}}.$$

$$\tag{29}$$

$\square$

## A.2 Implementation of the Loss Function

Data-driven contrastive learning [22] is used to compute various complex distributions. Specifically, representations are treated as probability vectors over $D$ classes (corresponding to $d$ dimension) via a Softmax activation function, and the joint distribution matrix is obtained by

$$\boldsymbol{P}^{(v,v^*)} = \sum_{i=1}^{N} \left(\boldsymbol{s}_i^{(v)}\right)^T \boldsymbol{s}_i^{(v^*)}, \ \boldsymbol{Q}^{(v,v^*)} = \sum_{i=1}^{N} \left(\boldsymbol{o}_i^{(v)}\right)^T \boldsymbol{o}_i^{(v^*)}, \ \boldsymbol{M}^{(s^v,o^v)} = \sum_{i=1}^{N} \left(\boldsymbol{s}_i^{(v)}\right)^T \boldsymbol{o}_i^{(v^*)}.$$

Due to the strong coupling between $\boldsymbol{s}^{(v)}$ and $\boldsymbol{s}^{(v^*)}$, the variational distribution $q(\boldsymbol{s}^{(v)}|\boldsymbol{s}^{(v^*)})$ can be effectively estimated by the obtained conditional distribution. In contrast, owing to the extremely

weak correlation between $\boldsymbol{o}^{(v)}$ and $\boldsymbol{o}^{(v^*)}$, as well as $\boldsymbol{o}^{(v)}$ and $\boldsymbol{s}^{(v)}$, the distributions $q(\boldsymbol{o}^{(v)}/\boldsymbol{o}^{(v^*)})$ and $q(\boldsymbol{s}^{(v)}/\boldsymbol{o}^{(v)})$ can be approximated by the corresponding marginal distribution. Since the first information shift term is equivalent to the reconstruction loss, we construct a decoder for each view $v$ to obtain $\hat{\boldsymbol{z}}^v$, which is used to approximate the original view. Subsequently, by converting the integral to a summation form, each term in Eq. (29) can be expressed as

$$
\begin{cases}
\displaystyle \int \int p(\boldsymbol{s}^{(v)}|\boldsymbol{z}^{(v)}, \boldsymbol{s}^{(v^*)}|\boldsymbol{z}^{(v^*)}) \log q(\boldsymbol{s}^{(v)}|\boldsymbol{s}^{v^*}) d\boldsymbol{s}^{(v^*)} d\boldsymbol{s}^{(v)} = \sum_{d=1}^{D} \sum_{d'=1}^{D} \boldsymbol{P}_{d,d'}^{(v,v^*)} \log \frac{\boldsymbol{P}_{d,d'}^{(v,v^*)}}{\boldsymbol{P}_{d'}^{(v^*)}} \\[2em]
D_{KL}(p(\boldsymbol{o}^{(v^*)}|\boldsymbol{z}^{(v^*)}; \boldsymbol{o}^{(v)}|\boldsymbol{z}^{(v)}) \| p(\boldsymbol{o}^{(v^*)}|\boldsymbol{z}^{(v^*)}) q(\boldsymbol{o}^{(v)}|\boldsymbol{o}^{(v^*)})) = \sum_{d=1}^{D} \sum_{d'=1}^{D} \boldsymbol{Q}_{d,d'}^{(v,v^*)} \log \frac{\boldsymbol{Q}_{d,d'}^{(v,v^*)}}{(\boldsymbol{Q}_d^{(v)})^\alpha \boldsymbol{Q}_{d'}^{(v^*)}} \\[2em]
D_{KL}(p(\boldsymbol{s}^{(v)}|\boldsymbol{z}^{(v)}; \boldsymbol{o}^{(v)}|\boldsymbol{z}^{(v)}) \| p(\boldsymbol{o}^{(v)}|\boldsymbol{z}^{(v)}) q(\boldsymbol{s}^{(v)}|\boldsymbol{o}^{(v)})) = \sum_{d=1}^{D} \sum_{d'=1}^{D} \boldsymbol{M}_{d,d'}^{(s^v,o^v)} \log \frac{\boldsymbol{M}_{d,d'}^{(s^v,o^v)}}{(\boldsymbol{M}_d^{(s^v)})^\alpha \boldsymbol{M}_{d'}^{(o^v)}},
\end{cases}
\tag{30}
$$

where the Lagrange multiplie $\beta$ is fixed to 1, and the marginal distributions $\boldsymbol{P}_d^{(v)}$ and $\boldsymbol{P}_{d'}^{(v^*)}$ can be obtained by summing over the $d$-th row and the $d'$-th column of the joint distribution matrix $\boldsymbol{P}^{(v,v^*)}$, with other symbols defined as the same. In this conversion, we let $q(\boldsymbol{s}^v/\boldsymbol{s}^{v^*}) = (\psi(\boldsymbol{Q}^{(v)}))^\alpha$ and $q(\boldsymbol{z}^v/\boldsymbol{s}^v) = (\psi(\boldsymbol{M}^{(s^v)}))^\alpha$, where $\psi$ is a fully connected layer, and $\alpha$ is a balance factor to preserve crucial information and ensure model stability [17]. We set the value of $\alpha$ to 10 in our experiments.

## B  Detailed Proof for the Theoretical Results

In this section, we will provide a rigorous proof of the theoretical results mentioned in the main text.

### B.1  Proof of the Theorem 1

**Theorem 4.** *(**Discriminability of Label-specific Representation.**) For label prototypes $\boldsymbol{E}_j$ and $\boldsymbol{E}_k$ such that $k \neq j$ for all $k$, and view representations $\boldsymbol{X}^{(v)}$ and $\boldsymbol{X}^{(v^*)}$ such that $v^* \neq v$ for all $v^*$, the discriminability of $\hat{\boldsymbol{P}}^{(v)}$ for class $j$ necessitates that either of the following conditions be satisfied:*

$$
\begin{cases}
\mathbb{E}\left[\boldsymbol{X}^{(v)} \sigma_S(\boldsymbol{E}_j)^\top\right] = \frac{1}{N} \sum_{i=1}^{N} \boldsymbol{x}_i^{(v)} \sigma_S(\boldsymbol{E}_j)^\top > \mathbb{E}\left[\boldsymbol{X}^{(v)} \sigma_S(\boldsymbol{E}_k)^\top\right] = \frac{1}{N} \sum_{i=1}^{N} \boldsymbol{x}_i^{(v)} \sigma_S(\boldsymbol{E}_k)^\top \\[1.5em]
\mathbb{E}\left[\boldsymbol{X}^{(v)} \sigma_S(\boldsymbol{E}_j)^\top\right] = \frac{1}{N} \sum_{i=1}^{N} \boldsymbol{x}_i^{(v)} \sigma_S(\boldsymbol{E}_j)^\top > \mathbb{E}\left[\boldsymbol{X}^{(v^*)} \sigma_S(\boldsymbol{E}_j)^\top\right] = \frac{1}{N} \sum_{i=1}^{N} \boldsymbol{x}_i^{(v^*)} \sigma_S(\boldsymbol{E}_j)^\top.
\end{cases}
$$

*Proof.* Since our training data is multi-view and multi-label, the discriminative power of view $v$ with respect to label $j$ involves two key considerations. First, among all labels, view $v$ yields the most accurate prediction for label $j$; second, across all views, view $v$ provides the prediction for label $j$ with the highest confidence. Thus, we proceed with the proof from the following two aspects:

(i) For $\boldsymbol{E}_j$ and $\boldsymbol{E}_k$ such that $k \neq j$ for all $k$, the following inequality regrading view $v$ holds:

$$
\mathbb{E}\left[\boldsymbol{X}^{(v)} \sigma_S(\boldsymbol{E}_j)^\top\right] = \frac{1}{N} \sum_{i=1}^{N} \boldsymbol{x}_i^{(v)} \sigma_S(\boldsymbol{E}_j)^\top > \mathbb{E}\left[\boldsymbol{X}^{(v)} \sigma_S(\boldsymbol{E}_k)^\top\right] = \frac{1}{N} \sum_{i=1}^{N} \boldsymbol{x}_i^{(v)} \sigma_S(\boldsymbol{E}_k)^\top. \tag{31}
$$

The $j$-th component of $\hat{\boldsymbol{P}}_i^{(v)}$ (corresponding to label $j$) is

$$
\hat{\boldsymbol{P}}_{i,j}^{(v)} = \sigma_S(\boldsymbol{E}_j) \odot \boldsymbol{x}_i^{(v)},
$$

where $\hat{\boldsymbol{P}}_{i,j}^{(v)}$ is a vector where each element is the dot product of the corresponding elements of $\sigma_S(\boldsymbol{E}_j)$ and $\boldsymbol{x}_i^{(v)}$. Since a linear classifier (e.g., fully connected layers) is used for classification, the prediction score for sample $i$ is

$$
\boldsymbol{U}_i^{(v)} = \boldsymbol{W} \hat{\boldsymbol{P}}_i^{(v)} + \boldsymbol{b},
$$

where $\boldsymbol{W} \in \mathbb{R}^{C \times (C \times d)}$ is the weight matrix and $\boldsymbol{b} \in \mathbb{R}^C$ is the bias term. Give that $\hat{\boldsymbol{P}}_i^{(v)} \in \mathbb{R}^{C \times d}$, we can flatten it into $C$ individual vectors for each label. Besides, $\boldsymbol{W}$ has a block-diagonal structure, i.e., the weights for each label $j$ only act on $\hat{\boldsymbol{P}}_{i,j}^{(v)}$. Then, the prediction score for label $j$ simplifies to

$$U_{i,j}^{(v)} = \boldsymbol{w}_j^\top \hat{\boldsymbol{P}}_{i,j}^{(v)} + b_j = \boldsymbol{w}_j^\top \left( \sigma_S(\boldsymbol{E}_j) \odot \boldsymbol{x}_i^{(v)} \right) + b_j,$$

where $\boldsymbol{w}_j \in \mathbb{R}^d$ is the classifier weight for label $j$. To express discriminability, we need to show

$$\mathbb{E}\left[ U_{:,j}^{(v)} \right] > \mathbb{E}\left[ U_{:,k}^{(v)} \right], \quad \forall k \neq j.$$

The expectation is obtained by averaging over all empirical samples:

$$\mathbb{E}\left[ U_{:,j}^{(v)} \right] = \frac{1}{N} \sum_{i=1}^{N} \boldsymbol{w}_j^\top \left( \sigma_S(\boldsymbol{E}_j) \odot \boldsymbol{x}_i^{(v)} \right) + b_j,$$

$$\mathbb{E}\left[ U_{:,k}^{(v)} \right] = \frac{1}{N} \sum_{i=1}^{N} \boldsymbol{w}_k^\top \left( \sigma_S(\boldsymbol{E}_k) \odot \boldsymbol{x}_i^{(v)} \right) + b_k.$$

Assuming classifier weights $\boldsymbol{w}_k$ are independent of input data (e.g., optimized during training) and biases $b_k$ are constants, the core comparison reduces to

$$\sum_{i=1}^{N} \boldsymbol{w}_j^\top \left( \sigma_S(\boldsymbol{E}_j) \odot \boldsymbol{x}_i^{(v)} \right) \quad \text{vs.} \quad \sum_{i=1}^{N} \boldsymbol{w}_k^\top \left( \sigma_S(\boldsymbol{E}_k) \odot \boldsymbol{x}_i^{(v)} \right). \tag{32}$$

During training, the classifier weights $\boldsymbol{w}_j$ tend to align with label prototypes $\boldsymbol{E}_j$ since $\boldsymbol{E}_j$ captures the semantic meaning of label $j$. Thus, we approximate $\boldsymbol{w}_k \approx \boldsymbol{E}_k^\top$. Substituting it into Eq. (32), the comparison terms simplify to

$$\sum_{i=1}^{N} \boldsymbol{E}_j \left( \sigma_S(\boldsymbol{E}_j) \odot \boldsymbol{x}_i^{(v)} \right) \quad \text{vs.} \quad \sum_{i=1}^{N} \boldsymbol{E}_k \left( \sigma_S(\boldsymbol{E}_k) \odot \boldsymbol{x}_i^{(v)} \right).$$

Using properties of dot products:

$$\boldsymbol{E}_j \left( \sigma_S(\boldsymbol{E}_j) \odot \boldsymbol{x}_i^{(v)} \right) = \sum_{l=1}^{d} E_{j,l} \cdot \sigma_S(E_{j,l}) \cdot x_{i,l}^{(v)},$$

$$\boldsymbol{E}_k \left( \sigma_S(\boldsymbol{E}_k) \odot \boldsymbol{x}_i^{(v)} \right) = \sum_{l=1}^{d} E_{k,l} \cdot \sigma_S(E_{k,l}) \cdot x_{i,l}^{(v)}.$$

Then, the comparison is further refined into

$$\sum_{i=1}^{N} \sum_{l=1}^{d} E_{j,l} \cdot \sigma_S(E_{j,l}) \cdot x_{i,l}^{(v)} \quad \text{vs.} \quad \sum_{i=1}^{N} \sum_{l=1}^{d} E_{k,l} \cdot \sigma_S(E_{k,l}) \cdot x_{i,l}^{(v)}.$$

Note that $\sigma_S(E_{j,l}) \in (0,1)$ and is monotonically increasing. For positive samples of label $i$, $\boldsymbol{E}_j$ and $\boldsymbol{x}_i^{(v)}$ are better aligned, meaning $E_{j,l} \cdot x_{i,l}^{(v)}$ has a higher expectation. Thus, if the condition is

$$\mathbb{E}\left[ \boldsymbol{X}^{(v)} \sigma_S(\boldsymbol{E}_j)^\top \right] > \mathbb{E}\left[ \boldsymbol{X}^{(v)} \sigma_S(\boldsymbol{E}_k)^\top \right].$$

Then, the following inequality holds:

$$\sum_{i=1}^{N} \sum_{l=1}^{d} \sigma_S(E_{k,l}) \cdot x_{i,l}^{(v)} > \sum_{i=1}^{N} \sum_{l=1}^{d} \sigma_S(E_{j,l}) \cdot x_{i,l}^{(v)}.$$

Assuming $\boldsymbol{E}_j$ and $\boldsymbol{E}_k$ have similar scales ($\|\boldsymbol{E}_j\|_2^2 \approx \|\boldsymbol{E}_k\|_2^2$), we conclude

$$\sum_{i=1}^{N} \boldsymbol{E}_j \left( \sigma_S(\boldsymbol{E}_j) \odot \boldsymbol{x}_i^{(v)} \right) > \sum_{i=1}^{N} \boldsymbol{E}_k \left( \sigma_S(\boldsymbol{E}_k) \odot \boldsymbol{x}_i^{(v)} \right). \tag{33}$$

Furthermore, we have

$$\mathbb{E}\left[U_{:,j}^{(v)}\right] > \mathbb{E}\left[U_{:,k}^{(v)}\right], \quad \forall k \neq j.$$

Therefore, the prediction score for label $j$ exceeds that of all other labels, demonstrating the discriminative capability of view $v$ for class $j$.

(ii) For view $v$ and $v^*$ such that $v^* \neq v$ for all $v^*$, the following inequality regrading label $j$ holds:

$$\mathbb{E}\left[\boldsymbol{X}^{(v)}\sigma_S(\boldsymbol{E}_j)^\top\right] = \frac{1}{N}\sum_{i=1}^{N}\boldsymbol{x}_i^{(v)}\sigma_S(\boldsymbol{E}_j)^\top > \mathbb{E}\left[\boldsymbol{X}^{(v^*)}\sigma_S(\boldsymbol{E}_j)^\top\right] = \frac{1}{N}\sum_{i=1}^{N}\boldsymbol{x}_i^{(v^*)}\sigma_S(\boldsymbol{E}_j)^\top. \tag{34}$$

The prediction score for label $j$ assigned by view $v$ and $v^*$, respectively, is

$$U_{i,j}^{(v)} = \boldsymbol{w}_j^\top \hat{\boldsymbol{P}}_{i,j}^{(v)} + b_j = \boldsymbol{w}_j^\top\left(\sigma_S(\boldsymbol{E}_j) \odot \boldsymbol{x}_i^{(v)}\right) + b_j,$$

$$U_{i,j}^{(v^*)} = \boldsymbol{w}_j^\top \hat{\boldsymbol{P}}_{i,j}^{(v^*)} + b_j = \boldsymbol{w}_j^\top\left(\sigma_S(\boldsymbol{E}_j) \odot \boldsymbol{x}_i^{(v^*)}\right) + b_j,$$

To express discriminability, we need to show

$$\mathbb{E}\left[U_{:,j}^{(v)}\right] > \mathbb{E}\left[U_{:,j}^{(v^*)}\right], \quad \forall v \neq v^*.$$

Following the same rationale, we can transform the comparison into

$$\sum_{i=1}^{N}\sum_{l=1}^{d}E_{j,l} \cdot \sigma_S(E_{j,l}) \cdot x_{i,l}^{(v)} \quad \text{vs.} \quad \sum_{i=1}^{N}\sum_{l=1}^{d}E_{j,l} \cdot \sigma_S(E_{j,l}) \cdot x_{i,l}^{(v^*)}.$$

Thus, if the condition is

$$\mathbb{E}\left[\boldsymbol{X}^{(v)}\sigma_S(\boldsymbol{E}_j)^\top\right] > \mathbb{E}\left[\boldsymbol{X}^{(v^*)}\sigma_S(\boldsymbol{E}_j)^\top\right].$$

Then, the following inequality holds:

$$\sum_{i=1}^{N}\sum_{l=1}^{d}\sigma_S(E_{j,l}) \cdot x_{i,l}^{(v)} > \sum_{i=1}^{N}\sum_{l=1}^{d}\sigma_S(E_{j,l}) \cdot x_{i,l}^{(v^*)}.$$

Similarly, we conclude

$$\sum_{i=1}^{N}\boldsymbol{E}_j\left(\sigma_S(\boldsymbol{E}_j) \odot \boldsymbol{x}_i^{(v)}\right) > \sum_{i=1}^{N}\boldsymbol{E}_j\left(\sigma_S(\boldsymbol{E}_j) \odot \boldsymbol{x}_i^{(v^*)}\right). \tag{35}$$

Furthermore, we have

$$\mathbb{E}\left[U_{:,j}^{(v)}\right] > \mathbb{E}\left[U_{:,k}^{(v^*)}\right], \quad \forall v \neq v^*.$$

Therefore, the prediction score of the $v$-th view for class $j$ is higher than that of all other views, indicating that view $v$ exhibits discriminative power for label $j$. Combining (i) and (ii), we complete the proof. □

## B.2 Proof of the Theorem 2

**Theorem 5.** *(**Effectiveness of Disentangled Representation.**) Let the disentangled representation be denoted as $\boldsymbol{R} = (\boldsymbol{S}^{(1)}, \ldots, \boldsymbol{S}^{(V)}, \boldsymbol{O}^{(1)}, \ldots, \boldsymbol{O}^{(V)})$, where the information entropy of each representation is assumed to be fixed, i.e., $H(\boldsymbol{S}^{(v)}) = H(\boldsymbol{O}^{(v)}) = H^0$ $(1 \leq v \leq V)$. Then, in the case where each shared and specific representation is indispensable for prediction, $I(\boldsymbol{R}; \boldsymbol{Y})$ will attain its maximum when $\boldsymbol{R} = \boldsymbol{R}_*$, with $\boldsymbol{R}_*$ being the optimal solution of the problem (16).*

*Proof.* By integrating core features from the raw data to construct shared representations and leveraging classification loss to derive view-specific representations aligned with particular labels, our approach enables the shared features to encapsulate primary generalization information across multiple views, while the specific features are designed to capture discriminative information that is

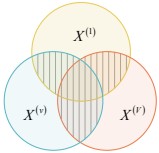
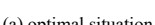
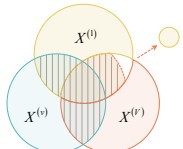
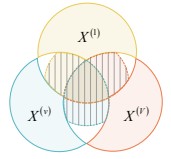
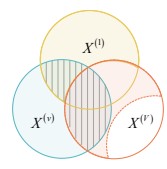

(a) optimal situation    (b) lack of information shift    (c) lack of shared information interaction    (d) lack of specific information interaction    (e) lack of information orthogonality

Figure 5: Feature extraction visualization. Fig. (a) illustrates the optimal solution obtained by our model, while the remaining four figures display cases where the constraints of information shift, interaction, and orthogonality are successively removed. The shaded areas represent the shared representations, whereas the other colored regions indicate the view-specific representations.

finely attuned to task-specific characteristics. Thus, the assumption that each shared and specific representation is indispensable for prediction aligns well with our model. Besides, as all representations are extracted into the same dimensional space, it is reasonable to posit that they encode equivalent quantities of information, i.e., $H(\boldsymbol{S}^{(v)}) = H(\boldsymbol{O}^{(v)}) = H^0$ $(1 \leq v \leq V)$. Let define the optimal solution of the problem (16) as $\boldsymbol{R}_* = (\boldsymbol{S}_*^{(1)}, \ldots, \boldsymbol{S}_*^{(V)}, \boldsymbol{O}_*^{(1)}, \ldots, \boldsymbol{O}_*^{(V)})$. Then, we specify that identifying each class $\boldsymbol{Y}^c$ requires a corresponding representation setting $\boldsymbol{R}^{\boldsymbol{Y}^c}$ to be mined, i.e, $H(\boldsymbol{Y}^c) = H(\boldsymbol{R}^{\boldsymbol{Y}^c})$, which indicates that the successful extraction of $\boldsymbol{R}^{\boldsymbol{Y}^c}$ is sufficient for predicting $\boldsymbol{Y}^c$. Based on this, we establish the following formulation:

$$\begin{cases} H(\boldsymbol{R}^{\boldsymbol{Y}^c}) = H(\boldsymbol{S}^{\boldsymbol{Y}^c}) + H(\boldsymbol{O}^{\boldsymbol{Y}^c}), \ H(\boldsymbol{S}^{\boldsymbol{Y}^c}) = \sum_{v=1}^{V} f_c^v \cdot H(\boldsymbol{S}_*^{(v)}), \ H(\boldsymbol{O}^{\boldsymbol{Y}^c}) = \sum_{v=1}^{V} g_c^v \cdot H(\boldsymbol{O}_*^{(v)}) \\ \sum_{c=1}^{C} \sum_{v=1}^{V} f_c^v \cdot H(\boldsymbol{S}_*^{(v)}) + \sum_{c=1}^{C} \sum_{v=1}^{V} g_c^v \cdot H(\boldsymbol{O}_*^{(v)}) = H(\boldsymbol{R}_*). \end{cases}$$
(36)

where $f_c^v$ and $g_c^v$ are indicator functions representing the shared and specific information components that constitute $\boldsymbol{R}^{\boldsymbol{Y}^c}$. According to Eq. (36), all information contained in the extracted representations is essential for predicting the target labels. Next, we demonstrate from three perspectives that our feature extraction model is capable of maximizing the mutual information between the representations and the labels, i.e., the representations contain insufficient information for prediction, just enough information, or redundant information beyond what is necessary for prediction.

(i) $(\boldsymbol{S}^{(1)}, \ldots, \boldsymbol{S}^{(V)}, \boldsymbol{O}^{(1)}, \ldots, \boldsymbol{O}^{(V)}) \subset \boldsymbol{R}^{\boldsymbol{Y}^1} \bigcup \boldsymbol{R}^{\boldsymbol{Y}^2} \cdots \bigcup \boldsymbol{R}^{\boldsymbol{Y}^C}$.

In this case, we denote the mutual information between $\boldsymbol{R}_*$ and $\boldsymbol{Y}$ as $I^0$, i.e.,

$$\begin{aligned} I(\boldsymbol{R}_*; \boldsymbol{Y}) &= \sum_{c=1}^{C} I((\boldsymbol{S}_*^{(1)}, \ldots, \boldsymbol{S}_*^{(V)}, \boldsymbol{O}_*^{(1)}, \ldots, \boldsymbol{O}_*^{(V)}); \boldsymbol{Y}^c) \\ &= \sum_{c \in \mathcal{L}} I(\boldsymbol{R}^{\boldsymbol{Y}^c}; \boldsymbol{Y}^c) + \sum_{c \in \mathcal{U}} I((\boldsymbol{S}_*^{(1)}, \ldots, \boldsymbol{S}_*^{(V)}, \boldsymbol{O}_*^{(1)}, \ldots, \boldsymbol{O}_*^{(V)}); \boldsymbol{Y}^c) \\ &= \sum_{c \in \mathcal{L}} H(\boldsymbol{Y}^c) - H(\boldsymbol{Y}^c|\boldsymbol{R}^{\boldsymbol{Y}^c}) + \sum_{c \in \mathcal{U}} H((\boldsymbol{S}_*^{(1)}, \ldots, \boldsymbol{S}_*^{(V)}, \boldsymbol{O}_*^{(1)}, \ldots, \boldsymbol{O}_*^{(V)})) \\ &\quad - H((\boldsymbol{S}_*^{(1)}, \ldots, \boldsymbol{S}_*^{(V)}, \boldsymbol{O}_*^{(1)}, \ldots, \boldsymbol{O}_*^{(V)})|\boldsymbol{Y}^c) \\ &= \sum_{c \in \mathcal{L}} H(\boldsymbol{Y}^c) + \sum_{c \in \mathcal{U}} H((\boldsymbol{S}_*^{(1)}, \ldots, \boldsymbol{S}_*^{(V)}, \boldsymbol{O}_*^{(1)}, \ldots, \boldsymbol{O}_*^{(V)})) \\ &= \sum_{c \in \mathcal{L}} H(\boldsymbol{Y}^c) + 2V|\mathcal{U}|H^0 - (V-1)|\mathcal{U}|I(\boldsymbol{S}_*^{(1)}; \ldots; \boldsymbol{S}_*^{(V)}) = I^0. \end{aligned}$$

where $\mathcal{L}$ and $\mathcal{U}$ denote the subsets of labels that are fully predictable and not adequately predictable, respectively, and $|\mathcal{L}| + |\mathcal{U}| = C$. When the core shared feature information is essential to all labels, reducing $I(\boldsymbol{S}_*^{(1)}; \ldots; \boldsymbol{S}_*^{(V)})$ to enhance label-specific information associated with certain labels will not lead to an increase in $I(\boldsymbol{R}_*; \boldsymbol{Y})$. Since $\sum_{c \in \mathcal{L}} H(\boldsymbol{Y}^c)$ is fixed, postulating the existence of an alternative representation $\boldsymbol{R}$ satisfying $I(\boldsymbol{R}; \boldsymbol{Y}) > I(\boldsymbol{R}_*; \boldsymbol{Y})$ would inevitably contradict the

assumption that the information entropy of each representation is fixed at $H^0$. Thus, we can obtain $I(\boldsymbol{R}_*; \boldsymbol{Y}) \geq I(\boldsymbol{R}; \boldsymbol{Y})$. Given the insufficiency of prediction information and a fixed amount of information encoded in the representations, enhancing performance necessitates incorporating usable content beyond raw information and introducing additional constraints offers no further gain in the mutual information between representations and labels.

(ii) $(\boldsymbol{S}^{(1)}, \ldots, \boldsymbol{S}^{(V)}, \boldsymbol{O}^{(1)}, \ldots, \boldsymbol{O}^{(V)}) = \boldsymbol{R}^{\boldsymbol{Y}^1} \bigcup \boldsymbol{R}^{\boldsymbol{Y}^2} \cdots \bigcup \boldsymbol{R}^{\boldsymbol{Y}^C}$.

In this case, similarly, we have the following equality:

$$
\begin{aligned}
I(\boldsymbol{R}_*; \boldsymbol{Y}) &= \sum_{c=1}^{C} I((\boldsymbol{S}_*^{(1)}, \ldots, \boldsymbol{S}_*^{(V)}, \boldsymbol{O}_*^{(1)}, \ldots, \boldsymbol{O}_*^{(V)}); \boldsymbol{Y}^c) \\
&= \sum_{c=1}^{C} I(\boldsymbol{R}^{\boldsymbol{Y}^c}; \boldsymbol{Y}^c) \\
&= \sum_{c=1}^{C} H(\boldsymbol{R}^{\boldsymbol{Y}^c}) - H(\boldsymbol{R}^{\boldsymbol{Y}^c} | \boldsymbol{Y}^c) \\
&= \sum_{c=1}^{C} \sum_{v=1}^{V} f_c^v \cdot H(\boldsymbol{S}_*^{(v)}) + \sum_{c=1}^{C} \sum_{v=1}^{V} g_c^v \cdot H(\boldsymbol{O}_*^{(v)}).
\end{aligned}
$$

If the information shift term is not sufficiently optimized as shown in Fig. 5.(b), there will exist a shared representation $\tilde{\boldsymbol{S}}^{(m)}$ in the setting $\boldsymbol{R}$ such that

$$
\begin{aligned}
I(\boldsymbol{R}; \boldsymbol{Y}) = &\sum_{c \in T_m^n} \sum_{v=1}^{V} \tilde{f}_c^v \cdot H(\boldsymbol{S}^{(v)}) + \sum_{c \in T_m} \sum_{v=1, \neq m}^{V} \tilde{f}_c^v \cdot H(\boldsymbol{S}^{(v)}) \\
&+ \sum_{c \in T_m} \tilde{f}_c^m \cdot H(\tilde{\boldsymbol{S}}^{(m)}) + \sum_{c=1}^{C} \sum_{v=1}^{V} g_c^v \cdot H(\boldsymbol{O}_*^{(v)}).
\end{aligned}
$$

where $T_m$ denotes the set of labels whose prediction processes involve contributions from $\tilde{\boldsymbol{S}}^{(m)}$, while $T_m^n$ is the set unaffected by $\tilde{\boldsymbol{S}}^{(m)}$. Since the change in information entropy occurs exclusively in $\tilde{\boldsymbol{S}}^{(m)}$, we have

$$
\begin{cases}
\sum_{c \in T_m^n} \sum_{v=1}^{V} \tilde{f}_c^v \cdot H(\boldsymbol{S}^{(v)}) = \sum_{c \in T_m^n} \sum_{v=1}^{V} f_c^v \cdot H(\boldsymbol{S}_*^{(v)}) \\
\sum_{c \in T_m} \sum_{v=1, \neq m}^{V} \tilde{f}_c^v \cdot H(\boldsymbol{S}^{(v)}) = \sum_{c \in T_m} \sum_{v=1, \neq m}^{V} f_c^v \cdot H(\boldsymbol{S}_*^{(v)}) \\
\sum_{c \in T_m} \tilde{f}_c^m \cdot H(\tilde{\boldsymbol{S}}^{(m)}) \leq \sum_{c \in T_m} f_c^m \cdot H(\boldsymbol{S}_*^{(m)}).
\end{cases}
$$

Furthermore, we have the following comparison:

$$
\begin{aligned}
I(\boldsymbol{R}; \boldsymbol{Y}) \leq &\sum_{c \in T_m^n} \sum_{v=1}^{V} f_c^v \cdot H(\boldsymbol{S}_*^{(v)}) + \sum_{c \in T_m} \sum_{v=1, \neq m}^{V} f_c^v \cdot H(\boldsymbol{S}_*^{(v)}) \\
&+ \sum_{c \in T_m} f_c^m \cdot H(\boldsymbol{S}_*^{(m)}) + \sum_{c=1}^{C} \sum_{v=1}^{V} g_c^v \cdot H(\boldsymbol{O}_*^{(v)}) \\
&= I(\boldsymbol{R}_*; \boldsymbol{Y}).
\end{aligned}
$$

If the shared information interaction term is not sufficiently optimized (Fig. 5.(c)), there will exist a set of shared representations whose task-relevant information is significantly reduced due to ineffective interactions. Under such circumstances, it similarly follows that $I(\boldsymbol{R}; \boldsymbol{Y}) \leq I(\boldsymbol{R}_*; \boldsymbol{Y})$. Moreover, insufficient suppression of interactions among specific representations causes overlap, thereby reducing the useful information entropy of each representation (Fig. 5.(d)). Additionally, inadequate optimization of the orthogonality constraint can introduce redundancy between shared and specific representations, which in turn hinders the extraction of the specific representations (Fig. 5.(e)). For this two aspects, we also have $I(\boldsymbol{R}; \boldsymbol{Y}) \leq I(\boldsymbol{R}_*; \boldsymbol{Y})$.

(iii) $\boldsymbol{R}^{\boldsymbol{Y}^1} \bigcup \boldsymbol{R}^{\boldsymbol{Y}^2} \cdots \bigcup \boldsymbol{R}^{\boldsymbol{Y}^C} \subset (\boldsymbol{S}^{(1)}, \ldots, \boldsymbol{S}^{(V)}, \boldsymbol{O}^{(1)}, \ldots, \boldsymbol{O}^{(V)})$.

In this case, we can obtain

$$I(\boldsymbol{R}_*; \boldsymbol{Y}) = \sum_{c=1}^{C} I((\boldsymbol{S}_*^{(1)}, \ldots, \boldsymbol{S}_*^{(V)}, \boldsymbol{O}_*^{(1)}, \ldots, \boldsymbol{O}_*^{(V)}); \boldsymbol{Y}^c)$$

$$= \sum_{c=1}^{C} I(\boldsymbol{R}^{\boldsymbol{Y}^c}; \boldsymbol{Y}^c)$$

$$= \sum_{c=1}^{C} H(\boldsymbol{Y}^c) - H(\boldsymbol{Y}^c | \boldsymbol{R}^{\boldsymbol{Y}^c})$$

$$= \sum_{c=1}^{C} H(\boldsymbol{Y}^c).$$

Given that $\sum_{c=1}^{C} H(\boldsymbol{Y}^c)$ quantifies the total label information, the mutual information $I(\boldsymbol{R}; \boldsymbol{Y})$ obtained from any alternative representation is inherently constrained by this upper bound. Therefore, we conclude that $I(\boldsymbol{R}; \boldsymbol{Y}) \leq I(\boldsymbol{R}_*; \boldsymbol{Y})$.

Combining (i) (ii) and (iii), we deduce that the maximization of $I(\boldsymbol{R}; \boldsymbol{Y})$ is attained when $\boldsymbol{R}$ serves as the optimal solution $\boldsymbol{R}_*$ of the problem (16). Moreover, the principles of information shift, interaction, and orthogonality are all indispensable for achieving this. □

### B.3 Proof of the Theorem 3

**Theorem 6.** *(**Generalization Error Bound.**) Our model is designed to learn a vector-valued function $f = (f_1, \ldots, f_C) : \mathcal{X} \mapsto \mathbb{R}^C$. The expected risk and empirical risk w.r.t. the training dataset $D$ are denoted as $R(f) = \mathbb{E}_{(\boldsymbol{X}, \boldsymbol{Y}) \sim \mathcal{X} \times \mathcal{Y}}[\ell(f^{av}(\boldsymbol{X}, \boldsymbol{Q}), \boldsymbol{Y})]$ and $\widehat{R}_D(f) = \frac{1}{NC} \sum_{i=1}^{N} \sum_{c=1}^{C} \ell(\sum_{v=1}^{V} (\boldsymbol{Q}_{i,v} f_c(\boldsymbol{x}_i^{(v)})), \boldsymbol{Y}_{i,c})$, where $f^{av}(\cdot)$ refers to the late fusion of multiple views. With probability at least $1 - \delta$, we have the following generalization error bound:*

$$R(f) - \widehat{R}_D(f) \leq \frac{\widetilde{\mathcal{K}}_1}{\sqrt{NV}} + \frac{\widetilde{\mathcal{K}}_2}{N^{3/4}V^{1/4}} + \overline{\mathrm{gen}}_{rec}(\boldsymbol{Q}, \boldsymbol{X}, \boldsymbol{Y})$$

$$+ \widetilde{\mathcal{K}}_3 \sqrt{\frac{\sum_{c=1}^{C} \left( \sum_{v=1}^{V} I(\boldsymbol{X}^{(v)}; \boldsymbol{S}^{(v)}, \boldsymbol{O}^{(v)} | \boldsymbol{Y}^c) + \widetilde{\mathcal{K}}_4 \right)}{NC}},$$

*where $\widetilde{\mathcal{K}}_1 = \widetilde{\mathcal{K}}_3 = \mathcal{O}(C)$, $\widetilde{\mathcal{K}}_2 = \mathcal{O}(\sqrt{C})$, $\widetilde{\mathcal{K}}_4$ is constant of order $\widetilde{\mathcal{O}}(1)$ as $N, V \to \infty$, and $\overline{\mathrm{gen}}_{rec}(\boldsymbol{Q}, \boldsymbol{X}, \boldsymbol{Y})$ is the generalization error related to the view reconstruction quality. Moreover, the generalization error bound becomes increasingly tighter during the optimization of the problem (16).*

*Proof.* The training dataset denoted as $D = \{(\boldsymbol{x}_i, \boldsymbol{y}_i) : i \in [N]\}$ is drawn from a probability distribution over $\mathcal{X} \times \mathcal{Y}$. Each $\boldsymbol{x}_i = (\boldsymbol{x}_i^{(1)}, \ldots, \boldsymbol{x}_i^{(V)})$ consists of $V$ views and $\boldsymbol{y}_i = (Y_{i,1}, \ldots, Y_{i,C})$. Our strategy is to learn a vector-valued function $\boldsymbol{f} = (f_1, \ldots, f_C) : \mathcal{X} \mapsto \mathbb{R}^C$ and determine relevant labels by applying a thresholding criterion. The goal of learning is to find a hypothesis $f \in \mathcal{F}$ with good generalization performance by minimizing the loss $\ell$ on the dataset $D$. As missing labels inevitably degrade generalization, we focus on deriving the tightest generalization error bound under complete labeling. Thus, the expected risk and empirical risk w.r.t. the training dataset $D$ are denoted as $R(f) = \mathbb{E}_{(\boldsymbol{X}, \boldsymbol{Y}) \sim \mathcal{X} \times \mathcal{Y}}[\ell(f^{av}(\boldsymbol{X}, \boldsymbol{Q}), \boldsymbol{Y})]$ and $\widehat{R}_D(f) = \frac{1}{NC} \sum_{i=1}^{N} \sum_{c=1}^{C} \ell(\sum_{v=1}^{V} (\boldsymbol{Q}_{i,v} f_c(\boldsymbol{x}_i^{(v)})), \boldsymbol{Y}_{i,c})$, respectively, where $f^{av}(\cdot)$ refers to the result fusion of multiple views, $\boldsymbol{Q}$ is the reconstruction quality score serving as the fusion weights. We define the function class of TDLSR as follows:

$$\mathcal{F} = \{\boldsymbol{X}^{(v)} \mapsto \boldsymbol{f}(\boldsymbol{X}^{(v)}) : f(\boldsymbol{X}^{(v)}) = (f_1(\boldsymbol{X}^{(v)}), \ldots, f_C(\boldsymbol{X}^{(v)})), f_c(\boldsymbol{X}^{(v)}) = \psi_c \phi^{(v)}(\boldsymbol{X}^{(v)}),$$
$$\phi^{(v)} = (\phi_v^s, \phi_v^o), \psi_c = (\boldsymbol{w}_c^s \zeta_c, \boldsymbol{w}_c^o \zeta_c), \boldsymbol{w} \in \mathbb{R}^{1 \times d}\},$$

(37)

where $\phi^s$ and $\phi^o$ denote the view-common and view-specific representation extractors, $\zeta_j$ is a nonlinear mapping induced by the label-specific representation construction and $\boldsymbol{w}$ is the classification head. Besides, the disentangled representation $\boldsymbol{R}$ is expressed by $\boldsymbol{R} = \phi(\boldsymbol{X}) = (\phi^s(\boldsymbol{X}), \phi^o(\boldsymbol{X})) =$

$(\boldsymbol{S}^{(1)}, \ldots, \boldsymbol{S}^{(V)}, \boldsymbol{O}^{(1)}, \ldots, \boldsymbol{O}^{(V)})$. Let multi-view data $\boldsymbol{X} = \{\boldsymbol{X}^{(v)}\}_{v=1}^V$ be produced with a hidden label-specific function $\theta^c$ by $\boldsymbol{X} = \theta^c(\boldsymbol{Y}^c, \boldsymbol{V})$, where $\boldsymbol{Y}^c$ is the randomly generated single label and $\boldsymbol{V} = \{\boldsymbol{V}^{(v)} = (\boldsymbol{V}_1^{(v)}, \ldots, \boldsymbol{V}_d^{(v)})\}_{v=1}^V \in \mathbb{R}^{m \times d}$ are nuisance variables. Denote the conditional random variables of $\boldsymbol{X}$ and $\boldsymbol{R}$ given the category $\boldsymbol{Y}^c = \boldsymbol{y}^c$ as $\boldsymbol{X}_{\boldsymbol{y}^c}$ and $\boldsymbol{R}_{\boldsymbol{y}^c}$, respectively. For any $\boldsymbol{y}^c \in \mathcal{Y}^* = \{-1, 1\}$, the sensitivity $c_\phi^{\boldsymbol{y}_c}$ of the representation function $\phi = \{\phi^{(v)}\}_{v=1}^V$ w.r.t the nuisance variable $\boldsymbol{V}_i^{(v)}$ is defined as

$$
\begin{aligned}
c_\phi^{\boldsymbol{y}^c} = \sup_{v \in [V]} \sup_{\boldsymbol{v}_1^{(v)}, \ldots, \hat{\boldsymbol{v}}_i^{(v)}, \ldots, \boldsymbol{v}_d^{(v)}} &| \log(p_{\boldsymbol{r}}(\phi^{(v)} \circ \theta_{\boldsymbol{y}_c}(\boldsymbol{v}_1^{(v)}, \ldots, \boldsymbol{v}_i^{(v)}, \ldots, \boldsymbol{v}_d^{(v)}))) \\
&- \log(p_{\boldsymbol{r}}(\phi^{(v)} \circ \theta_{\boldsymbol{y}_c}\left(\boldsymbol{v}_1^{(v)}, \ldots, \hat{\boldsymbol{v}}_i^{(v)}, \ldots, \boldsymbol{v}_d^{(v)}\right)))|,
\end{aligned}
\tag{38}
$$

where $\theta_{\boldsymbol{y}^c}(\boldsymbol{v}^{(v)}) = \theta(\boldsymbol{y}^c, \boldsymbol{v}^{(v)})$ and $p_{\boldsymbol{r}}(\boldsymbol{r}) = \mathbb{P}(\boldsymbol{R} = \boldsymbol{r})$. Based on Eq. (38), we set the global sensitivity of $\phi$ as $c_\phi = \sup_{c \in [C]} c_\phi^{\boldsymbol{y}^c}$. Let $\mathcal{R}_c^x = \{\theta_{\boldsymbol{y}^c}(\boldsymbol{v}), \phi \circ \theta_{\boldsymbol{y}^c}(\boldsymbol{v}) : \boldsymbol{v} \in \mathcal{V}, \boldsymbol{y}^c \in \mathcal{Y}^*\}$ denote the complete set of multi-view data and their corresponding representations. For any $\gamma > 0$, we construct the following typical representation subset for each class [39]:

$$
\mathcal{R}_{c,\gamma}^x = \left\{ \boldsymbol{X}, \boldsymbol{R} \in \mathcal{R}_c^x : -\log p_{\boldsymbol{r}|\boldsymbol{y}^c}(\boldsymbol{r}) - H(\boldsymbol{R}_{\boldsymbol{Y}^c}) \leq c_\phi \sqrt{\frac{d \log(\sqrt{NV}/\gamma)}{2}} \right\}.
\tag{39}
$$

Define the function $h_{\boldsymbol{y}^c}'(\boldsymbol{v}) = -\log p_{\boldsymbol{r}|\boldsymbol{y}^c}(h_{\boldsymbol{y}^c}''(\boldsymbol{v}))$, where $h_{\boldsymbol{y}^c}''(\boldsymbol{v}) = \phi \circ (\theta_{\boldsymbol{y}^c}(\boldsymbol{v}))$. Let $p_{\boldsymbol{v}}(\boldsymbol{v}) = P(\boldsymbol{V} = \boldsymbol{v})$ and $h_{\boldsymbol{y}^c}^{-1}(\boldsymbol{r}) = \{\boldsymbol{v} \in \mathcal{V} : h_{\boldsymbol{y}^c}(\boldsymbol{v}) = \boldsymbol{r}\}$, we have

$$
\begin{aligned}
\mathbb{E}_{\boldsymbol{V}}\left[h_{\boldsymbol{y}^c}'(\boldsymbol{V})\right] &= -\sum_{\boldsymbol{v} \in \mathcal{V}} p_{\boldsymbol{v}}(\boldsymbol{v}) \log p_{\boldsymbol{r}|\boldsymbol{y}^c}\left(h_{\boldsymbol{y}^c}''(\boldsymbol{v})\right) \\
&= -\sum_{\boldsymbol{r} \in \mathcal{R}_c^x} \sum_{\boldsymbol{v} \in h_{\boldsymbol{y}^c}^{-1}(\boldsymbol{r})} p_{\boldsymbol{v}}(\boldsymbol{v}) \log p_{\boldsymbol{r}|\boldsymbol{y}^c}\left(h_{\boldsymbol{y}^c}''(\boldsymbol{v})\right) \\
&= -\sum_{\boldsymbol{r} \in \mathcal{R}_c^x} \left(\sum_{\boldsymbol{v} \in h_{\boldsymbol{y}^c}^{-1}(\boldsymbol{r})} p_{\boldsymbol{v}}(\boldsymbol{v})\right) \log p_{\boldsymbol{r}|\boldsymbol{y}^c}(\boldsymbol{r}) \\
&= -\sum_{\boldsymbol{r} \in \mathcal{R}_c^x} p_{\boldsymbol{r}|\boldsymbol{y}^c}(\boldsymbol{r}) \log p_{\boldsymbol{r}|\boldsymbol{y}^c}(\boldsymbol{r}) \\
&= H\left(R_{\boldsymbol{y}^c}\right).
\end{aligned}
$$

By applying McDiarmid's inequality on $h_{\boldsymbol{y}^c}'(\boldsymbol{V})$, we get

$$
\mathbb{P}\left(-\log p_{\boldsymbol{r}|\boldsymbol{y}^c}(\boldsymbol{r}) - H(\boldsymbol{R}_{\boldsymbol{Y}^c}) \geq \epsilon\right) \leq \exp\left(-\frac{2\epsilon^2}{d(c_\phi)^2}\right).
$$

Taking $\delta = \exp\left(-\frac{2\epsilon^2}{d(c_\phi)^2}\right)$, we have

$$
\epsilon = c_\phi \sqrt{\frac{d \log(1/\delta)}{2}}.
$$

After setting $\delta = \gamma/\sqrt{NV}$, i.e., $\mathbb{P}\left(\boldsymbol{X}, \boldsymbol{R} \notin \mathcal{R}_{c,\gamma}^x\right) \leq \delta = \frac{\gamma}{\sqrt{NV}}$, we can obtain

$$
\exp(-H(\boldsymbol{R}_{\boldsymbol{Y}^c}) - \epsilon) \leq \log p_{\boldsymbol{r}|\boldsymbol{y}^c}(\boldsymbol{r}).
$$

Then, we further derive the following transformation:

$$
\left|\mathcal{R}_{c,\gamma}^x\right| \exp(-H(\boldsymbol{R}_{\boldsymbol{Y}^c}) - \epsilon) = \sum_{\boldsymbol{r} \in \mathcal{R}_{c,\gamma}^x} \exp(-H(\boldsymbol{R}_{\boldsymbol{Y}^c}) - \epsilon) \leq \sum_{\boldsymbol{r} \in \mathcal{R}_{c,\gamma}^x} p_{\boldsymbol{r}}(\boldsymbol{r}) = 1,
$$

which implies

$$
\left|\mathcal{R}_{c,\gamma}^x\right| \leq \exp\left(H(\boldsymbol{R}_{\boldsymbol{Y}^c}) + c_\phi \sqrt{\frac{d \log(\sqrt{NV}/\gamma)}{2}}\right).
$$

Therefore, regarding the property of the typical subset, the following lemma can be derived:

**Lemma 1.** *For any $\gamma > 0$ and all $v \in [V]$, we have*

$$\mathbb{P}(\boldsymbol{X}, \boldsymbol{R} \notin \mathcal{R}_{c,\gamma}^x) \leq \frac{\gamma}{\sqrt{NV}}, \quad |\mathcal{R}_{c,\gamma}^x| \leq \exp\left(H(\boldsymbol{R}_{\boldsymbol{Y}^c}) + c_\phi \sqrt{\frac{d\log(\sqrt{NV}/\gamma)}{2}}\right).$$

Besides, we need the following lemma for subsequent proof:

**Lemma 2.** *[18] The vector $X = (X_1, \ldots, X_k)$ is defined to follow the multinomial distribution with parameters $p = (p_1, \ldots, p_k)$. Let $\bar{a}_1, \ldots, \bar{a}_k \geq 0$ be fixed such that $\sum_{i=1}^k \bar{a}_i p_i \neq 0$. Then, for any $\epsilon > 0$, the following inequality holds:*

$$\mathbb{P}\left(\sum_{i=1}^k \bar{a}_i\left(p_i - \frac{X_i}{m}\right) > \epsilon\right) \leq \exp\left(-\frac{m\epsilon^2}{\beta}\right),$$

*where $\beta = 2\sum_{i=1}^k \bar{a}_i^2 p_i$.*

We further define $T^{\boldsymbol{y}^c} = |\mathcal{R}_{c,\gamma}^x|$, where the typical subset $\mathcal{R}_{c,\gamma}^x$ consists of elements $\mathcal{R}_{c,\gamma}^x = \{(a_{c,1}^x, a_{c,1}^s, a_{c,1}^o), \ldots, (a_{c,T}^x, a_{c,T}^s, a_{c,T}^o)\}$. Besides, we introduce the following label-specific sets:

$$\begin{aligned}
\mathcal{U}^{\boldsymbol{y}^c} &= \left\{i \in [N], v \in [V] : \boldsymbol{x}_i^{(v)}, \phi^{(v)}(\boldsymbol{x}_i^{(v)}) \notin \mathcal{R}_{c,\gamma}^x, \boldsymbol{Y}_{i,c} = \boldsymbol{y}^c\right\}, \\
\mathcal{U}_k^{\boldsymbol{y}^c} &= \left\{i \in [N], v \in [V] : \boldsymbol{x}_i^{(v)} = a_k^{c,x}, \phi^{(v)}(\boldsymbol{x}_i^{(v)}) = (a_k^{c,s}, a_k^{c,o}), \boldsymbol{Y}_{i,c} = \boldsymbol{y}^c\right\}.
\end{aligned} \quad (40)$$

Then, we conduct an analysis of the classification generalization error:

$$R(f) - \widehat{R}_D(f) = \mathbb{E}_{(\boldsymbol{X},\boldsymbol{Y})\sim\mathcal{X}\times\mathcal{Y}}[\ell(f^{av}(\boldsymbol{X},\boldsymbol{Q}),\boldsymbol{Y})] - \frac{1}{NC}\sum_{i=1}^N\sum_{c=1}^C \ell\left(\sum_{v=1}^V(\boldsymbol{Q}_{i,v}f_c(\boldsymbol{x}_i^{(v)})),\boldsymbol{Y}_{i,c}\right). \quad (41)$$

Since $\ell$ is a convex function, we can apply Jensen's inequality to obtain

$$\ell(\sum_{v=1}^V(\boldsymbol{Q}_{i,v}f_c(\boldsymbol{x}_i^{(v)})),\boldsymbol{Y}_{i,c}) \leq \boldsymbol{Q}_{i,v}\sum_{v=1}^V \ell(f_c(\boldsymbol{x}_i^{(v)}),\boldsymbol{Y}_{i,c}). \quad (42)$$

Thus, we define a instance-level loss difference term $\Delta\ell_i^c(\boldsymbol{Q}_{i,:}, \boldsymbol{X}_{i,:}, \boldsymbol{Y}_{i,c})$ such that

$$\Delta\ell_i^c = \boldsymbol{Q}_{i,v}\sum_{v=1}^V \ell(f_c(\boldsymbol{x}_i^{(v)}),\boldsymbol{Y}_{i,c}) - \ell(\sum_{v=1}^V(\boldsymbol{Q}_{i,v}f_c(\boldsymbol{x}_i^{(v)})),\boldsymbol{Y}_{i,c}). \quad (43)$$

Define $q = \sup_{i\in[N]}\sup_{v\in[V]} Q_{i,v}$ as the maximum view reconstruction quality among all views. Regarding the cumulative loss calculation for individual views, we have

$$\begin{aligned}
&\frac{1}{NC}\sum_{i=1}^N\sum_{c=1}^C\left(\boldsymbol{Q}_{i,v}\sum_{v=1}^V \ell(f_c(\boldsymbol{x}_i^{(v)}),\boldsymbol{Y}_{i,c})\right) \\
&= \frac{\boldsymbol{Q}_{i,v}}{NC}\sum_{c=1}^C\sum_{\boldsymbol{y}^c\in\mathcal{Y}^*}\left(\sum_{i,v\in\mathcal{U}^{\boldsymbol{y}^c}}\ell(\psi_c\circ\phi^{(v)}(\boldsymbol{x}_i^{(v)}),\boldsymbol{y}^c) + \sum_{k=1}^{T^{\boldsymbol{y}^c}}\sum_{i,v\in\mathcal{U}_k^{\boldsymbol{y}^c}}\ell(\psi_c\circ\phi^{(v)}(\boldsymbol{x}_i^{(v)}),\boldsymbol{y}^c)\right) \\
&= \frac{\boldsymbol{Q}_{i,v}}{NC}\sum_{c=1}^C\sum_{\boldsymbol{y}^c\in\mathcal{Y}^*}\sum_{i,v\in\mathcal{U}^{\boldsymbol{y}^c}}\ell(\psi_c\circ\phi^{(v)}(\boldsymbol{x}_i^{(v)}),\boldsymbol{y}^c) + \frac{\boldsymbol{Q}_{i,v}}{NC}\sum_{c=1}^C\sum_{\boldsymbol{y}^c\in\mathcal{Y}^*}\sum_{k=1}^{T^{\boldsymbol{y}^c}}\sum_{i,v\in\mathcal{U}_k^{\boldsymbol{y}^c}}\ell(\psi_c(a_k^{c,s},a_k^{c,o}),\boldsymbol{y}^c) \\
&\geq \frac{q}{NC}\sum_{c=1}^C\sum_{\boldsymbol{y}^c\in\mathcal{Y}^*}\sum_{i,v\in\mathcal{U}^{\boldsymbol{y}^c}}\ell(\psi_c\circ\phi^{(v)}(\boldsymbol{x}_i^{(v)}),\boldsymbol{y}^c) + \sum_{c=1}^C\sum_{\boldsymbol{y}^c\in\mathcal{Y}^*}\sum_{k=1}^{T^{\boldsymbol{y}^c}}\frac{q|\mathcal{U}_k^{\boldsymbol{y}^c}|}{NC}\ell(\psi_c(a_k^{c,s},a_k^{c,o}),\boldsymbol{y}^c).
\end{aligned} \quad (44)$$

Furthermore, we can obtain the upper bound for the empirical risk:

$$\frac{1}{NC}\sum_{i=1}^{N}\sum_{c=1}^{C}\ell\left(\sum_{v=1}^{V}(\boldsymbol{Q}_{i,v}f_c(\boldsymbol{x}_i^{(v)})),\boldsymbol{Y}_{i,c}\right)\geq\frac{q}{NC}\sum_{c=1}^{C}\sum_{\boldsymbol{y}^c\in\mathcal{Y}^*}\sum_{i,v\in\mathcal{U}^{\boldsymbol{y}^c}}\ell(\psi_c\circ\phi^{(v)}(\boldsymbol{x}_i^{(v)}),\boldsymbol{y}^c)$$

$$+\sum_{c=1}^{C}\sum_{\boldsymbol{y}^c\in\mathcal{Y}^*}\sum_{k=1}^{T^{\boldsymbol{y}^c}}\frac{q|\mathcal{U}_k^{\boldsymbol{y}^c}|}{NC}\ell(\psi_c(a_k^{c,s},a_k^{c,o}),\boldsymbol{y}^c)+\frac{1}{NC}\sum_{i=1}^{N}\sum_{c=1}^{C}\Delta\ell_i^c(\boldsymbol{Q}_{i,:},\boldsymbol{X}_{i,:},\boldsymbol{Y}_{i,c}). \tag{45}$$

The random variables of the quality score of the $v$-th view and the $c$-th category are denoted as $\boldsymbol{Q}^{(v)}$ and $\boldsymbol{Y}^c$. Similarly, the population risk can be decomposed as

$$\mathbb{E}_{(\boldsymbol{X},\boldsymbol{Y})\sim\mathcal{X}\times\mathcal{Y}}[\ell(f^{av}(\boldsymbol{X},\boldsymbol{Q}),\boldsymbol{Y})]$$

$$=\sum_{c=1}^{C}\sum_{\boldsymbol{y}^c\in\mathcal{Y}^*}\mathbb{P}(\boldsymbol{Y}^c=y^c)\mathbb{E}_{(\boldsymbol{X},\boldsymbol{Y}^c)\sim\mathcal{X}\times\mathcal{Y}^*}[\ell(f^{av}(\boldsymbol{X},\boldsymbol{Q}),\boldsymbol{Y}^c)|\boldsymbol{Y}^c=y^c]$$

$$=\sum_{c=1}^{C}\sum_{\boldsymbol{y}^c\in\mathcal{Y}^*}\mathbb{P}(\boldsymbol{Y}^c=y^c,(\boldsymbol{X},\boldsymbol{R})\notin\mathcal{R}_{c,\gamma}^x)\mathbb{E}_{(\boldsymbol{X},\boldsymbol{Y}^c)\sim\mathcal{X}\times\mathcal{Y}^*}[\ell(f_c^{av}(\boldsymbol{X},\boldsymbol{Q}),\boldsymbol{Y}^c)|\boldsymbol{Y}^c=y^c,(\boldsymbol{X},\boldsymbol{R})\notin\mathcal{R}_{c,\gamma}^x]$$

$$+\sum_{c=1}^{C}\sum_{\boldsymbol{y}^c\in\mathcal{Y}^*}\mathbb{P}(\boldsymbol{Y}^c=y^c,(\boldsymbol{X},\boldsymbol{R})\in\mathcal{R}_{c,\gamma}^x)\mathbb{E}_{(\boldsymbol{X},\boldsymbol{Y}^c)\sim\mathcal{X}\times\mathcal{Y}^*}[\ell(f_c^{av}(\boldsymbol{X},\boldsymbol{Q}),\boldsymbol{Y}^c)|\boldsymbol{Y}^c=y^c,(\boldsymbol{X},\boldsymbol{R})\in\mathcal{R}_{c,\gamma}^x].$$

$$\tag{46}$$

Since $\mathbb{E}_{(\boldsymbol{X},\boldsymbol{Y}^c)\sim\mathcal{X}\times\mathcal{Y}^*}[\ell(f_c^{av}(\boldsymbol{X},\boldsymbol{Q}),\boldsymbol{Y}^c)]=\mathbb{E}_{(\boldsymbol{X},\boldsymbol{Y}^c)\sim\mathcal{X}\times\mathcal{Y}^*}[\ell(\sum_{v=1}^{V}\boldsymbol{Q}^{(v)}f_c(\boldsymbol{X}^{(v)}),\boldsymbol{Y}^c))]$, we can further decompose it based on the convexity of $\ell$:

$$\mathbb{E}_{(\boldsymbol{X},\boldsymbol{Y}^c)\sim\mathcal{X}\times\mathcal{Y}^*}[\ell(f_c^{av}(\boldsymbol{X},\boldsymbol{Q}),\boldsymbol{Y}^c)]=\mathbb{E}_{(\boldsymbol{X},\boldsymbol{Y}^c)\sim\mathcal{X}\times\mathcal{Y}^*}[\ell(\sum_{v=1}^{V}\boldsymbol{Q}^{(v)}f_c(\boldsymbol{X}^{(v)}),\boldsymbol{Y}^c)]$$

$$=\mathbb{E}_{(\boldsymbol{X},\boldsymbol{Y}^c)\sim\mathcal{X}\times\mathcal{Y}^*}[\sum_{v=1}^{V}\boldsymbol{Q}^{(v)}\ell(f_c(\boldsymbol{X}^{(v)}),\boldsymbol{Y}^c)]+\mathbb{E}_{(\boldsymbol{X},\boldsymbol{Y}^c)\sim\mathcal{X}\times\mathcal{Y}^*}[\Delta\ell^c(\boldsymbol{Q},\boldsymbol{X},\boldsymbol{Y}^c)], \tag{47}$$

where $\ell^c(\boldsymbol{Q},\boldsymbol{Y}^c)$ represents the change in loss $\ell$ associated with class $c$ induced by view fusion. Then, we can obtain the following transformation combing Eqs. (46) and (47):

$$\mathbb{E}_{(\boldsymbol{X},\boldsymbol{Y})\sim\mathcal{X}\times\mathcal{Y}}[\ell(f^{av}(\boldsymbol{X},\boldsymbol{Q}),\boldsymbol{Y})]$$

$$=\sum_{c=1}^{C}\sum_{\boldsymbol{y}^c\in\mathcal{Y}^*}\mathbb{P}(\boldsymbol{Y}^c=y^c,(\boldsymbol{X},\boldsymbol{R})\notin\mathcal{R}_{c,\gamma}^x)\mathbb{E}_{(\boldsymbol{X},\boldsymbol{Y}^c)\sim\mathcal{X}\times\mathcal{Y}^*}[\sum_{v=1}^{V}\boldsymbol{Q}^{(v)}\ell(f_c(\boldsymbol{X}^{(v)}),\boldsymbol{Y}^c)|\boldsymbol{Y}^c=y^c,(\boldsymbol{X},\boldsymbol{R})\notin\mathcal{R}_{c,\gamma}^x]$$

$$+\sum_{c=1}^{C}\sum_{\boldsymbol{y}^c\in\mathcal{Y}^*}\mathbb{P}(\boldsymbol{Y}^c=y^c,(\boldsymbol{X},\boldsymbol{R})\in\mathcal{R}_{c,\gamma}^x)\mathbb{E}_{(\boldsymbol{X},\boldsymbol{Y}^c)\sim\mathcal{X}\times\mathcal{Y}^*}[\sum_{v=1}^{V}\boldsymbol{Q}^{(v)}\ell(f_c(\boldsymbol{X}^{(v)}),\boldsymbol{Y}^c)|\boldsymbol{Y}^c=y^c,(\boldsymbol{X},\boldsymbol{R})\in\mathcal{R}_{c,\gamma}^x]$$

$$+\sum_{c=1}^{C}\sum_{\boldsymbol{y}^c\in\mathcal{Y}^*}\mathbb{P}(\boldsymbol{Y}^c=y^c)\mathbb{E}_{(\boldsymbol{X},\boldsymbol{Y}^c)\sim\mathcal{X}\times\mathcal{Y}^*}[\Delta\ell^c(\boldsymbol{Q},\boldsymbol{X},\boldsymbol{Y}^c)|\boldsymbol{Y}^c=y^c]$$

$$\leq\sum_{c=1}^{C}\sum_{\boldsymbol{y}^c\in\mathcal{Y}^*}\mathbb{P}(\boldsymbol{Y}^c=y^c,(\boldsymbol{X}^{(j)},\phi^j(\boldsymbol{X}^{(j)}))\notin\mathcal{R}_{c,\gamma}^x)\mathbb{E}_{(\boldsymbol{X},\boldsymbol{Y}^c)\sim\mathcal{X}\times\mathcal{Y}^*,\boldsymbol{X}^j\sim\boldsymbol{X}}[\ell(\psi_c\circ\phi^j(\boldsymbol{X}^j),y^c)|\boldsymbol{Y}^c=y^c,(\boldsymbol{X}^{(j)},$$

$$\phi^j(\boldsymbol{X}^{(j)}))\notin\mathcal{R}_{c,\gamma}^x]+\sum_{c=1}^{C}\sum_{\boldsymbol{y}^c\in\mathcal{Y}^*}\sum_{k=1}^{T^{\boldsymbol{y}^c}}\mathbb{P}(\boldsymbol{Y}^c=y^c,\boldsymbol{X}^{(j)}=a_k^{c,x},\phi^j(\boldsymbol{X}^{(j)})=(a_k^{c,s},a_k^{c,o}))\ell(\psi_c(a_k^{c,s},a_k^{c,o}),y^c))$$

$$+\sum_{c=1}^{C}\sum_{\boldsymbol{y}^c\in\mathcal{Y}^*}\mathbb{P}(\boldsymbol{Y}^c=y^c)\mathbb{E}_{(\boldsymbol{X},\boldsymbol{Y}^c)\sim\mathcal{X}\times\mathcal{Y}^*}[\Delta\ell^c(\boldsymbol{Q},\boldsymbol{X},\boldsymbol{Y}^c)|\boldsymbol{Y}^c=y^c].$$

$$\tag{48}$$

Putting Eqs. (58) and (48) back into Eq. (41), we have

$$
\mathbb{E}_{(\boldsymbol{X},\boldsymbol{Y})\sim\mathcal{X}\times\mathcal{Y}}[\ell(f^{av}(\boldsymbol{X},\boldsymbol{Q}),\boldsymbol{Y})] - \frac{1}{NC}\sum_{i=1}^{N}\sum_{c=1}^{C}\ell\left(\sum_{v=1}^{V}(\boldsymbol{Q}_{i,v}f_c(\boldsymbol{x}_i^{(v)})),Y_{i,c}\right)
$$

$$
\leq \sum_{c=1}^{C}\sum_{\boldsymbol{y}^c\in\mathcal{Y}^*}\mathbb{P}(\boldsymbol{Y}^c=y^c,(\boldsymbol{X}^{(j)},\phi^j(\boldsymbol{X}^{(j)}))\notin\mathcal{R}_{c,\gamma}^x)\mathbb{E}_{(\boldsymbol{X},\boldsymbol{Y}^c)\sim\mathcal{X}\times\mathcal{Y}^*,\boldsymbol{X}^j\sim\boldsymbol{X}}[\ell(\psi_c\circ\phi^j(\boldsymbol{X}^j),y^c)|\boldsymbol{Y}^c=y^c,(\boldsymbol{X}^{(j)},
$$

$$
\phi^j(\boldsymbol{X}^{(j)}))\notin\mathcal{R}_{c,\gamma}^x] - \sum_{c=1}^{C}\sum_{\boldsymbol{y}^c\in\mathcal{Y}^*}\mathbb{P}(\boldsymbol{Y}^c=y^c,(\boldsymbol{X}^{(j)},\phi^j(\boldsymbol{X}^{(j)}))\notin\mathcal{R}_{c,\gamma}^x)\frac{1}{|\mathcal{U}^{\boldsymbol{y}^c}|}\sum_{i,v\in\mathcal{U}^{\boldsymbol{y}^c}}\ell(\psi_c\circ\phi^{(v)}(\boldsymbol{x}_i^{(v)}),\boldsymbol{y}^c)
$$

$$
+ \sum_{c=1}^{C}\sum_{\boldsymbol{y}^c\in\mathcal{Y}^*}\mathbb{P}(\boldsymbol{Y}^c=y^c,(\boldsymbol{X}^{(j)},\phi^j(\boldsymbol{X}^{(j)}))\notin\mathcal{R}_{c,\gamma}^x)\frac{1}{|\mathcal{U}^{\boldsymbol{y}^c}|}\sum_{i,v\in\mathcal{U}^{\boldsymbol{y}^c}}\ell(\psi_c\circ\phi^{(v)}(\boldsymbol{x}_i^{(v)}),\boldsymbol{y}^c)
$$

$$
- \frac{q}{NC}\sum_{c=1}^{C}\sum_{\boldsymbol{y}^c\in\mathcal{Y}^*}\sum_{i,v\in\mathcal{U}^{\boldsymbol{y}^c}}\ell(\psi_c\circ\phi^{(v)}(\boldsymbol{x}_i^{(v)}),\boldsymbol{y}^c)
$$

$$
+ \sum_{c=1}^{C}\sum_{\boldsymbol{y}^c\in\mathcal{Y}^*}\sum_{k=1}^{T^{\boldsymbol{y}^c}}\mathbb{P}(\boldsymbol{Y}^c=y^c,\boldsymbol{X}^{(j)}=a_k^{c,x},\phi^j(\boldsymbol{X}^{(j)})=(a_k^{c,s},a_k^{c,o}))\ell(\psi_c(a_k^{c,s},a_k^{c,o}),y^c))
$$

$$
- \sum_{c=1}^{C}\sum_{\boldsymbol{y}^c\in\mathcal{Y}^*}\sum_{k=1}^{T^{\boldsymbol{y}^c}}\frac{q|\mathcal{U}_k^{\boldsymbol{y}^c}|}{NC}\ell(\psi_c(a_k^{c,s},a_k^{c,o}),\boldsymbol{y}^c)
$$

$$
+ \sum_{c=1}^{C}\sum_{\boldsymbol{y}^c\in\mathcal{Y}^*}\mathbb{P}(\boldsymbol{Y}^c=y^c)\mathbb{E}_{(\boldsymbol{X},\boldsymbol{Y}^c)\sim\mathcal{X}\times\mathcal{Y}^*}[\Delta\ell^c(\boldsymbol{Q},\boldsymbol{X},\boldsymbol{Y}^c)|\boldsymbol{Y}^c=y^c] - \frac{1}{NC}\sum_{i=1}^{N}\sum_{c=1}^{C}\sum_{\boldsymbol{y}^c\in\mathcal{Y}^*}\Delta\ell_i^c(\boldsymbol{Q}_{i,:},\boldsymbol{X}_{i,:},Y_{i,c}=y^c).
$$

$$(49)$$

Let $\mathcal{S}_{x,y}=\sup_{c\in[C]}\sup_{(\boldsymbol{X},\boldsymbol{Y}^c)\in\mathcal{X}\times\mathcal{Y}^*}\sum_{v=1}^{V}\boldsymbol{Q}^{(v)}\ell(f_c(\boldsymbol{X}^{(v)}),\boldsymbol{Y}^c)$ denote the maximum attainable losses among all classes and $\mathcal{S}_{x,y}^s=\sup_{c\in[C]}\sup_{i\in[n]}\boldsymbol{Q}_{i,v}\sum_{v=1}^{V}\ell(f_c(\boldsymbol{x}_i^{(v)}),Y_{i,c})$ represent the maximum instance-level losses. We consider the results of Eq. (49) as four terms, with the upper bound of the first term as follows:

$$
\mathcal{P}_1=\sum_{c=1}^{C}\sum_{\boldsymbol{y}^c\in\mathcal{Y}^*}\mathbb{P}(\boldsymbol{Y}^c=y^c,(\boldsymbol{X}^{(j)},\phi^j(\boldsymbol{X}^{(j)}))\notin\mathcal{R}_{c,\gamma}^x)(\mathbb{E}_{(\boldsymbol{X},\boldsymbol{Y}^c)\sim\mathcal{X}\times\mathcal{Y}^*,\boldsymbol{X}^j\sim\boldsymbol{X}}[q\ell(\psi_c\circ\phi^j(\boldsymbol{X}^j),y^c)|\boldsymbol{Y}^c=y^c,
$$

$$
(\boldsymbol{X}^{(j)},\phi^j(\boldsymbol{X}^{(j)}))\notin\mathcal{R}_{c,\gamma}^x] - \frac{1}{|\mathcal{U}^{\boldsymbol{y}^c}|}\sum_{i,v\in\mathcal{U}^{\boldsymbol{y}^c}}\ell(\psi_c\circ\phi^{(v)}(\boldsymbol{x}_i^{(v)}),\boldsymbol{y}^c))
$$

$$
\leq \sum_{c=1}^{C}\sum_{\boldsymbol{y}^c\in\mathcal{Y}^*}\mathbb{P}(\boldsymbol{Y}^c=y^c,(\boldsymbol{X},\boldsymbol{R})\notin\mathcal{R}_{c,\gamma}^x)\mathbb{E}_{(\boldsymbol{X},\boldsymbol{Y}^c)\sim\mathcal{X}\times\mathcal{Y}^*,\boldsymbol{X}^j\sim\boldsymbol{X}}[\sum_{v=1}^{V}\boldsymbol{Q}^{(v)}\ell(f_c(\boldsymbol{X}^{(v)}),\boldsymbol{Y}^c)|\boldsymbol{Y}^c=y^c,(\boldsymbol{X},\boldsymbol{R})\notin\mathcal{R}_{c,\gamma}^x]
$$

$$
\leq \sum_{c=1}^{C}\sum_{\boldsymbol{y}^c\in\mathcal{Y}^*}\mathbb{P}(\boldsymbol{Y}^c=y^c)\frac{\gamma}{\sqrt{NV}}\mathcal{S}_{x,y}=\frac{C\gamma}{\sqrt{NV}}\mathcal{S}_{x,y}.
$$

$$(50)$$

Define $p_k^c=\mathbb{P}(\boldsymbol{Y}^c=y^c,\boldsymbol{X}^{(j)}=a_k^{c,x},\phi^j(\boldsymbol{X}^{(j)})=(a_k^{c,s},a_k^{c,o}))$ for $k\in[T^{\boldsymbol{y}^c}]$, $p_{T^{\boldsymbol{y}^c}+1}^c=\mathbb{P}(\boldsymbol{Y}^c=y^c,(\boldsymbol{X}^{(j)},\phi^j(\boldsymbol{X}^{(j)}))\notin\mathcal{R}_{c,\gamma}^x)$ and $b_k^c=\ell(\psi_c(a_k^{c,s},a_k^{c,o}),y^c))$ for $k\in[T^{\boldsymbol{y}^c}+1]$. Then, we can obtain the following term:

$$
\mathcal{P}_{3,k}^c=\sum_{t=1}^{T^{\boldsymbol{y}^c}}(p_t^c-\frac{|\mathcal{U}_t^{\boldsymbol{y}^c}|}{NCq^{-1}})b_t^c-(p_k^c-\frac{|\mathcal{U}_k^{\boldsymbol{y}^c}|}{NCq^{-1}})b_k^c. \tag{51}
$$

Applying Lemma 2, for any $\epsilon>0$ and $k\in\left[T^{\boldsymbol{y}^c}\right]$, we have

$$
\mathbb{P}\left(\mathcal{P}_{3,k}^c\geq\epsilon\right)\leq\exp\left(-\frac{NCq^{-1}\epsilon^2}{2\left(\sum_{t=1}^{T^{\boldsymbol{y}^c}}p_t^c\left(b_t^c\right)^2-p_k^c\left(b_k^c\right)^2\right)}\right). \tag{52}
$$

Similarly, we get the following inequality regarding $k = T^{\boldsymbol{y}^c} + 1$:

$$\mathbb{P}\left(p_{T^{\boldsymbol{y}^c}+1}^c - \frac{|\mathcal{U}^{\boldsymbol{y}^c}|}{NCq^{-1}} \geq \epsilon\right) \leq \exp\left(-\frac{NCq^{-1}\epsilon^2}{2p_{T^{\boldsymbol{y}^c}+1}^c}\right). \tag{53}$$

Let $\delta$ be the right-hand side of Eq. (52) and (53), respectively, we can obtain the following variants:

$$\begin{cases} \mathbb{P}\left(\mathcal{P}_{3,k}^c \geq \sqrt{\sum_{t=1}^{T^{\boldsymbol{y}^c}} p_t^c \left(b_t^c\right)^2 - p_k^c \left(b_k^c\right)^2} \sqrt{\frac{2\log(1/\delta)}{NCq^{-1}}}\right) \leq \delta, \forall k \in [T^{\boldsymbol{y}^c}] \\ \\ \mathbb{P}\left(p_{T^{\boldsymbol{y}^c}+1}^c - \frac{|\mathcal{U}^{\boldsymbol{y}^c}|}{NCq^{-1}} \geq \sqrt{\frac{2p_{T^{\boldsymbol{y}^c}+1}^c \log(1/\delta)}{NCq^{-1}}}\right) \leq \delta. \end{cases} \tag{54}$$

Taking union bounds over all $\boldsymbol{y}^c \in \boldsymbol{y}^*$ ($|\boldsymbol{y}^*| = 2|$) and $k \in [T^{\boldsymbol{y}^c}]$, we have

$$\begin{cases} \mathbb{P}\left(\mathcal{P}_{3,k}^c \geq \sqrt{\sum_{t=1}^{T^{\boldsymbol{y}^c}} p_t^c \left(b_t^c\right)^2 - p_k^c \left(b_k^c\right)^2} \sqrt{\frac{2\log(2T^{\boldsymbol{y}^c}/\delta)}{NCq^{-1}}}\right) \leq \delta, \forall k \in [T^{\boldsymbol{y}^c}] \\ \\ \mathbb{P}\left(p_{T^{\boldsymbol{y}^c}+1}^c - \frac{|\mathcal{U}^{\boldsymbol{y}^c}|}{NCq^{-1}} \geq \sqrt{\frac{2p_{T^{\boldsymbol{y}^c}+1}^c \log(2/\delta)}{NCq^{-1}}}\right) \leq \delta. \end{cases} \tag{55}$$

Then, for any $\delta > 0$, with probability at least $1 - \delta$, we have the following scaling regrading the second term of Eq. (49) by using Jensen's inequality:

$$\mathcal{P}_2 = \frac{1}{|\mathcal{U}^{\boldsymbol{y}^c}|} \sum_{c=1}^{C} \sum_{y^c \in \mathcal{Y}^*} \left(\mathbb{P}\left(\boldsymbol{Y}^c = y^c, (\boldsymbol{X}^{(j)}, \phi^j(\boldsymbol{X}^{(j)})) \notin \mathcal{R}_{c,\gamma}^x\right) - \frac{q|\mathcal{U}^{\boldsymbol{y}^c}|}{NC}\right) \sum_{i,v \in \mathcal{U}^{\boldsymbol{y}^c}} \ell(\psi_c \circ \phi^{(v)}(\boldsymbol{x}_i^{(v)}), \boldsymbol{y}^c)$$

$$\leq \sum_{c=1}^{C} \sum_{y^c \in \mathcal{Y}^*} \sqrt{\mathbb{P}(\boldsymbol{Y}^c = y^c, (\boldsymbol{X}^{(j)}, \phi^j(\boldsymbol{X}^{(j)})) \notin \mathcal{R}_{c,\gamma}^x)} \frac{\sum_{i,j \in \mathcal{U}^{\boldsymbol{y}^c}} \ell(\psi_c \circ \phi^{(v)}(\boldsymbol{x}_i^{(v)}), \boldsymbol{y}^c)}{|\mathcal{U}^{\boldsymbol{y}^c}|} \sqrt{\frac{2\log(2/\delta)}{NCq^{-1}}}$$

$$= \sum_{c=1}^{C} \sum_{y^c \in \mathcal{Y}^*} \sqrt{\mathbb{P}(\boldsymbol{Y}^c = y^c)} \sqrt{\mathbb{P}(\boldsymbol{X}^{(j)}, \phi^j(\boldsymbol{X}^{(j)})) \notin \mathcal{R}_{c,\gamma}^x)} \frac{\sum_{i,j \in \mathcal{U}^{\boldsymbol{y}^c}} \ell(\psi_c \circ \phi^{(v)}(\boldsymbol{x}_i^{(v)}), \boldsymbol{y}^c)}{|\mathcal{U}^{\boldsymbol{y}^c}|} \sqrt{\frac{2\log(2/\delta)}{NCq^{-1}}}$$

$$\leq \sum_{c=1}^{C} \frac{\sqrt{\gamma}}{(NV)^{1/4}} \frac{\sum_{i,j \in \mathcal{U}^{\boldsymbol{y}^c}} \ell(\psi_c \circ \phi^{(v)}(\boldsymbol{x}_i^{(v)}), \boldsymbol{y}^c)}{|\mathcal{U}^{\boldsymbol{y}^c}|} \sqrt{|\mathcal{Y}^*|} \sqrt{\sum_{y^c \in \mathcal{Y}^*} \mathbb{P}(\boldsymbol{Y}^c = y^c)} \sqrt{\frac{2\log(2/\delta)}{NCq^{-1}}}$$

$$\leq \mathcal{S}_{x,y}^s \frac{2\sqrt{\gamma}}{(NV)^{1/4}} \sqrt{\frac{C\log(2/\delta)}{Nq^{-1}}}. \tag{56}$$

Similarly, for $\mathcal{P}_{3,k}^c$, with probability at least $1 - \delta$, we have

$$\mathcal{P}_{3,k}^c \leq \sqrt{\sum_{t=1}^{T^{\boldsymbol{y}^c}} p_t^y \left(b_t^y\right)^2 - p_k^c \left(b_k^y\right)^2} \sqrt{\frac{2\log(2T^{\boldsymbol{y}^c}/\delta)}{NCq^{-1}}}$$

$$\leq \mathcal{S}_{x,y} \sqrt{\sum_{t=1}^{T^{\boldsymbol{y}^c}} p_t^y - p_k^c} \sqrt{\frac{2\log(2T^{\boldsymbol{y}^c}/\delta)}{NCq^{-1}}}$$

$$= \mathcal{S}_{x,y} \sqrt{\mathbb{P}\left(\boldsymbol{Y}^c = \boldsymbol{y}^c \bigcap (\boldsymbol{X}, \boldsymbol{R}) \in \mathcal{R}_{c,\gamma}^x \bigcap \left(\boldsymbol{X}^{(j)}, \phi^{(j)}\left(\boldsymbol{X}^{(j)}\right)\right) \neq (a_k^{c,x}, a_k^{c,s}, a_k^{c,o})\right)} \sqrt{\frac{2\log(2T^{\boldsymbol{y}^c}/\delta)}{NCq^{-1}}}$$

$$\leq \mathcal{S}_{x,y} \sqrt{\mathbb{P}(\boldsymbol{Y}^c = \boldsymbol{y}^c)} \sqrt{\frac{2\log(2T^{\boldsymbol{y}^c}/\delta)}{NCq^{-1}}}. \tag{57}$$

Next, we can derive the constraint result for the third term:

$$
\begin{aligned}
\mathcal{P}_3 &= \sum_{c=1}^{C} \sum_{\boldsymbol{y}^c \in \mathcal{Y}^*} \sum_{k=1}^{T^{\boldsymbol{y}^c}} \left( \mathbb{P}(\boldsymbol{Y}^c = y^c, \boldsymbol{X}^{(j)} = a_k^{c,x}, \phi^j(\boldsymbol{X}^{(j)}) = (a_k^{c,s}, a_k^{c,o})) - \frac{q|\mathcal{U}_k^{\boldsymbol{y}^c}|}{NC} \right) \ell(\psi_c(a_k^{c,s}, a_k^{c,o}), \boldsymbol{y}^c) \\
&= \sum_{c=1}^{C} \sum_{\boldsymbol{y}^c \in \mathcal{Y}^*} \frac{1}{T^{\boldsymbol{y}^c} - 1} \sum_{k=1}^{T^{\boldsymbol{y}^c}} \mathcal{P}_{3,k}^c \\
&\leq \sum_{c=1}^{C} \sum_{\boldsymbol{y}^c \in \mathcal{Y}^*} \frac{T^{\boldsymbol{y}^c}}{T^{\boldsymbol{y}^c} - 1} \mathcal{S}_{x,y} \sqrt{\mathbb{P}(\boldsymbol{Y}^c = \boldsymbol{y}^c)} \sqrt{\frac{2\log(2T^{\boldsymbol{y}^c}/\delta)}{NCq^{-1}}} \\
&\leq 2 \sum_{c=1}^{C} \sum_{\boldsymbol{y}^c \in \mathcal{Y}^*} \mathcal{S}_{x,y} \sqrt{\mathbb{P}(\boldsymbol{Y}^c = \boldsymbol{y}^c)} \sqrt{\frac{2\log(2T^{\boldsymbol{y}^c}/\delta)}{NCq^{-1}}}.
\end{aligned}
$$

(58)

The property of $T^{\boldsymbol{y}^c}$ is as follows:

$$
T^{\boldsymbol{y}^c} = |\mathcal{R}_{c,\gamma}^x| \leq \exp\left( H(\boldsymbol{R}_{\boldsymbol{Y}^c}) + c_\phi \sqrt{\frac{d\log(\sqrt{NV}/\gamma)}{2}} \right).
$$

(59)

Combining Eqs. (58) and (59), we can get the following bound by using Jensen's inequality:

$$
\begin{aligned}
\mathcal{P}_3 &\leq 2 \sum_{c=1}^{C} \mathcal{S}_{x,y} \sum_{\boldsymbol{y}^c \in \mathcal{Y}^*} \sqrt{\mathbb{P}(\boldsymbol{Y}^c = \boldsymbol{y}^c)} \sqrt{\frac{2\left( H(\boldsymbol{R}_{\boldsymbol{Y}^c}) + c_\phi \sqrt{\frac{d\log(\sqrt{NV}/\gamma)}{2}} \right) + 2\log(2/\delta)}{NCq^{-1}}} \\
&\leq 2 \sum_{c=1}^{C} \mathcal{S}_{x,y} \sqrt{|\mathcal{Y}^*|} \sqrt{\sum_{y^c \in \mathcal{Y}^*} \mathbb{P}(\boldsymbol{Y}^c = y^c)} \sqrt{\frac{2\left( H(\boldsymbol{R}_{\boldsymbol{Y}^c}) + c_\phi \sqrt{\frac{d\log(\sqrt{NV}/\gamma)}{2}} \right) + 2\log(2/\delta)}{NCq^{-1}}} \\
&= 4 \sum_{c=1}^{C} \mathcal{S}_{x,y} \sqrt{\frac{H(\boldsymbol{R}_{\boldsymbol{Y}^c}) + c_\phi \sqrt{\frac{d\log(\sqrt{NV}/\gamma)}{2}} + \log(2/\delta)}{NCq^{-1}}} \\
&\leq 4C \mathcal{S}_{x,y} \sqrt{\sum_{c=1}^{C} \frac{H(\boldsymbol{R}|\boldsymbol{Y}^c) + c_\phi \sqrt{\frac{d\log(\sqrt{NV}/\gamma)}{2}} + \log(2/\delta)}{NCq^{-1}}}.
\end{aligned}
$$

(60)

By applying the chain rule, we obtain the following conclusion:

$$
\begin{aligned}
H(\boldsymbol{R}|\boldsymbol{Y}^c) &= I(\boldsymbol{X}^{(1)}; \boldsymbol{R}|\boldsymbol{Y}^c) + H(\boldsymbol{R}|\boldsymbol{Y}^c, \boldsymbol{X}^{(1)}) \\
&= I(\boldsymbol{X}^{(1)}; \boldsymbol{S}^{(1)}, \ldots, \boldsymbol{S}^{(V)}, \boldsymbol{O}^{(1)}, \ldots, \boldsymbol{O}^{(V)}|\boldsymbol{Y}^c) + H(\boldsymbol{R}|\boldsymbol{Y}^c, \boldsymbol{X}^{(1)}) \\
&= I(\boldsymbol{X}^{(1)}; \boldsymbol{S}^{(1)}, \ldots, \boldsymbol{S}^{(V)}, \boldsymbol{O}^{(1)}|\boldsymbol{Y}^c) + I(\boldsymbol{X}^{(1)}; \{\boldsymbol{O}^{(v)}\}_{v=2}^{V}|\boldsymbol{Y}^c, \boldsymbol{S}^{(1)}, \ldots, \boldsymbol{S}^{(V)}, \boldsymbol{O}^{(1)}) + H(\boldsymbol{R}|\boldsymbol{Y}^c, \boldsymbol{X}^{(1)}).
\end{aligned}
$$

(61)

Since our model achieves complete feature separation, with seamlessly integrated shared representations, no overlap among special representations, and no redundancy between shared and special features, we have the following equality:

$$
I(\boldsymbol{X}^{(1)}; \{\boldsymbol{O}^{(v)}\}_{v=2}^{V}|\boldsymbol{Y}^c, \boldsymbol{S}^{(1)}, \ldots, \boldsymbol{S}^{(V)}, \boldsymbol{O}^{(1)}) = I(\boldsymbol{X}^{(1)}; \{\boldsymbol{O}^{(v)}\}_{v=2}^{V}) = 0.
$$

(62)

Thus, we can further obtain

$$
\begin{aligned}
H(\boldsymbol{R}|\boldsymbol{Y}^c) &\leq I(\boldsymbol{X}^{(1)}; \boldsymbol{S}^{(1)}, \ldots, \boldsymbol{S}^{(V)}, \boldsymbol{O}^{(1)}|\boldsymbol{Y}^c) + H(\boldsymbol{R}|\boldsymbol{Y}^c, \boldsymbol{X}^{(1)}) \\
&\leq \sum_{v=1}^{V} I(\boldsymbol{X}^{(v)}; \{\boldsymbol{S}^{(v)}\}_{v=1}^{V}, \boldsymbol{O}^{(v)}|\boldsymbol{Y}^c) + H(\boldsymbol{R}|\boldsymbol{Y}^c, \boldsymbol{X}^{(1)})
\end{aligned}
$$

(63)

Putting the above back into Eq. 60, we have

$$\mathcal{P}_3 \leq 4C\mathcal{S}_{x,y}\sqrt{\sum_{c=1}^{C}\frac{\sum_{v=1}^{V}I(\boldsymbol{X}^{(v)};\boldsymbol{S}^{(v)},\boldsymbol{O}^{(v)}|\boldsymbol{Y}^c)+H(\boldsymbol{R}|\boldsymbol{Y}^c,\boldsymbol{X}^{(1)})+c_\phi\sqrt{\frac{d\log(\sqrt{NV}/\gamma)}{2}}+\log(2/\delta)}{NCq^{-1}}}.$$

(64)

Regarding the fourth term, as the accuracy of the view reconstruction increases, indicated by a larger value of $\boldsymbol{Q}$, the loss discrepancy $\Delta\ell_i^c(\boldsymbol{Q}_{i,:},\boldsymbol{X}_{i,:},\boldsymbol{Y}_{i,c})$ becomes more significant, while the corresponding generalization error bound becomes tighter. Therefore, we interpret the fourth term as the generalization error induced by the quality of view reconstruction, i.e.,

$$\mathcal{P}_4 = \overline{\text{gen}}_{rec}(\boldsymbol{Q},\boldsymbol{X},\boldsymbol{Y}) = \sum_{c=1}^{C}\sum_{\boldsymbol{y}^c\in\mathcal{Y}^*}\mathbb{P}(\boldsymbol{Y}^c=\boldsymbol{y}^c)\mathbb{E}_{(\boldsymbol{X},\boldsymbol{Y}^c)\sim\mathcal{X}\times\mathcal{Y}^*}[\Delta\ell^c(\boldsymbol{Q},\boldsymbol{X},\boldsymbol{Y}^c)|\boldsymbol{Y}^c=\boldsymbol{y}^c]$$

$$-\frac{1}{NC}\sum_{i=1}^{N}\sum_{c=1}^{C}\sum_{\boldsymbol{y}^c\in\mathcal{Y}^*}\Delta\ell_i^c(\boldsymbol{Q}_{i,:},\boldsymbol{X}_{i,:},\boldsymbol{Y}_{i,c}=\boldsymbol{y}^c).$$

(65)

Therefore, with probability at least $1-\delta$, the generalization error bound of our model is

$$\mathbb{E}_{(\boldsymbol{X},\boldsymbol{Y})\sim\mathcal{X}\times\mathcal{Y}}[\ell(f^{av}(\boldsymbol{X},\boldsymbol{Q}),\boldsymbol{Y})] - \frac{1}{NC}\sum_{i=1}^{N}\sum_{c=1}^{C}\ell\left(\sum_{v=1}^{V}(\boldsymbol{Q}_{i,v}f_c(\boldsymbol{x}_i^{(v)})),\boldsymbol{Y}_{i,c}\right)$$

$$\leq \frac{C\gamma}{\sqrt{NV}}\mathcal{S}_{x,y} + \mathcal{S}_{x,y}^s\frac{2\sqrt{\gamma q}}{(NV)^{1/4}}\sqrt{\frac{C\log(2/\delta)}{N}} + \overline{\text{gen}}_{rec}(\boldsymbol{Q},\boldsymbol{X},\boldsymbol{Y})$$

$$+ 4C\mathcal{S}_{x,y}\sqrt{q}\sqrt{\sum_{c=1}^{C}\frac{\sum_{v=1}^{V}I(\boldsymbol{X}^{(v)};\boldsymbol{S}^{(v)},\boldsymbol{O}^{(v)}|\boldsymbol{Y}^c)+H(\boldsymbol{R}|\boldsymbol{Y}^c,\boldsymbol{X}^{(1)})+c_\phi\sqrt{\frac{d\log(\sqrt{NV}/\gamma)}{2}}+\log(2/\delta)}{NC}}$$

$$= \frac{\widetilde{\mathcal{K}}_1}{\sqrt{NV}} + \frac{\widetilde{\mathcal{K}}_2}{N^{3/4}V^{1/4}} + \widetilde{\mathcal{K}}_3\sqrt{\frac{\sum_{c=1}^{C}\left(\sum_{v=1}^{V}I(\boldsymbol{X}^{(v)};\boldsymbol{S}^{(v)},\boldsymbol{O}^{(v)}|\boldsymbol{Y}^c)+\widetilde{\mathcal{K}}_4\right)}{NC}} + \overline{\text{gen}}_{rec}(\boldsymbol{Q},\boldsymbol{X},\boldsymbol{Y}),$$

(66)

where

$$\widetilde{\mathcal{K}}_1 = C\gamma\mathcal{S}_{x,y}$$
$$\widetilde{\mathcal{K}}_2 = 2\mathcal{S}_{x,y}^s\sqrt{C\gamma q\log(2/\delta)}$$
$$\widetilde{\mathcal{K}}_3 = 4C\mathcal{S}_{x,y}\sqrt{q}$$
$$\widetilde{\mathcal{K}}_4 = H(\boldsymbol{R}|\boldsymbol{Y}^c,\boldsymbol{X}^{(1)}) + c_\phi\sqrt{\frac{d\log(\sqrt{NV}/\gamma)}{2}} + \log(2/\delta).$$

(67)

Since $\boldsymbol{S}^{(v)}$ and $\boldsymbol{O}^{(v)}$ are generated by $\boldsymbol{X}^{(v)}$, we have the following transformation:

$$\sum_{c=1}^{C}\sum_{v=1}^{V}I(\boldsymbol{X}^{(v)};\boldsymbol{S}^{(v)},\boldsymbol{O}^{(v)}|\boldsymbol{Y}^c)$$

$$= \sum_{c=1}^{C}H((\boldsymbol{S}^{(1)},\ldots,\boldsymbol{S}^{(V)},\boldsymbol{O}^{(1)},\ldots,\boldsymbol{O}^{(V)})|\boldsymbol{Y}^c)$$

$$= H((\boldsymbol{S}^{(1)},\ldots,\boldsymbol{S}^{(V)},\boldsymbol{O}^{(1)},\ldots,\boldsymbol{O}^{(V)})|\boldsymbol{Y})$$

$$= H((\boldsymbol{S}^{(1)},\ldots,\boldsymbol{S}^{(V)},\boldsymbol{O}^{(1)},\ldots,\boldsymbol{O}^{(V)})) - I((\boldsymbol{S}^{(1)},\ldots,\boldsymbol{S}^{(V)},\boldsymbol{O}^{(1)},\ldots,\boldsymbol{O}^{(V)});\boldsymbol{Y}).$$

Based on Theorem 5, it can be inferred that during the model optimization process, $I((\boldsymbol{S}^{(1)},\ldots,\boldsymbol{S}^{(V)},\boldsymbol{O}^{(1)},\ldots,\boldsymbol{O}^{(V)});\boldsymbol{Y})$ continuously approaches its maximum value, leading to a decrease in $\sum_{c=1}^{C}\sum_{v=1}^{V}I(\boldsymbol{X}^{(v)};\boldsymbol{S}^{(v)},\boldsymbol{O}^{(v)}|\boldsymbol{Y}^c)$, which indicates an improvement in generalization performance. $\qquad\square$

## C  Experiment

### C.1  Experiment Setup

Table 3: Detailed information of datasets.

| View | Object | VOC 2007 | Corel 5k | Esp Game | IAPR TC-12 | MIR FLICKR |
|------|--------|----------|----------|----------|------------|------------|
| 1 | CH(64) | DenseHue(100) | DenseHue(100) | DenseHue(100) | DenseHue(100) | DenseHue(100) |
| 2 | CM(225) | DenseSift(1000) | DenseSift(1000) | DenseSift(1000) | DenseSift(1000) | DenseSift(1000) |
| 3 | CORR(144) | GIST(512) | GIST(512) | GIST(512) | GIST(512) | GIST(512) |
| 4 | EDH(73) | HSV(4096) | HSV(4096) | HSV(4096) | HSV(4096) | HSV(4096) |
| 5 | WT(128) | RGB(4096) | RGB(4096) | RGB(4096) | RGB(4096) | RGB(4096) |
| 6 | - | LAB(4096) | LAB(4096) | LAB(4096) | LAB(4096) | LAB(4096) |
| #Label | 31 | 20 | 260 | 268 | 291 | 38 |
| #Instance | 6047 | 9963 | 4999 | 20770 | 19627 | 25000 |

**Datasets and Comparison Methods.** In our experiments, six public multi-view multi-label datasets are selected as shown in Table 3. Their specific descriptions are as follows. **Corel 5k** is composed of 4999 image samples and 260 words, where each word can be regarded as an annotation or label. **IAPRTC12** comprises 19627 high-quality natural images and each image contains 261 labels, including sports, actions, animals, cities, and so on. **ESPGame** is a multi-view multi-label dataset containing 20770 images with 268 corresponding tags. **Pascal07** is a widely utilized dataset for visual object detection and recognition, which contains 9963 images and 20 kinds of objects. **Mirflickr** consists of 25,000 images from the Flickr platform, annotated with a total of 38 tags. **OBJECT** has 6047 instances requiring recognition, which are characterized by five distinct perspectives and annotated with 31 attributes. To validate the effectiveness of TDLSR, we compare it with nine state-of-the-art approaches, i.e., AIMNet [23], DICNet [25], DIMC [38], iMVWL [33], LMVCAT [26], MTD [24], SIP [27], LVSL [43], DM2L [28]. We also provide a comprehensive overview of their sources and functions in Table 4.

**Construction of the Application NBA Dataset.** The NBA dataset crawled from Basketball-Reference [2] includes 16,992 player samples from the regular and playoff seasons spanning 2002 to 2022. Each sample comprises six views capturing different aspects of player performance and background information: 1) *Scoring Statistics*, 20 features including shooting attempts, shooting percentages (scaled by 1000), points, and per-minute scoring efficiency, representing scoring ability and shooting efficiency. 2) *Rebounding and Physical Attributes*, 14 features such as games played, rebounds, fouls, and playing time averages, highlighting physical competitiveness and playing consistency. 3) *Technical Statistics*, 15 features including assists, steals, blocks, turnovers, and triple-doubles, reflecting player's playmaking and defensive contributions. 4) *Advanced Efficiency Metrics*, 10 features measuring comprehensive performance like true shooting percentage, usage rate, player efficiency rating, and per-minute rates. 5) *Player Background*, 41 features encoding age and team membership via one-hot encoding. 6) *Seasonal Context*, 22 features encompassing season indicators, playoff status, also one-hot encoded. Player attributes are structured as an 18-dimensional multi-label vector per sample, which includes 10 award-related labels (e.g., All-Star selection, All-NBA teams, MVP nomination), 5 positional one-hot labels (PG, SG, SF, PF, C) and 3 career stage one-hot labels categorizing each player-season into Early Career (first 25%), Prime Career (middle 50%), and Late Career (last 25%) according to the player's elapsed and remaining career years.

**Implementation Details.** We employ Hamming Loss (HL), Ranking Loss (RL), OneError (OE), Coverage (Cov), Average Precision (AP), and Area Under Curve (AUC) as six metrics to unify the experimental standards. Higher AP and AUC values indicate better performance, while lower HL, RL, OE, and Cov values are preferred. Their evaluation contents are described below: 1) ACC measures the proportion of correctly predicted labels across all samples. 2) RL evaluates the accuracy of the model's ranking of predicted labels compared to true labels. 3) AP computes the area under the precision-recall curve, indicating the average precision achieved across all recall levels. 4) AUC quantifies the probability that a randomly selected positive instance is ranked higher by the model than a randomly selected negative instance across all possible threshold values. 5) OE evaluates whether the top-ranked label predicted by the model is incorrect. 6) Cov computes the number of labels the model needs to traverse to cover all true labels, reflecting the efficiency of the model's predicted label range. The neighbor number $k$ is fixed to 10 for all datasets. Adam optimizer with the initial learning rate of 0.0001 is used for optimization of all datasets. All methods use the same

dataset partition when conducting experiments, while the locations of view missing and label absence are recorded and kept consistent.

Table 4: Detailed information of comparison methods. ✓ represent the method is able to handle the corresponding problem.

| Method | Source | Year | Multi-label | Multi-view | Missing-view | Missing-label |
|--------|--------|------|-------------|------------|--------------|---------------|
| iMVWL | IJCAI | 2018 | ✓ | ✓ | ✓ | ✓ |
| DM2L | PR | 2021 | ✓ | ✗ | ✗ | ✓ |
| LVLS | TMM | 2022 | ✓ | ✓ | ✗ | ✗ |
| DICNet | AAAI | 2023 | ✓ | ✓ | ✓ | ✓ |
| DIMC | TNNLS | 2023 | ✓ | ✓ | ✓ | ✓ |
| LMVCAT | AAAI | 2023 | ✓ | ✓ | ✓ | ✓ |
| AIMNet | AAAI | 2024 | ✓ | ✓ | ✓ | ✓ |
| MTD | NeurIPS | 2024 | ✓ | ✓ | ✓ | ✓ |
| SIP | ICML | 2024 | ✓ | ✓ | ✓ | ✓ |

## C.2  Experiment Result

Table 5: Experimental results of nine methods on the six datasets with 90% PER and 90% LMR. 'Ave.R' refers to the mean ranking of the corresponding method across all six metrics.

| DATA | METRIC | AIMNet | DICNet | DIMC | DM2L | iMVWL | LMVCAT | LVSL | MTD | SIP | TDLSR |
|------|--------|--------|--------|------|------|-------|--------|------|-----|-----|-------|
| COR | 1-HL | $0.987_{0.000}$ | $0.987_{0.000}$ | $0.987_{0.000}$ | $0.987_{0.000}$ | $0.976_{0.000}$ | $0.987_{0.000}$ | $0.987_{0.000}$ | $0.987_{0.000}$ | $0.986_{0.000}$ | $0.987_{0.000}$ |
| | 1-OE | $0.277_{0.007}$ | $0.242_{0.006}$ | $0.239_{0.009}$ | $0.171_{0.010}$ | $0.181_{0.005}$ | $0.229_{0.008}$ | $0.247_{0.006}$ | $0.274_{0.012}$ | $0.289_{0.014}$ | $0.374_{0.010}$ |
| | 1-Cov | $0.605_{0.003}$ | $0.515_{0.009}$ | $0.518_{0.005}$ | $0.481_{0.013}$ | $0.524_{0.004}$ | $0.600_{0.004}$ | $0.608_{0.002}$ | $0.573_{0.005}$ | $0.601_{0.006}$ | $0.692_{0.004}$ |
| | 1-RL | $0.823_{0.001}$ | $0.774_{0.003}$ | $0.772_{0.005}$ | $0.750_{0.005}$ | $0.762_{0.004}$ | $0.817_{0.003}$ | $0.823_{0.001}$ | $0.809_{0.001}$ | $0.821_{0.001}$ | $0.866_{0.002}$ |
| | AP | $0.240_{0.002}$ | $0.208_{0.004}$ | $0.206_{0.004}$ | $0.181_{0.002}$ | $0.163_{0.004}$ | $0.214_{0.001}$ | $0.228_{0.001}$ | $0.234_{0.005}$ | $0.242_{0.002}$ | $0.323_{0.004}$ |
| | AUC | $0.826_{0.001}$ | $0.776_{0.004}$ | $0.774_{0.005}$ | $0.754_{0.006}$ | $0.766_{0.004}$ | $0.820_{0.003}$ | $0.827_{0.001}$ | $0.811_{0.002}$ | $0.823_{0.001}$ | $0.869_{0.002}$ |
| | AVE | 3.167 | 6.833 | 7.333 | 8.333 | 9 | 5 | 3 | 5.667 | 4.167 | 1 |
| ESP | 1-HL | $0.982_{0.000}$ | $0.982_{0.000}$ | $0.982_{0.000}$ | $0.983_{0.000}$ | $0.969_{0.000}$ | $0.982_{0.000}$ | $0.983_{0.000}$ | $0.982_{0.000}$ | $0.982_{0.000}$ | $0.982_{0.000}$ |
| | 1-OE | $0.310_{0.007}$ | $0.289_{0.012}$ | $0.283_{0.006}$ | $0.210_{0.001}$ | $0.204_{0.009}$ | $0.266_{0.023}$ | $0.265_{0.004}$ | $0.302_{0.008}$ | $0.327_{0.008}$ | $0.385_{0.010}$ |
| | 1-Cov | $0.508_{0.003}$ | $0.464_{0.000}$ | $0.456_{0.004}$ | $0.447_{0.003}$ | $0.421_{0.007}$ | $0.468_{0.003}$ | $0.489_{0.001}$ | $0.492_{0.003}$ | $0.500_{0.004}$ | $0.576_{0.002}$ |
| | 1-RL | $0.792_{0.001}$ | $0.773_{0.001}$ | $0.769_{0.002}$ | $0.758_{0.002}$ | $0.729_{0.004}$ | $0.771_{0.001}$ | $0.783_{0.001}$ | $0.786_{0.001}$ | $0.785_{0.001}$ | $0.825_{0.001}$ |
| | AP | $0.222_{0.003}$ | $0.210_{0.002}$ | $0.207_{0.003}$ | $0.172_{0.002}$ | $0.155_{0.004}$ | $0.201_{0.006}$ | $0.204_{0.001}$ | $0.219_{0.002}$ | $0.225_{0.002}$ | $0.271_{0.003}$ |
| | AUC | $0.797_{0.001}$ | $0.777_{0.000}$ | $0.772_{0.002}$ | $0.762_{0.002}$ | $0.733_{0.005}$ | $0.775_{0.002}$ | $0.787_{0.000}$ | $0.790_{0.000}$ | $0.790_{0.002}$ | $0.830_{0.001}$ |
| | AVE | 3.333 | 5.333 | 6.5 | 7.667 | 10 | 6.333 | 5.167 | 4.167 | 4 | 1.333 |
| IAP | 1-HL | $0.980_{0.000}$ | $0.980_{0.000}$ | $0.980_{0.000}$ | $0.980_{0.000}$ | $0.966_{0.000}$ | $0.980_{0.000}$ | $0.980_{0.000}$ | $0.980_{0.000}$ | $0.980_{0.000}$ | $0.980_{0.000}$ |
| | 1-OE | $0.342_{0.004}$ | $0.333_{0.009}$ | $0.318_{0.002}$ | $0.247_{0.004}$ | $0.245_{0.011}$ | $0.290_{0.007}$ | $0.294_{0.004}$ | $0.344_{0.007}$ | $0.355_{0.005}$ | $0.397_{0.009}$ |
| | 1-Cov | $0.521_{0.002}$ | $0.472_{0.004}$ | $0.468_{0.002}$ | $0.443_{0.005}$ | $0.438_{0.009}$ | $0.471_{0.005}$ | $0.496_{0.002}$ | $0.510_{0.006}$ | $0.519_{0.003}$ | $0.616_{0.001}$ |
| | 1-RL | $0.818_{0.003}$ | $0.799_{0.004}$ | $0.795_{0.002}$ | $0.780_{0.003}$ | $0.761_{0.005}$ | $0.793_{0.002}$ | $0.807_{0.001}$ | $0.816_{0.003}$ | $0.817_{0.003}$ | $0.860_{0.001}$ |
| | AP | $0.229_{0.002}$ | $0.222_{0.004}$ | $0.215_{0.002}$ | $0.186_{0.004}$ | $0.167_{0.004}$ | $0.202_{0.003}$ | $0.208_{0.001}$ | $0.232_{0.002}$ | $0.235_{0.002}$ | $0.278_{0.003}$ |
| | AUC | $0.822_{0.002}$ | $0.801_{0.003}$ | $0.798_{0.002}$ | $0.783_{0.003}$ | $0.766_{0.005}$ | $0.797_{0.001}$ | $0.811_{0.001}$ | $0.819_{0.002}$ | $0.820_{0.002}$ | $0.861_{0.001}$ |
| | AVE | 2.5 | 4.833 | 5.833 | 7.667 | 10 | 6.667 | 5 | 3.167 | 2.333 | 1 |
| MIR | 1-HL | $0.875_{0.001}$ | $0.879_{0.001}$ | $0.877_{0.002}$ | $0.876_{0.000}$ | $0.827_{0.004}$ | $0.865_{0.004}$ | $0.874_{0.000}$ | $0.880_{0.002}$ | $0.875_{0.002}$ | $0.885_{0.002}$ |
| | 1-OE | $0.506_{0.023}$ | $0.533_{0.005}$ | $0.511_{0.005}$ | $0.439_{0.005}$ | $0.406_{0.023}$ | $0.470_{0.020}$ | $0.485_{0.006}$ | $0.535_{0.008}$ | $0.540_{0.009}$ | $0.612_{0.006}$ |
| | 1-Cov | $0.598_{0.006}$ | $0.594_{0.003}$ | $0.589_{0.001}$ | $0.572_{0.003}$ | $0.530_{0.012}$ | $0.581_{0.002}$ | $0.584_{0.001}$ | $0.606_{0.006}$ | $0.604_{0.006}$ | $0.645_{0.005}$ |
| | 1-RL | $0.827_{0.006}$ | $0.828_{0.001}$ | $0.823_{0.001}$ | $0.809_{0.002}$ | $0.765_{0.011}$ | $0.817_{0.003}$ | $0.819_{0.001}$ | $0.834_{0.003}$ | $0.830_{0.002}$ | $0.861_{0.003}$ |
| | AP | $0.494_{0.017}$ | $0.512_{0.001}$ | $0.501_{0.002}$ | $0.467_{0.004}$ | $0.415_{0.009}$ | $0.485_{0.010}$ | $0.482_{0.002}$ | $0.519_{0.006}$ | $0.519_{0.002}$ | $0.575_{0.002}$ |
| | AUC | $0.820_{0.003}$ | $0.823_{0.001}$ | $0.818_{0.001}$ | $0.805_{0.000}$ | $0.769_{0.007}$ | $0.808_{0.004}$ | $0.816_{0.002}$ | $0.827_{0.002}$ | $0.823_{0.001}$ | $0.852_{0.000}$ |
| | AVE | 5.333 | 4 | 5.333 | 8.333 | 10 | 8 | 7.333 | 2.167 | 3.333 | 1 |
| OBJ | 1-HL | $0.937_{0.001}$ | $0.938_{0.001}$ | $0.938_{0.000}$ | $0.935_{0.000}$ | $0.882_{0.005}$ | $0.927_{0.002}$ | $0.934_{0.000}$ | $0.938_{0.000}$ | $0.937_{0.001}$ | $0.946_{0.001}$ |
| | 1-OE | $0.468_{0.005}$ | $0.453_{0.009}$ | $0.439_{0.005}$ | $0.415_{0.009}$ | $0.335_{0.036}$ | $0.405_{0.019}$ | $0.364_{0.004}$ | $0.474_{0.004}$ | $0.485_{0.013}$ | $0.603_{0.020}$ |
| | 1-Cov | $0.727_{0.009}$ | $0.720_{0.006}$ | $0.709_{0.010}$ | $0.682_{0.002}$ | $0.657_{0.023}$ | $0.705_{0.017}$ | $0.712_{0.002}$ | $0.740_{0.007}$ | $0.727_{0.006}$ | $0.795_{0.003}$ |
| | 1-RL | $0.829_{0.003}$ | $0.823_{0.006}$ | $0.814_{0.008}$ | $0.800_{0.004}$ | $0.768_{0.012}$ | $0.806_{0.011}$ | $0.811_{0.001}$ | $0.835_{0.003}$ | $0.828_{0.004}$ | $0.881_{0.002}$ |
| | AP | $0.506_{0.010}$ | $0.502_{0.006}$ | $0.489_{0.010}$ | $0.470_{0.002}$ | $0.394_{0.026}$ | $0.476_{0.009}$ | $0.446_{0.001}$ | $0.519_{0.008}$ | $0.522_{0.009}$ | $0.624_{0.011}$ |
| | AUC | $0.842_{0.003}$ | $0.836_{0.005}$ | $0.828_{0.008}$ | $0.814_{0.000}$ | $0.784_{0.012}$ | $0.821_{0.011}$ | $0.827_{0.001}$ | $0.848_{0.003}$ | $0.840_{0.004}$ | $0.891_{0.001}$ |
| | AVE | 3.667 | 4.667 | 5.833 | 8.167 | 10 | 8 | 7.667 | 2.333 | 3.667 | 1 |
| PAS | 1-HL | $0.923_{0.003}$ | $0.927_{0.001}$ | $0.927_{0.001}$ | $0.926_{0.000}$ | $0.871_{0.003}$ | $0.921_{0.003}$ | $0.926_{0.000}$ | $0.926_{0.001}$ | $0.923_{0.003}$ | $0.927_{0.000}$ |
| | 1-OE | $0.382_{0.023}$ | $0.402_{0.002}$ | $0.403_{0.002}$ | $0.380_{0.007}$ | $0.306_{0.037}$ | $0.376_{0.021}$ | $0.415_{0.001}$ | $0.395_{0.011}$ | $0.389_{0.011}$ | $0.415_{0.016}$ |
| | 1-Cov | $0.658_{0.002}$ | $0.636_{0.013}$ | $0.626_{0.013}$ | $0.636_{0.009}$ | $0.654_{0.014}$ | $0.630_{0.018}$ | $0.654_{0.005}$ | $0.674_{0.006}$ | $0.668_{0.018}$ | $0.753_{0.006}$ |
| | 1-RL | $0.727_{0.006}$ | $0.710_{0.011}$ | $0.703_{0.010}$ | $0.698_{0.008}$ | $0.658_{0.014}$ | $0.693_{0.016}$ | $0.726_{0.004}$ | $0.740_{0.003}$ | $0.729_{0.013}$ | $0.808_{0.007}$ |
| | AP | $0.440_{0.009}$ | $0.440_{0.005}$ | $0.434_{0.004}$ | $0.421_{0.006}$ | $0.368_{0.021}$ | $0.430_{0.005}$ | $0.444_{0.002}$ | $0.454_{0.007}$ | $0.449_{0.006}$ | $0.504_{0.009}$ |
| | AUC | $0.754_{0.003}$ | $0.737_{0.011}$ | $0.727_{0.012}$ | $0.733_{0.007}$ | $0.690_{0.013}$ | $0.727_{0.010}$ | $0.754_{0.003}$ | $0.766_{0.006}$ | $0.761_{0.013}$ | $0.831_{0.006}$ |
| | AVE | 5.333 | 5.167 | 6 | 7.167 | 10 | 8.5 | 4.167 | 3.167 | 4.167 | 1 |

**Comparison Experiment.** To validate that our method can adapt to varying degrees of data missing, we conduct comparison experiments with PER and LMR ranging from $\{30\%, 50\%, 70\%, 90\%\}$. Table 5 presents the comparison results and algorithm rankings when PER and LMR are fixed at 90%. Fig. 8, 9 and 10 illustrate the distributional trends of all metrics as PER increases from 30% to 70%, with LER fixed at 30%, 50%, and 70%, respectively. It can be observed that our method

outperforms the other nine methods in almost all cases, which demonstrates the robustness of our TDLSR in handling incomplete multi-view multi-label problems. Moreover, the effectiveness of our method is especially evident when dealing with high levels of data unavailability. For instance, as shown in Fig. 8, on Corel 5k, the performance margin of our method over the second-best gradually widens. Additionally, as reported in Table 5, our method achieves more than a 10% improvement in the most representative metric AP on both OBJECT and Pascal07.

**Parameter Determination and Convergence.** The parameters $\lambda_1$ and $\lambda_2$ are used to balance the effects of $\mathcal{L}_{IB}$ and $\mathcal{L}_{le}$. Two parameters are selected from the range of $\{0.01, 0.1, 1, 10, 100, 1000\}$ and the joint influence are presented in the heatmap as shown in Fig. 6. From the result, the performance of TDLSR exhibits variability under different parameter setting. Besides, the optimal results are typically achieved when $\lambda_1$ falls within the range of (0.01, 1). Overall, the parameter sensitivity is relatively low on certain datasets, such as on ESPgame and OBJECT. We also present the simultaneous evolution of training loss and performance metric on the validation set throughout the training process in Fig. 7. The results show that our TDLSR demonstrates great convergence and gradually approaches the optimal network parameters.

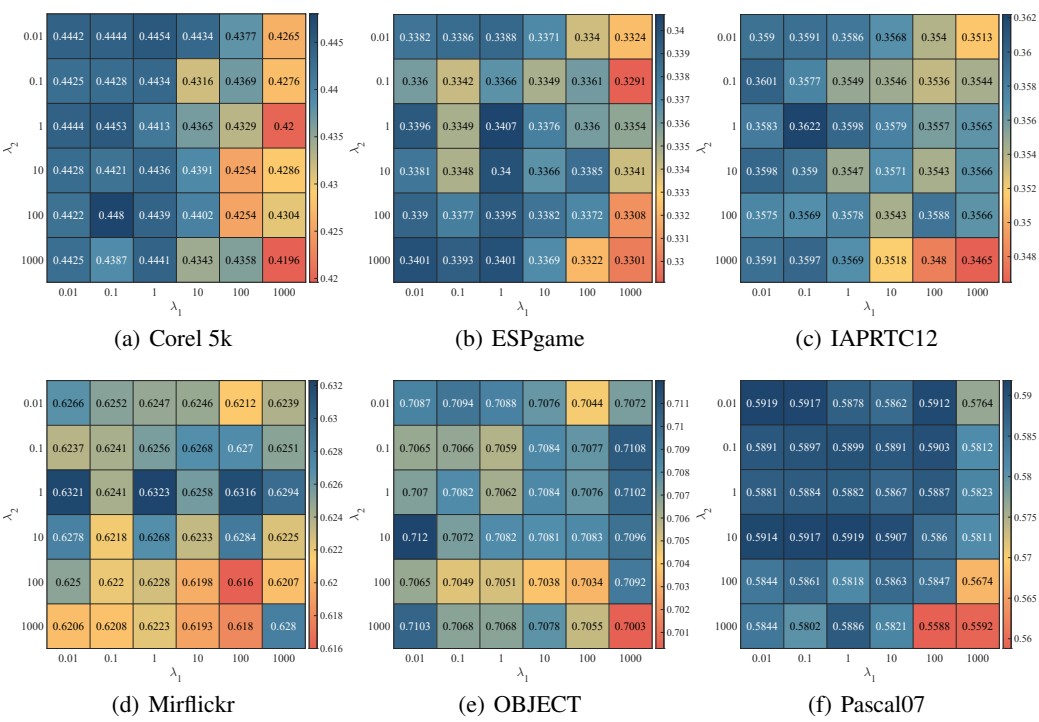

Figure 6: Parameter analysis of the trade-off parameters $\lambda_1$ and $\lambda_2$.

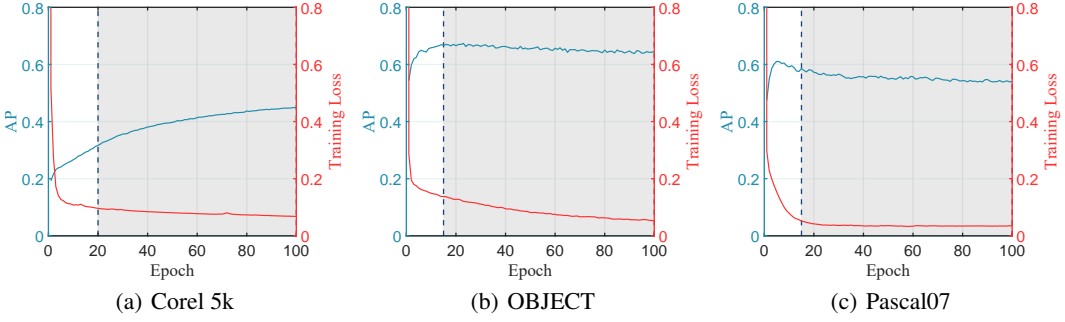

Figure 7: The convergence behavior of our TDLSR during training.

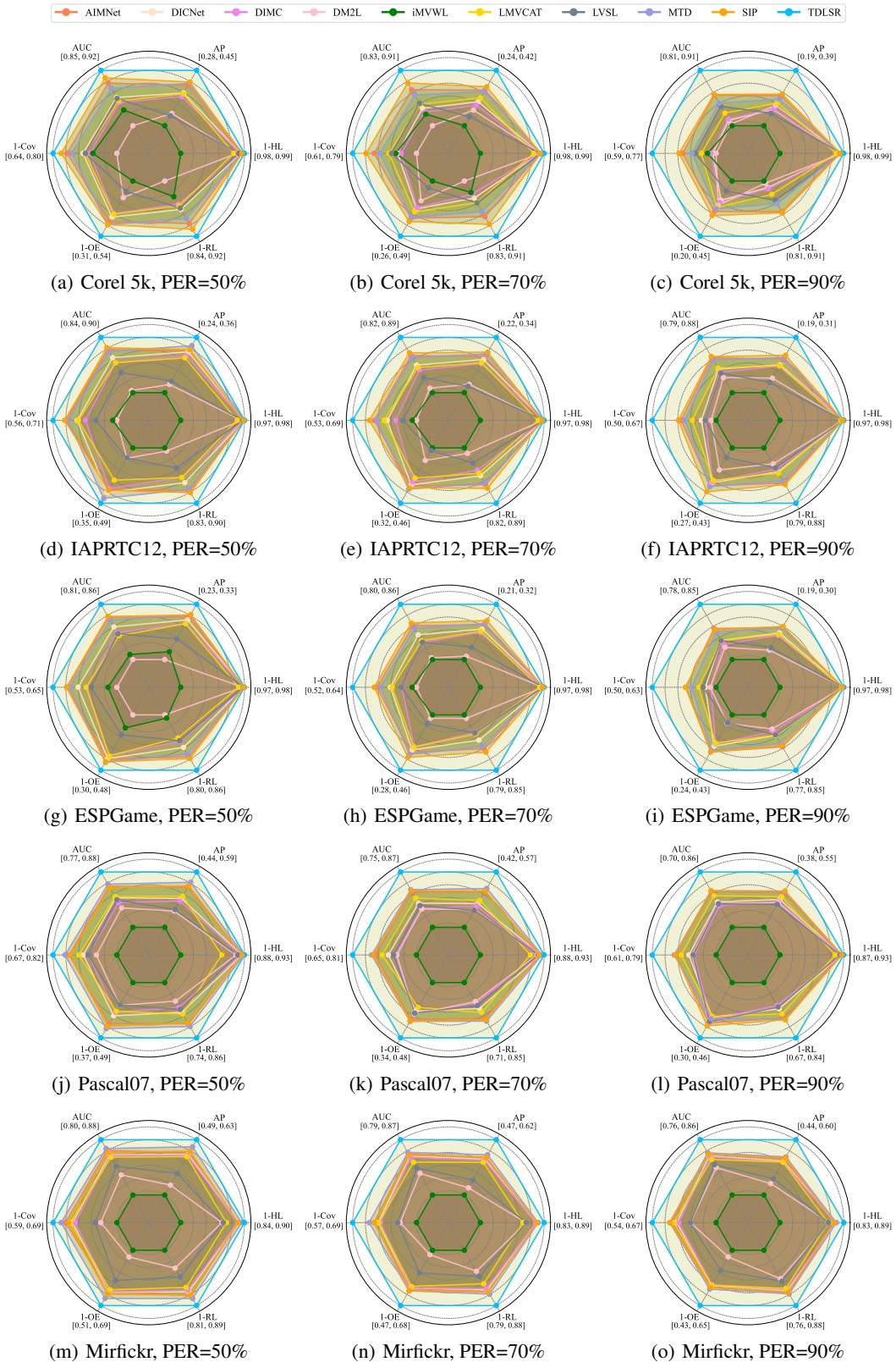

Figure 8: Experimental results of ten methods on five datasets with PER varying from 50% to 90% while LMR= 50%.

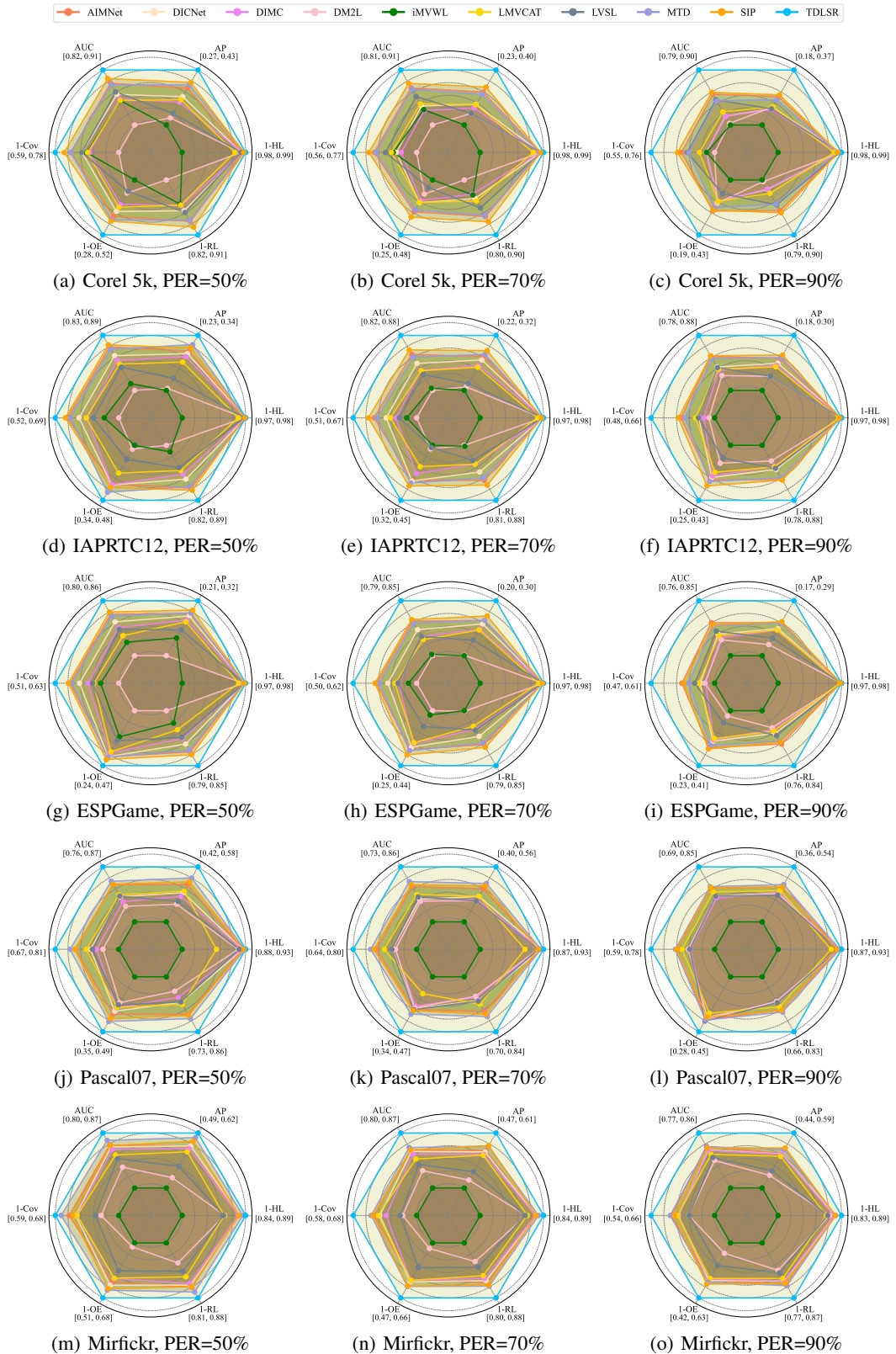

Figure 9: Experimental results of ten methods on five datasets with PER varying from 50% to 90% while LMR= 70%.

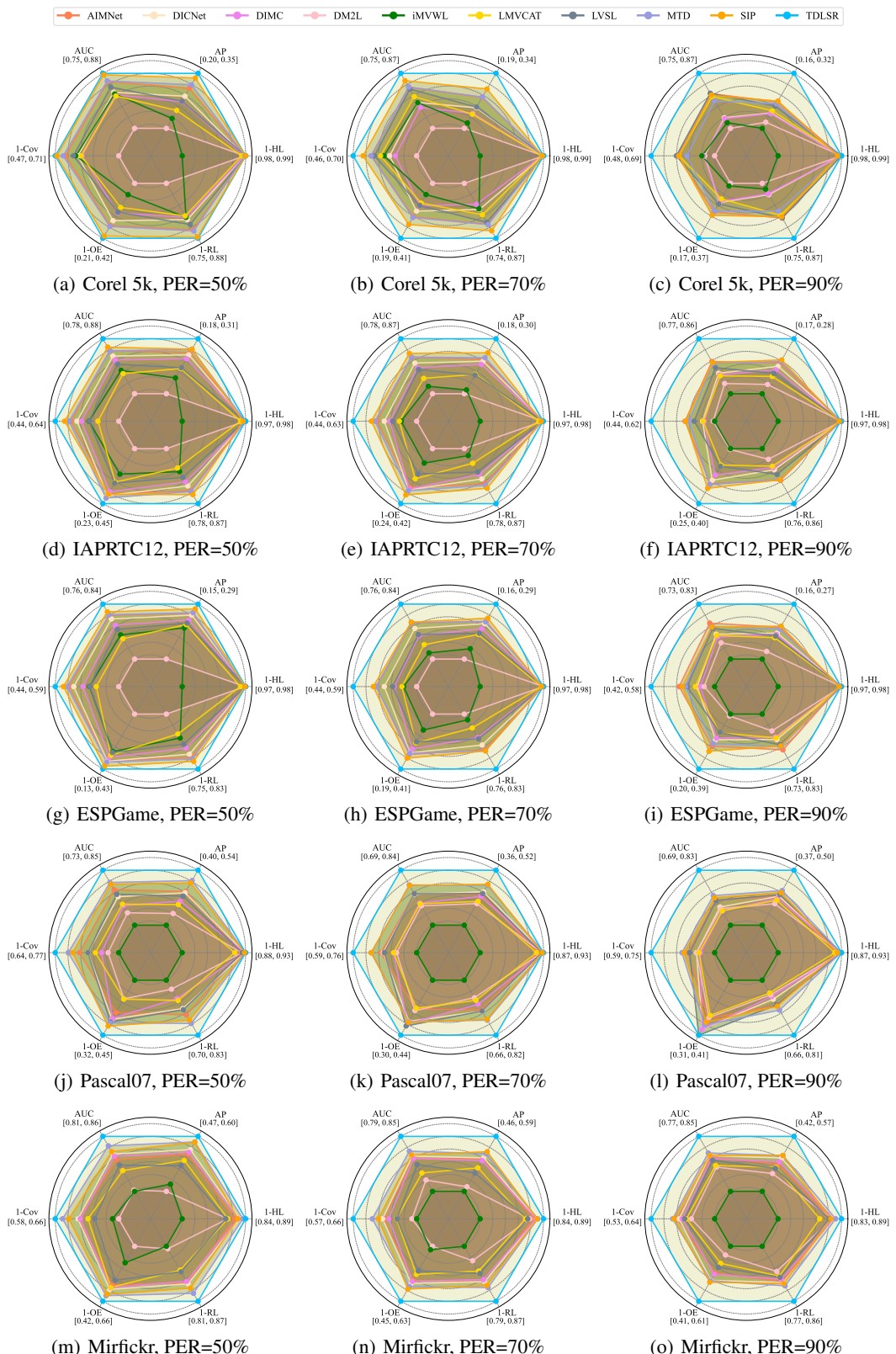

Figure 10: Experimental results of ten methods on five datasets with PER varying from 50% to 90% while LMR= 90%.

