# OpenReview forum: "Theory-Driven Label-Specific Representation for Incomplete Multi-View Multi-Label Learning"
_NeurIPS.cc/2025/Conference — NeurIPS 2025 spotlight_

### Official Review · Reviewer_P58t · 2025-06-27

**Clarity:** 3
**Significance:** 3
**Originality:** 4
**Rating:** 5
**Confidence:** 4

**Summary:**

This paper presents a theory-driven framework named TDLSR to address the problem of incomplete multi-view multi-label learning. The proposed method introduces a proximity-aware graph attention mechanism for missing view recovery, optimizes mutual information to separate shared and specific features, and models label prototypes to capture label correlation semantics. Theoretical contributions include discriminability analysis and generalization error bounds. Experiments across six public datasets and a real-world NBA scenario demonstrate significant advantages over state-of-the-art methods.

**Questions:**

(1) Unlike graph neural networks, the proposed view restoration strategy does not involve network parameter fine-tuning. Intuitively, parameter optimization is expected to reduce errors. The authors are requested to explain why the absence of fine-tuning leads to better performance.

(2) Given that mutual information has been employed in prior multi-view feature extraction methods, please clarify the main distinguishing contribution of this work. Furthermore, are all three proposed principles indispensable?

(3) Label-specific representation learning is a common approach in multi-label classification. How is it specifically implemented in this work? Furthermore, how does the proposed method integrate label semantics learning and label-specific representation learning in a unified manner?

**Ethical Concerns:**

["NO or VERY MINOR ethics concerns only"]

**Final Justification:**

I have read all the reviews and the authors' response. All my questions have been addressed. I keep the ratings unchanged.

**Limitations:**

Yes, the other aspects have all been mentioned under the weaknesses and questions.

**Paper Formatting Concerns:**

N

**Quality:**

3

**Strengths And Weaknesses:**

Strengths:

(1) The method is forward-looking, with its innovative design built upon the latest SOTA, such as MTD and SIP.
(2) The paper emphasizes the integration of method and theory, providing theoretical justifications from multiple perspectives, including discriminability, effectiveness, and generalization error.
(3) This work involves substantial effort and thoroughly validates the effectiveness of the proposed method under a wide range of missing data ratios. Additionally, a real-world NBA dataset is constructed via web crawling to conduct comparative experiments.

Weaknesses：

(1)This paper lacks an algorithmic flowchart. Please provide a detailed explanation of the optimization process, parameter selection, and other relevant implementation details to facilitate reproducibility.
(2)Figures 1 and 2 lack detailed captions. Please provide comprehensive explanations to help readers understand. In addition, please clarify the intended purpose of these two figures, as they occupy substantial space in the paper.
(3)As is well known, a general theoretical framework for generalization error in multi-view learning has yet to be well established. Therefore, please elaborate on the technology used to derive the generalization error bound of NAMLL, and explain how the correctness of the proposed theory is validated.

---

> ### Author Rebuttal · Authors · 2025-07-31
>
> Thank you for your careful review and insightful comments on my manuscript. Your recognition of the work’s pioneering contributions, its seamless integration of theoretical foundations with practical implementation, and the substantial scope of the experimental efforts is deeply encouraging and serves as a great motivation for our continued research. Meanwhile, we also sincerely accept the shortcomings you have pointed out and will revise the manuscript carefully in accordance with your suggestions. Next, we provide a detailed response to the concerns you have raised regarding our work.
>  ### **Weakness1: The detailed algorithmic implementation**
> ● Algorithmic procedure. Our method first reconstructs the missing views using Eq. (2) and (3), which results in a new multi-view data denoted as $\\{{Z}^{(v)}\\}\_{v=1}^{V}$. This data $\\{{Z}^{(v)}\\}\_{v=1}^{V}$ is then fed into dual-channel encoders to obtain the shared representations $\\{{S}^{(v)}\\}\_{v=1}^{V}$ and  private representations $\\{{O}^{(v)}\\}\_{v=1}^{V}$. Simultaneously, a label prototype matrix H is sampled and processed through a GIN layer to produce the matrix $E$. Subsequently, element-wise interactions are performed between the label prototypes and each representation, followed by a confidence-based fusion process guided by the reconstruction quality, leading to the final prediction matrix $\overline{{P}}$. Three types of losses are computed, i.e.,  $\mathcal{L}\_{IB}$ for feature extraction,  $\mathcal{L}\_{le}$ for label prototype learning, and $\mathcal{L}\_{bce}$ for classification. These are summed to form the total loss $\mathcal{L}=\mathcal{L}\_{bce}+\lambda\_{1} \mathcal{L}\_{IB}+\lambda\_{2}\mathcal{L}\_{le}$. The network parameters are then updated batch-wise using the Adam optimizer with a learning rate of 0.0001. After training for the predefined number of epochs, predictions are performed on the test set, and classification metrics are calculated accordingly.
>
> ● Parameter Setting. Except for the balanced coefficients $\lambda_{1}$ and $\lambda_{2}$, which are tuned within the range of $\\{0.01,0.1,1,10,100,1000\\}$, all other parameters are kept fixed. The temperature parameter $\tau$ is fixed to 0.2, the balanced factor $\alpha$ is fixed to 10, the number of nearest neighbors $k$ for each sample takes the value of 10,  the Lagrange multiplier
> $\beta$ is 1, the sampling number $s$ is 5.
>  ### **Weakness2: Descriptions of Figures 1 and 2**
> ● Figure 1 presents the overall framework of our proposed algorithm, which consists of three components, i.e., "Proximity-aware Graph Attention Recovery", "Universal View Extraction Framework", and "Multi-Label Semantic and Label-Specific Representation Learning". The first component describes the reconstruction of missing views by propagating cross-view neighborhood information, guided by attention-based similarity mechanisms. The second component focuses on extracting both shared and view-specific representations, with core information enhanced and feature disentangled based on three principles, i.e., information shift, interaction, and orthogonality. The third component involves sampling initial prototype representations, followed by refinement through a GIN layer, where label correlations are used as edge weights. Subsequently, the refined label prototypes interact with each representation, and a confidence-aware late fusion is applied to produce the final multi-label classification results.
>
> ● Figure 2 provides  a visualization of the proposed information-theoretic modeling based on three guiding principles. The top-left part illustrates the original modalities, where the red region represents shared information, while the blue and green regions denote modality-specific private information. The bottom-left section explicitly shows the disentangled shared and private information. The right side demonstrates the roles of the three principles, i.e., information shift facilitates the extraction of core informative components from the original data, information interaction enhances cross-modal sharing of common representations while promoting separation of private features, information orthogonality encourages the shared and private representations extracted from the same view to be mutually non-redundant.
>
> Therefore, these figures are intended to help readers better understand the overall approach and the key innovations introduced in this work.
>
>  ### **Weakness3: How to derive the generalization error bound**
> The process of deriving the generalization error bound consists of five main stages, as outlined below.
>
> ● Definition of risk functions. The expected risk  $R({f})$ and empirical risk $\widehat{R}_{D}({f})$ are formally defined based on the multi-label classification loss over multi-view inputs, with a late fusion strategy applied using view-specific reconstruction quality scores $Q$ as fusion weights.
>
> ● Modeling of function class. The hypothesis space $\mathcal{F}$ is constructed to represent vector-valued functions mapping multi-view inputs to prediciton scores.  The function includes shared and specific feature extractors $\left(\phi\^{s}, \phi\^{o}\right)$, label-specific mappings $\zeta$, and classification heads $w$
>
>  ● Information-theoretic analysis. By defining ${R}\_{{Y}^{c}}$ as the informative representations with respect to labels $Y^c$, we introduce the concept of typical subsets $\mathcal{R}_{c,\gamma}^{x}$ and derive probabilistic bounds  from information entropy using McDiarmid's inequality.
>
>  ● Contrastive sampling and risk decomposition. The risk is decomposed across instance-view-label triplets. By selecting reliable samples as representatives of typical behavior, and the rest as outliers, the bounds for both regions are estimated.
>
>  ● Bounding the individual terms. By leveraging the concentration inequality and Jensen’s inequality, we estimate the upper bound on the difference between the expected risk and the empirical risk term by term.
>
> Since the derived generalization error bound becomes increasingly tighter as the representation disentanglement progresses, it indirectly supports the validity of our theoretical conclusions. A better separation between shared and specific representations is particularly beneficial for multi-label classification, as the discriminative features for each label may originate from either the shared information or view-specific components.
>  ### **Question1: Explain for non-parametric view reconstruction**
> If network parameters are introduced into the view reconstruction process, they are likely to be poorly trained in the early stages, resulting in inaccurate or low-quality reconstructed samples. These erroneous reconstructions may mislead the learning of the subsequent feature extractors and lead to a form of negative interplay between the reconstruction module and the representation learning process.  As a consequence, the joint optimization may be steered in an undesirable direction or converge more slowly. Thus, employing a non-parametric view reconstruction mechanism not only reduces computational overhead, but also yields more reliable reconstructed samples in the early stages by leveraging the intrinsic structure of the original views. This approach effectively ensures the stability of subsequent network updates and enhances the robustness of the model.
>  ### **Question2: Unique advantage of mutual information-guided representation learning**
> Our unique contribution lies in proposing a general framework for multi-view representation learning under mutual information constraints. This framework is built upon three principles, i.e., information shift, interaction, and orthogonality. Specifically, the proposed information-theoretic framework for is grounded in three core aspects, i.e., the alignment with original views, the cross-view correlations, and the interactions between  intra-view representations. Moreover,  we theoretically establish the effectiveness of our framework for representation extraction, i.e., to maximize the mutual information between the learned representations and the final labels, the representations must correspond to the optimal solution defined within this framework. This highlights the necessity of all three principles, as removing any one leads to reduced mutual information between representations and labels.
>  ### **Question3: Design for label semantic learning and label-specific representation learning**
> For label semantic learning, we first sample the prototype representation for each class. These prototypes are then refined through a GIN layer, where the edge weights are determined by quantitatively measured label correlations. Thus, we can obtain label prototypes that encapsulate inter-label semantic relationships. Next, we perform element-wise interactions between each label prototype and both the shared and specific representations, which results in label-specific shared and specific representations. These label-specific representations are then passed through a fully connected layer, followed by late-stage fusion to produce the final soft labels. This process facilitates the extraction of the most discriminative features for each label prototype from both the shared and specific feature pool, while also refining the label preference across different representations.

---

> > ### Comment · Reviewer_P58t · 2025-08-06
> >
> > Thank you for your response. All my concerns have been addressed and I will keep the scores.

---

> > > ### Author Response · Authors · 2025-08-07
> > > **Thank you!**
> > >
> > > We are deeply grateful for your careful consideration of our rebuttal. It is encouraging to know that our clarifications have resolved your concerns, and we truly value your constructive feedback and endorsement. We will revise the manuscript  to ensure that all discussed points are appropriately reflected.

---

### Official Review · Reviewer_NS9F · 2025-06-28

**Clarity:** 4
**Significance:** 4
**Originality:** 3
**Rating:** 5
**Confidence:** 3

**Summary:**

To address the challenges of view and label incompleteness caused by feature storage limitations and annotation costs, this paper proposes the TDLSR framework. It integrates a proximity-aware graph recovery mechanism and mutual information-constrained feature disentanglement to balance cross-view complementarity and consistency. Moreover, label semantics-guided representation is employed to model inter-label correlations.

**Questions:**

1.In Theorem 2, it is assumed that the information entropy of both shared and private representations is fixed. Does this assumption hold in practice for multimodal or multi-view data? Given the heterogeneous nature of views, large entropy discrepancies might arise and compromise the discriminability of private representations.

2.Constructing and propagating view-specific graphs across all views can be computationally expensive. The paper lacks a discussion or analysis on the training cost or scalability of the proposed method.

3.How is mutual information estimated in Eq. (4)? Additionally, how is the Lagrange multiplier in Eq. (7) set or tuned during training?

**Ethical Concerns:**

["NO or VERY MINOR ethics concerns only"]

**Final Justification:**

The authors’ response addressed all of my questions, and I agree with their explanation. They described the adjustments made to each technical component. They also provided the reason for using fixed entropy, the mutual information estimation process, and the running times. Although TDLSR involves a more complex computation process, this is generally in line with expectations. I will maintain my original score.

**Limitations:**

Yes

**Quality:**

4

**Strengths And Weaknesses:**

Strengths:

1.The paper presents extensive experiments demonstrating the superior performance of the TDLSR framework.

2.It is well-motivated by addressing both view and label incompleteness, a limitation shared by many existing approaches.

3.The theoretical derivation, especially on representation disentanglement and generalization bounds, is clear and well-articulated.

Weaknesses:

1.Many technical components (e.g., MI constraints, graph-based imputation, and prototype learning) have been widely adopted in prior work. The novelty of this work lies more in its integration and theoretical refinement rather than introducing fundamentally new paradigms.

---

> ### Author Rebuttal · Authors · 2025-07-30
>
> I am grateful for your careful review of my paper and your valuable comments.  We are deeply motivated by your acknowledgment of the strengths of our work, particularly the breadth of experiments, the methodological advances, and the clarity of the theoretical analysis.  At the same time,  we sincerely appreciate your identification of the weaknesses and issues in our work, which offers significant insights for its further improvement. Next, we will spare no effort to address your concerns and questions,  and carefully revise the manuscript and pursue additional research to resolve its deficiencies.
>  ### **Weakness1: Advancing without new paradigms**
> Our approach is grounded in three technically mature components, which are widely adopted in multi-view learning and multi-label classification, i.e., the graph attention recovery mechanism, mutual information constrained feature extraction, and prototype representation learning.  Although these techniques have been previously utilized, our method incorporates distinct adaptations in each technical framework to effectively accommodate the complexities of incomplete multi-view and multi-label learning scenarios.
> ● Firstly, regarding view recovery, previous methods such as AIMNet leverages all available samples to reconstruct the missing ones. However, not all samples exhibit strong relevance. DIMvSML employs graph neural networks to propagate cross-view structural dependencies for recovery, but it introduces a large number of network parameters to model edge weights on the graph, leading to increased computational cost. Besides, these recovery-related parameters, when insufficiently trained in the early stage, can negatively impact the parameter updates of subsequent feature extractors. To address this, we construct instance-level relation graphs by leveraging attention computation to drive neighborhood-aware selection. Without introducing additional network parameters, we perform cross-view propagation of highly correlated sample information on the relation graph to support view recovery.
>
> ● Secondly,  for representation extraction, many traditional methods have employed mutual information based technique to capture informative content and characterize view correlations. Currently, existing methods in the multi-view domain lack a unified framework or consensus on how to  employ mutual information constrained loss functions for feature extraction, and they are also unsupported by solid theoretical foundations. Besides, multi-view multi-label learning methods such as SIP focuses solely on extracting shared representations while neglecting the view-specific contributions essential for classification. Accordingly, we systematically formulate three information-theoretic principles for multi-view feature extraction, i.e., information shift, interaction and orthogonality. The information shift term $\sum_{v=1}^{V} I({s}^{(v)};{z}^{(v)})$ is critical for guaranteeing that the representations stay consistent with the core semantics of raw views. The information interaction term $\sum_{v=1}^{V}\sum_{v^*\neq v}^{V} (I({s}^{(v^{\*})}|{z}^{(v^{\*})};{s}^{(v)}|{z}^{(v)})-I({o}^{(v^{\*})}|{z}^{(v^{\*})};{o}^{(v)}|{z}^{(v)}))$ can promote information exchange among shared components while mitigating overlap among view-specific components. The information orthogonality term $\sum_{v=1}^{V} I({s}^{(v)}|{z}^{(v)};{o}^{(v)}|{z}^{(v)})$ is used to ensure that the shared and specific features extracted from each view are mutually independent and completely disentangled.  Therefore, we introduce a comprehensive information-theoretic framework for representation learning grounded in three core aspects, i.e., the alignment with original views, the cross-view correlations, and the interactions between intra-view representations. We theoretically show that all three principles are essential to maximize the mutual information between representations and labels. Besides, we also provide a theoretical guarantee that as representation disentanglement improves, the model's generalization ability is correspondingly enhanced.
>
> ● Thirdly,  for label prototype, some methods like SIP fail to account for label correlations during prototype extraction, even though capturing such dependencies is crucial for enhancing performance in multi-label learning. Thus, in our approach, we first sample the prototype representation for each class, and then update them through a graph network that incorporates label correlation information derived from the Jaccard distance. This yields label prototypes that embed inter-label dependencies.
>  ### **Question1:  The assumption of Theorem 2 is impractical**
> **Required for the proof:** The fixed entropy assumption in Theorem 2 facilitates a clear analysis of how mutual information objectives influence the association  between learned representations and label semantics, which serves as a necessary simplification for theoretical modeling.
>
> **The Theorem  remains valid under relaxed assumptions:** If the assumption is relaxed such that the information entropy of shared representations differs from that of specific representations  $(H({S}^{(v)})=H^{0}, H({O}^{(v)})=H^{1} (1 \leq {v} \leq {V}))$, the validity of Theorem 2 remains unaffected as long as the entropy values are constant.  This is because our proof is based on the relative magnitude between the information  entropy of the representations and that of the labels, rather than relying on the equality between the information entropy of shared and specific components.
>
> **Practical deployment and empirical evidence:** The model does not impose hard constraints on the entropy values.  Instead, by optimizing a variational lower bound of mutual information, the model emphasizes the relative mutual information between representations rather than their absolute entropy magnitudes. As a result, the optimization objective remains valid even when entropy discrepancies exist across different views. Although entropy variation may occur in practice, we mitigate its impact by extracting all representations with the same dimensionality, which effectively regulates the amount of information they can encode. Furthermore, ablation results show that removing the disentanglement module leads to performance degradation, directly validating the theoretical foundation and efficacy of the model.
>  ### **Question2: Method complexity**
> The complexity of our method primarily arises from three components, i.e., the construction of sample topology graph for each view, the dual-channel representation extraction, and the learning of  prototype representation for each label. Next, we present a comparison of the runtime(s) across different algorithms:
> ||AIMNet | DICNet | DIMC | LMVCAT|MTD|SIP|iMVWL|Ours|
> |-|-|-|-|-|-|-|-|-|
> | Corel5k  |408.37|765.02 | 559.77 |1198.72|2520.99|1039.93|513.41|2270.64|
> |Espgame| 607.02| 3047.17|995.89|3037.25|4080.72|1605.16|1822.2|3882.59|
> |Mirflickr|159.98|1228.27 |359.66|151.27|562.55|727.26|568.46|520.24|
>
> As indicated by the comparison, our method incurs higher complexity than most baselines due to its comprehensive treatment of view reconstruction, representation learning, and label correlation. Nevertheless, it achieves faster runtime than MTD, another dual-branch model, suggesting that our loss design contributes to efficient convergence.
>  ### **Question3: The estimation method for mutual information**
>  The mutual information terms in Eq. (4) are first reformulated via variational inference to obtain tractable bounds:
>
>  $I({s}^{(v)};{z}^{(v)}) \geq \mathbb{E}_{p\left({z}^{(v)}, {s}^{(v)}\right)}\left[\log q\left({s}^{(u)} | {z}^{(v)}\right)\right] $
>
> $I({s}^{(v^{\*})}|{z}^{(v^{\*})};{s}^{(v)}|{z}^{(v)}) \geq \int \int p({s}\^{(v)}|{z}\^{(v)};{s}\^{(v^{\*})}|{z}\^{(v^{\*})})\log q({s}\^{(v)}|{s}\^{v^{\*}}) d{s}\^{(v^{\*})} d{s}\^{(v)}$
>
> $I({o}^{(v^{\*})}|{z}^{(v^{\*})};{o}^{(v)}|{z}^{(v)}) \leq D_{KL} (p({o}^{(v^{\*})}|{z}^{(v^{\*})};{o}^{(v)}|{z}^{(v)}) \| p({o}^{(v^{\*})}|{z}^{(v^{\*})})q({o}^{(v)}|{o}^{(v^{\*})}))$
>
> $I({s}^{(v)}|{z}^{(v)};{o}^{(v)}|{z}^{(v)}) \leq D_{KL} (p({s}^{(v)}|{z}^{(v)};{o}^{(v)}|{z}^{(v)}) \| p({o}^{(v)}|{z}^{(v)})q({s}^{(v)}|{o}^{(v)}))$
>
> The detailed estimation procedures for these bounds are provided in Appendix A.2.  The core idea is to adopt a data-driven approach that leverages cross-view instance-level contrastive learning to approximate the joint distribution between views. By  aggregating along different directions of the joint distribution matrix, the marginal distributions of individual views can be derived, thereby enabling the computation of all distributions involved in the variational bounds. The Lagrange multiplier in Eq. (7) is fixed to 1 in our experiments.

---

> > ### Comment · Reviewer_NS9F · 2025-08-06
> >
> > Thank you for your response. My concerns have been adequately addressed. I will maintain my original score.

---

> > > ### Author Response · Authors · 2025-08-07
> > > **Thank you!**
> > >
> > > Thank you very much for your thorough review of our rebuttal. We are pleased that our clarifications have addressed your concerns, and we sincerely appreciate your support and recognition. We will carefully revise the manuscript to incorporate the key points discussed.

---

### Official Review · Reviewer_FGNW · 2025-06-29

**Clarity:** 3
**Significance:** 3
**Originality:** 4
**Rating:** 5
**Confidence:** 4

**Summary:**

The paper proposes TDLSR, a theory-driven label-specific representation learning framework tailored for incomplete multi-view multi-label classification (iMvMLC). The model integrates a proximity-aware graph attention mechanism, mutual information-based representation disentanglement, and label prototype modeling to effectively mitigate the challenges posed by data sparsity in both feature and label spaces.  It demonstrates superior robustness and performance across six public datasets and an real-word NBA application.

**Questions:**

(1)	A central challenge in multi-view multi-label learning lies in effective feature extraction and label correlation modeling. Please explain how this work addresses these two key difficulties.

(2)	The extraction of shared and private representations is no longer considered a novel strategy in the multi-view learning field. Please clarify whether the improvements made in this work on top of that framework are substantial, and whether they lead to significant performance gains in the experiments.

(3)	How should the notion of discriminability mentioned in the Theoretical Results section be understood? Could the authors provide a more precise definition? Additionally, why is the theoretical analysis structured from two separate aspects？

(4)	Given that many multi-view multi-label classification methods have released their code, would the authors be willing to make their implementation publicly available as well?

**Ethical Concerns:**

["NO or VERY MINOR ethics concerns only"]

**Final Justification:**

Thank you for the detailed response from the author, which has answered most of my questions. I have decided to maintain my rating.

**Limitations:**

Yes, and the relevant comments have already been provided above.

**Quality:**

4

**Strengths And Weaknesses:**

# Strengths:
(1)	Solid theoretical foundation. The mutual information formulation rigorously supports feature disentanglement, with derived theorems explaining generalization ability.

(2)	Innovative Methodology Design. For the first time, the information shift, interaction and orthogonality principle is applied to feature representation disentanglement, combined with label-specific shared and exclusive representation learning to enhance multi-label classification.

(3)	Comprehensive experiments. The work is evaluated on both public and real-world datasets with extensive metrics, proving its effectiveness and robustness.

# Weaknesses：
(1)	The paper introduces numerous symbols without a centralized notation list, making it difficult to follow the mathematical formulations. The clarity of the writing could be further improved.

(2)	In addition to the two hyperparameters in the final loss function, the paper also involves parameters such as top-k and the Lagrange multiplier. How were these parameters selected? Which ones were fixed, and which were tuned during experiments?

(3)	While mutual information is inherently difficult to optimize, the paper offers limited detail on the implementation and underlying rationale, which may leave readers confused.

---

> ### Author Rebuttal · Authors · 2025-07-31
>
> I sincerely appreciate your professional review of my manuscript and your recognition of its solid theoretical foundation, innovative methodology design, and comprehensive experimental validation. I am also  grateful for your thoughtful identification of the weaknesses and existing issues in my work, which provide essential guidance for its further enhancement. I will undertake a thorough revision in light of your suggestions and endeavor to address each of your concerns with clarity and analytical depth.
>  ### **Weakness1: Difficult to understand the numerous symbols**
> The definitions of the fundamental notations are provided in Section 2.1, and additional symbols are explained as they appear in the paper. Given the extensive use of mathematical symbols, I will conduct a meticulous review to identify and correct any writing issues such as inconsistent notation, redundant symbol usage, or inaccuracies in variable dimensionality. These revisions will ensure that the paper meets a high standard of academic rigor.  Furthermore, a comprehensive notation table will be included in the revised version to facilitate reader comprehension and improve overall clarity.
>  ### **Weakness2: A detailed explanation of all parameter configurations**
> In addition to the balanced coefficients $\lambda_{1}$ and $\lambda_{2}$ in the total training loss, the following parameters are also involved, i.e., the temperature parameter $\tau$, the balanced factor $\alpha$, the number of nearest neighbors $k$ for each sample,  the Lagrange multiplier
> $\beta$, the sampling number $s$, and the learning rate. Among these parameters, only the values of $\lambda_{1}$ and $\lambda_{2}$ are tuned during the experiments, while the rest are fixed throughout. The detailed configurations are summarized in the table below:
> |Parameter|Annotation|Value|
> |-|-|-|
> | $\lambda_{1}$  |Balanced coefficient for training loss|$\\{0.01,0.1,1,10,100,1000\\}$|
> | $\lambda_{2}$|Balanced coefficient for training loss|$\\{0.01,0.1,1,10,100,1000\\}$|
> |$\tau$|Temperature parameter|0.2|
> |$\alpha$|The balanced factor|10|
> |$k$|Number of nearest neighbors|10|
> |$\beta$|Lagrange multiplier|1|
> |$s$|The sampling number |5|
> |Learning rate|Network optimization parameter|0.0001|
>  ### **Weakness3: Implementation of mutual information optimization**
> By performing variational derivations for each mutual information term in Eq. (4), we obtain their estimable bounds, which in turn lead to the formulation of Eq. (7) as the optimization objective. In practice, the lower bound for information shift $\mathbb{E}_{p\left({z}^{(v)}, {s}^{(v)}\right)}\left[\log q\left({s}^{(u)} | {z}^{(v)}\right)\right] $ in Eq. (7) is implemented as a reconstruction loss between the decoded representation and the original view. The detailed computation of the remaining three terms is provided in Appendix A.2. Specifically, representations are treated as probability vectors over $D$ classes via a Softmax activation function, and the joint distribution matrices (${P}^{(v,v^\*)}, {Q}^{(v,v^\*)}, {M}\^{(s\^{v},o\^{v})}$)  between different representations are obtained by cross-view instance-level contrastive learning. By aggregating along different directions of the joint distribution matrices, the corresponding marginal distributions of individual views can be derived. Subsequently,  by converting the integral to a summation form, the remaining three terms can be expressed as
>
> $\int \int p({s}\^{(v)}|{z}\^{(v)};{s}\^{(v^{\*})}|{z}\^{(v^{\*})})\log q({s}\^{(v)}|{s}\^{v^{\*}}) d{s}\^{(v^{\*})} d{s}\^{(v)}=\sum_{d=1}^D\sum_{d'=1}^D {P}_{d,d^{\prime}}^{(v,v^\*)} \log \frac{{P}\_{d,d\^{\prime}}\^{(v,v^\*)}}{{P}\_{d\^{\prime}}\^{(v^\*)}}$
>
> $D_{KL} (p({o}^{(v^{\*})}|{z}^{(v^{\*})};{o}^{(v)}|{z}^{(v)}) \| p({o}^{(v^{\*})}|{z}^{(v^{\*})})q({o}^{(v)}|{o}^{(v^{\*})}))=\sum_{d=1}^D\sum_{d'=1}^D {Q}_{d,d^{\prime}}^{(v,v^\*)} \log \frac{{Q}\_{d,d^{\prime}}^{(v,v^\*)}} {({Q}\_{d}^{(v)})^{\alpha} {Q}\_{d^{\prime}}^{(v^\*)}}$
>
> $D_{KL} (p({s}^{(v)}|{z}^{(v)};{o}^{(v)}|{z}^{(v)}) \| p({o}^{(v)}|{z}^{(v)})q({s}^{(v)}|{o}^{(v)}))=\sum_{d=1}^D\sum_{d'=1}^D  {M}_{d,d^{\prime}}^{(s^v,o^v)} \log\frac{{M}\_{d,d^{\prime}}^{(s^v,o^v)}}{({M}\_{d}^{(s^v)})^{\alpha} {M}\_{d^{\prime}}^{(o^v)}}$
>
> In this conversion, we let  $q({s}^{v}/{s}^{v^*})= (\psi({Q}^{(v)}))^{\alpha}$ and $q({z}^v/{s}^v)=(\psi({M}^{(s^v)}))^{\alpha}$, where $\psi$ is a fully connected layer, and  $\alpha$ is a balance factor to preserve crucial information and ensure  model stability.
>  ### **Question1,2: How this work addresses the feature extraction and label correlation modeling**
> ● Previous methods either focus solely on shared representations while overlooking the unique contributions of each individual view, or rely on linear geometric constraints when extracting view-specific features, which are insufficient to capture complex representation interactions. To address this, we adopt a mutual information constrained optimization objective to achieve information disentanglement between shared and specific representations. Given the absence of a unified framework or guiding principles for leveraging mutual information objectives in representation learning, we systematically formulate three information-theoretic principles for multi-view feature extraction: information shift, interaction, and orthogonality. The proposed framework is grounded in three core aspects, i.e., alignment with the original views, cross-view correlations, and intra-view representational interactions. In addition, we provide a theoretical analysis that validates the effectiveness of our feature extraction model and proves its potential to reduce generalization error.
>
> ● Label correlation modeling is jointly performed with prototype representation learning in our work. First, we quantitatively estimate label correlations based on the training data. Rather than utilizing co-occurrence frequency to evaluate correlation degree, we use the Jaccard distance calculated  over positive labels, as we are only concerned with the categories assigned to each instance.  By  computing  the intersection and union of  two classes regarding positive values, we can obtain $A_{i,j}=\frac{\langle {Y}\_{:,i} \cdot {Y}\_{:,j}\rangle}{\sum\_{k=1}^{N}({Y}\_{k,i}+{Y}\_{k,j})-\langle {Y}\_{:,i} \cdot {Y}\_{:,j}\rangle}$. Then  we sample the prototype representation for each class, and then update them through a graph network that incorporates $A$ as the adjacency matrix, which yields label prototype matrix $E$ that encodes inter-label dependencies. Besides, to enhance the alignment among semantically related prototype representations while encouraging separation between unrelated ones, we introduce the following objective that aligns representation similarity with label correlation:
>
> $\mathcal{L}\_{le}= -\frac{1}{C^{2}} \sum\_{i=1}^{C} \sum\_{j=1}^{C}\hat{{A}}\_{i j} \log \left(\cos \left({E}\_{i}, {E}\_{j}\right)\right)+(1-\hat{{A}}\_{i j})\log \left(1-\cos \left({E}\_{i}, {E}_{j}\right)\right)$
>
> The ablation experiments demonstrate that the modeling of both challenges offers essential gains.
> ### **Question3: Understanding of discriminability**
> In our work, discriminability refers to the model’s ability to accurately distinguish between different classes by aligning learned prototype representations with view-level features, while also capturing the distinctiveness of each target category across multiple views.  Specifically, discriminability measures whether a given view can provide stronger predictive support for a specific class compared to other classes, and whether it is more competent than other views in predicting that particular class. Therefore, discriminability should be assessed from two complementary perspectives:
>
> ● Intra-class discriminability. For a given view $v$, the prediction score for the target class $j$ should be higher than that for any non-target class $k$:
>
> $\mathbb{E} \left[ U^{(v)}\_{:,j} \right] > \mathbb{E} \left[ U^{(v)}\_{:,k} \right], \quad \forall k \neq j.$
>
> ● cross-view discriminability. For a given class $j$, the prediction score produced by view $v$ should exceed that of any other view $v^*$:
>
>
> $\mathbb{E} \left[ U^{(v)}\_{:,j} \right] > \mathbb{E} \left[ U^{(v^{\*})}\_{:,j} \right], \quad \forall v \neq v^{\*}.$
>
> ### **Question4: The issue of code release**
> We commit to releasing all codes and data on GitHub upon acceptance of the paper.

---

### Official Review · Reviewer_rTba · 2025-07-02

**Clarity:** 3
**Significance:** 3
**Originality:** 3
**Rating:** 4
**Confidence:** 4

**Summary:**

This work studies incomplete multi-view multi-label classification, simultaneously addressing missing views and missing labels. It first employs available view samples to recover missing views using an attention-based neighborhood imputation approach. Subsequently, it learns view-shared and view-specific representations via mutual information estimation, with learning objectives comprising three terms: information shift, information interaction, and information orthogonality. The authors then introduce label prototype learning. By leveraging a label correlation matrix and a GNN, the semantic representational capacity of these so-called label prototypes is optimized. These prototypes are applied to select label-relevant features from both view-shared and view-specific features. Final predictions are generated via a linear classifier and late fusion: predictions from individual views are first fused, followed by fusion of predictions derived from shared and specific features.

**Questions:**

What evaluation metric is used for the results in Appendix Figure 2? I cannot find this information in the text or figure.

**Ethical Concerns:**

["NO or VERY MINOR ethics concerns only"]

**Final Justification:**

I thank the authors for their response. The detailed rebuttal addresses my main concerns regarding Weaknesses 4 and 5, and I will raise my score accordingly. I suggest the authors incorporate the explanation for Weakness 4 from the rebuttal into Section 2.2 of the main text and carefully revise the paper based on my identified weaknesses and the corresponding rebuttal.

**Limitations:**

The paper appears to lack a comprehensive evaluation of the limitations of the proposed method.

**Paper Formatting Concerns:**

I do not find any paper formatting issues.

**Quality:**

2

**Strengths And Weaknesses:**

Strengths：

1.This paper covers three main research aspects: missing view imputation, extraction of shared and specific features, and the use of label prototypes and label correlations.

2.The writing is clear and easy to follow, and the paper is well organized. The introduction provides sufficient background and related work, and the authors include extensive theoretical analysis.

3.The authors conduct a large number of comparative experiments under different missing rates, both in the main text and the appendix. They compare TDLSR with many existing methods, providing a thorough evaluation of the proposed approach.

Weaknesses：

1. The variable dimensions are not clearly specified. For example, the label correlation matrix $A$ and the label prototype matrix $E$ are not explicitly defined, even though they can be inferred. Clear definitions are necessary.

2. Variable confusion: For example, $S^{(v)}$ refers to the view-specific graph in Section 2.2, but to view-shared features $\\{S^{(v)}\\}_{v=1}^V$ in Section 2.3. In Section 2.4, $E$ is a matrix, and while readers can infer that $E_i$ is the prototype of the $i$-th label, its form is not clearly defined.

3. Dimension error: In line 178, $H \in \mathbb{R}^{q \times d}$ is incorrect, since the adjacency matrix $A \in \mathbb{R}^{C \times C}$, which makes the operation $E = f[(1+\epsilon)H + AH]$ in line 180 dimensionally inconsistent. It should be $H \in \mathbb{R}^{C \times d}$.

4. In Section 2.2 of the main text, Eq. (2) uses $B_{i,j}^{(v)}$ in the numerator to compute $\hat x_i^{(v)}$. However, if $x_i^{(v)}$ is a missing view, it is unclear how $B_{i,j}^{(v)}$ can be computed. It seems that in the case where the $i$-th sample is missing, $B_{i,j}^{(v)}$ is not computable. I cannot understand how Eq. (2) works in the case of missing views.

5. In Section 2.3, regarding the first term (information shift), $s^{(u)}$ in Eq. (5) and Eq. (7) appears to be a typo. Also, line 148 claims that $I(s^{(v)}; z^{(v)})$ is lower-bounded by a reconstruction loss, but Eq. (7) optimizes $-\mathbb{E}_{p(z^{(v)},s^{(v)})}[\log q(s^{(v)}|z^{(v)})]$. Since $z^{(v)}$ is the imputed original feature and $s^{(v)}$ is the shared feature, decoding $s^{(v)}$ from $z^{(v)}$ seems conceptually incorrect. Moreover, line 149 refers to $q^v(z^{(v)}|s^{(v)})$ for decoding, which contradicts the optimization objective. This issue also appears in the appendix: in Eq. (3), the final term $\log p(z^{(v)}|s^{(v)})$ becomes $\log p(s^{(v)}|z^{(v)})$ in Eq. (5), and in Eq. (6), the second line $\log q(z^{(v)}|s^{(v)})$ is incorrectly changed to $\log q(s^{(v)}|z^{(v)})$, which seems to be a derivation error.

6. In Eq. (11), the right-hand side term $\overline B_{i,j}W_{j,v}$ should be $\overline B_{i,j}W_{i,v}$. Also, in Figure 1, the imputed feature on the left should be $Z^{(v)}$, not $X^{(v)}$.

---

> ### Author Rebuttal · Authors · 2025-07-30
>
> I sincerely appreciate your thorough reading of my manuscript and your recognition of its strengths, including clear writing, rich theoretical content, and extensive experiments. I am also deeply grateful for your careful identification of the writing flaws, as your feedback is immensely helpful in guiding the refinement and enhancement of the paper.  Please accept my sincere apologies for any inconvenience my oversights may have caused during your review. I will pay closer attention to every detail in subsequent revisions to ensure that all writing errors are addressed and the work attains the highest level of academic rigor.
>
> ### **Weaknesses：** ###
>  **Clarity for variable dimension.**  $A \in \mathbb{R}^{C \times C}$ is the label correlation matrix, where each element $A_{i,j}$ denotes the degree of correlation between the $i$-th and $j$-th labels, as computed by Eq. (8). $H \in \mathbb{R}^{C \times d}$ denotes the aggregation matrix of all label prototypes, constructed by row-wise concatenation, where each row encapsulates the prototype representation of an individual label.
>
>  **Clarity for variable confusion.**  Following the revision of the symbol $S$, in Section 2.2, $\hat{S}^{(v)}$ denotes the proximity-aware graph (sample topology graph) for each view, where $\hat{S}^{(v)}\_\{i,j\}$ indicates that  samples $i$ and $j$ are neighbors and co-exist in the $v$-th view. In Section 2.3, $S^{(v)}$ refers to the view-specific representation matrix extracted from the raw view.  $E \in \mathbb{R}^{C \times d}$ denotes the updated label prototype matrix derived from $H$ via the GIN layer, with each row representing the refined prototype of a label. In Eq. (11), $Q_{i,v}$ represents the maximum confidence with which sample $i$ can be reconstructed using other samples in the $v$-th view.  The term $W_{j,v}$
>   is used here to ensure that only the samples present in the
> $v$-th view are considered for reconstructing sample
> $i$; those that are missing are assigned zero confidence via the missing data mask. In the "Proximity-aware Graph Attention Recovery" module of Figure 1, the reconstructed view on the right should be $\\{{Z}^{(v)}\\}\_{v=1}^{V}$ rather than $\\{{X}^{(v)}\\}_{v=1}^{V}$.
>
>  **Clarity for the information shift term.** The information shift term $I({s}^{(v)};{z}^{(v)})$ is used to guarantee
> that the representations stay consistent with the core semantics of raw views. Its lower bound $I({s}^{(v)};{z}^{(v)})) \geq \mathbb{E}\_{p\left({z}^{(v)}, {s}^{(v)}\right)} \left\[\log q\left({z}^{(v)} | {s}^{(v)}\right)\right\]$, derived via variational inference, was mistakenly written as $\mathbb{E}_{p\left({s}^{(u)}, {z}^{(v)}\right)}\left[\log q\left({s}^{(u)} | {z}^{(v)}\right)\right] $ in Eq. (5). The term $q^{v}({z}^{(v)} | {s}^{(v)})$ used for decoding in line 149 is the same as $q({z}^{(v)} | {s}^{(v)})$ in the optimization objective. Similarly, in Eq. (4) of the Appendix,  $q({z}^{(v)} | {s}^{(v)})$ is incorrectly written as $q({s}^{(v)} | {z}^{(v)})$, which leads to the subsequent notation errors in Eqs. (5) and (6). However, the overall derivation remains valid, as the mistake lies solely in the reversed notation, not in the underlying logic.
>
> **Explain how Eq. (2) is used for view reconstruction.**  Our reconstruction strategy is designed to exploit the sample correlation structure across the available views to recover the missing ones, without introducing additional network parameters. ${K}^{(v)}$ is a transferred graph that determines the available samples in the $v$-th view to be used for reconstructing the missing ones. In a specific view, the contribution of available samples to the reconstruction of missing samples is determined by the maximum computable sample correlation derived from other views. Specifically,  $\overline{{B}}\_{i, j}=\max \left({B}\_{i, j}^{(1)}\hat{S}\_{i, j}^{(1)}, {B}\_{i, j}^{(2)}\hat{S}\_{i, j}^{(2)}, \ldots, {B}\_{i, j}^{(V)}\hat{S}\_{i, j}^{(V)}\right)$, where $\overline{{B}}\_{i, j}$ denotes the contribution of the
>  sample $j$ to the reconstruction of the sample $i$.  The term $\overline{{B}}\_{i, j}$, which is formally defined in Eq. (11) of Section 2.4, is not explicitly introduced in Section 2.2, and the symbol $\hat{S}$ is incorrectly denoted as ${K}$ in Section 2.4. We acknowledge this gap and will incorporate the necessary clarifications and revisions to prevent any potential misinterpretation regarding the process of reconstruction.  Consequently, the reconstruction of missing samples is ultimately achieved through a weighted aggregation, i.e., $\hat{x}\_{i}^{(v)}=\frac{\sum_{{K}\_{i, j}^{(v)} \geq 1} {K}\_{i, j}^{(v)} \overline{{B}}\_{i, j}  {x}\_{j}^{(v)}}{\sum_{{K}\_{i, j}^{(v)} \geq 1} \overline{{B}}\_{i, j}}$.  As $\hat{{X}}$ serves as an approximate substitute for the missing instances,  it is integrated with the original available views to construct the final recovered data ${\hat{{Z}}}\_{i,:}^{(v)}=\hat{{X}}\_{i,:}^{(v)}\left(1-{W}_{i, v}\right)+{X}\_{i,:}^{(v)} {W}\_{i, v}$. It is also worth noting that although ${B}\_{i, j}^{(v)}=e^{h(x\_{i}^{(v)}) h(x\_{j}^{(v) T}) / \tau}$ is computed over all sample pairs,  the value of $\overline{{B}}\_{i, j}$ is only retained when both sample $i$ and sample $j$ are available,  as indicated by the sample topology graph   $\hat{S}^{(v)}$, where $\hat{S}^{(v)}$ equals 1 only if both corresponding samples exist. Therefore, all valid entries in $\overline{{B}}$ are computable, and the entire process ensures that no feature information from missing samples is leaked. Moreover, owing to the varying levels of reconstruction fidelity,  the reconstruction quality is closely tied to the weights assigned during the late-stage fusion in our model. The prediction from a reconstructed sample is weighted according to the maximum confidence with which the sample can be reconstructed from other samples, whereas the prediction from an observed sample is assigned a weight of 1. Next, we present a comparative analysis between our proposed reconstruction method and a baseline strategy that utilizes the mean of available views:
> || Pascal07 | OBJECT | Mirflickr |Corel 5k | ESPGame|IAPRTC12|
> |-|-|-|-|-|-|-|
> | average  | 0.852    | 0.903  | 0.859 | 0.898 | 0.850 |0.871 |
> |ours| 0.882    | 0.924  | 0.875| 0.919 | 0.863 | 0.899 |
>
> It is evident from the above results that the proposed view reconstruction method is instrumental in achieving superior classification performance. In light of the large number of symbols used in this paper, as well as the presence of certain writing inconsistencies, we provide a summary below listing the key notations along with their corresponding dimensions：
>
> | **Notation** | **Annotation** |
> | - | -|
> | $N$| Number of samples|
> | $V$ | Number of views|
> | $C$ | Number of labels/classes|
> | $d_v$| Feature dimension of view $v$|
> | ${X}^{(v)} \in \mathbb{R}^{N \times d_v}$| Feature matrix of $v$-th view|
> | ${Y}\in \\{0,1\\}^{N\times C}$| Binary multi-label matrix |
> | ${W}\in\\{0,1\\}^{N\times V}$| Missing view indicator matrix|
> | ${G}\in\\{0,1\\}^{N\times C}$| Missing label indicator matrix|
> | ${Z}^{(v)} \in \mathbb{R}^{N \times d_v}$| Reconstructed features for view $v$|
> | ${B}^{(v)} \in \mathbb{R}^{N \times N}$| Attention scores in view $v$|
> | $\hat{S}^{(v)}\in\mathbb{R}^{N\times N}$| Sample topology graph for view $v$|
> | ${K}^{(v)}\in\mathbb{R}^{N\times N}$| Propagation graph for imputation|
> | $\hat{{X}}^{(v)}\in\mathbb{R}^{N\times d_v}$| Imputed features for view $v$|
> | $M^S_{v},M^O_{v}$| MLP for shared/private representation|
> | ${S}^{(v)},{O}^{(v)}$| Shared / private representation for view $v$|
> | $I(\cdot ; \cdot)$| Mutual information|
> | ${b}_i \in \mathbb{R}^C$| One-hot vector of $i$-th category|
> | ${h}_i$| Prototype of label $i$ |
> | ${A}\in\mathbb{R}^{C\times C}$| Label correlation matrix (Jaccard)|
> | ${E} \in \mathbb{R}^{C \times d_v}$| Label embedding matrix|
> | $\hat{{U}}^{(v)},\hat{{V}}^{(v)} \in \mathbb{R}^{N\times C\times d}$ | Label-specific shared/private feature|
> | ${U}^{(v)},{V}^{(v)} \in \mathbb{R}^{N\times C}$| Class predictions from shared/private|
> | ${Q}\in\mathbb{R}^{N\times V}$| Confidence matrix |
> | $\overline{{P}}\in\mathbb{R}^{N\times C}$| Final fused prediction matrix|
> | $R(f),\widehat{R}_D(f)$| Expected/empirical risk|
> ### **Question 1: Figure 2 in the Appendix lacks evaluation metrics** ###
> Figure 2 in the Appendix illustrates how the balanced parameters $\lambda_{1}$ and $\lambda_{2}$ in the training loss jointly affect classification performance.  The values shown in the heatmap represent the AP metric, which evaluates the prediction quality for each class by computing the area under the precision-recall curve and reflects the model’s ranking performance across different labels.
> ### **Discussion for the limitations:** ###
> This study faces two principal limitations. The first concerns the adaptability of the approach to diverse scenarios. Our work primarily addresses situations in multi-view multi-label learning where both views and labels are missing. As the sample size increases, however, additional challenges such as view misalignment and label noise are likely to emerge. These issues warrant further investigation in future research within the multi-view multi-label domain. The second limitation lies in the method design.  We estimate the mutual information between views through a data-driven procedure that approximates their joint distribution as well as the marginal distribution of each view. Although this approach is effective, estimation errors are unavoidable, and future work should aim to develop more precise techniques. Moreover, our method assumes that label prototypes are generated directly from Gaussian distributions. This assumption may be restrictive and inconsistent with the complexity of real-world applications. Subsequent research should pursue more flexible strategies for constructing label prototypes free from conditional constraints.

---

> ### Comment · Reviewer_rTba · 2025-08-06
>
> I thank the authors for their response. The detailed rebuttal addresses my main concerns regarding Weaknesses 4 and 5, and I will raise my score accordingly. I suggest the authors incorporate the explanation for Weakness 4 from the rebuttal into Section 2.2 of the main text and carefully revise the paper based on my identified weaknesses and the corresponding rebuttal.

---

> ### Author Response · Authors · 2025-08-07
> **Thank you!**
>
> Thank you sincerely for your thoughtful review. We greatly appreciate your careful consideration of our rebuttal, especially your acknowledgement that our responses addressed all your concerns. We will incorporate the clarification for Weakness 4 into Section 2.2 of the main text as suggested, and carefully revise the paper based on your valuable feedback. Thank you again for your support and raising your score.

---

### Note · Authors · 2025-08-16

Dear AC and Reviewers,

We sincerely thank you for your professional and constructive feedback. We are encouraged that you found our innovative design, solid theoretical support and comprehensive experimental validation. Meanwhile, we have provided detailed replies to all the comments in the rebuttal. Below, we summarize the key concerns raised and our corresponding responses:

- **Writing errors** (Reviewer `rTba` and  `FGNW`) **and detailed captions** (Reviewer `P58t`)
We clarified all errors related to the formula symbols, provided a table listing all symbols for reference and offered detailed explanations of the content and purpose of Figures 1 and 2.

- **Understanding of the recovery mechanism** (Reviewer `rTba` and `P58t` )
We provided explanations for each formula required in the recovery process, clarified the underlying principles and benefits and conducted ablation experiments.

- **Implementation details** (Reviewer `FGNW` and `P58t`) and **running time** (Reviewer `NS9F`)
We provided detailed implementation process of the algorithm and the selection methods for all parameters and conducted comparative analysis of the time complexity.

- **The principle of feature extraction and the optimization problem** (Review `FGNW`, `NS9F` and `P58t`)
We clarified the optimization process in the appendix and explained our innovations in mutual information modeling.

- **Label correlation modeling** (Review `FGNW` and `P58t`)
We provided a detailed explanation of the design concepts and roles of label semantic learning and label-specific representation learning.

- **The rationale behind the theoretical proof and the required assumptions** (Review `FGNW`, `NS9F` and `P58t`)
We examined the justification for the assumptions required by Theorem 2 from various perspectives and provided a concise explanation of the algorithm's discriminability and the proof of its generalization error.

We will incorporate the revisions and additional experimental results into the updated version of the paper. We truly appreciate your valuable insights, which have significantly improved our work.  We have made every effort to thoroughly address all the issues raised. Thank you for the time you have dedicated. We look forward to your consideration.

Sincerely,

The Authors

---

### Decision · Program_Chairs · 2025-09-17

**Decision:**

Accept (spotlight)

**Comment:**

This paper proposes TDLSR, a theory-driven framework for incomplete multi-view multi-label classification. It addresses missing views via an attention-based imputation method and disentangles shared/specific representations using mutual information constraints. Theoretically, it provides discriminability analysis and generalization bounds. Experiments on six datasets and a real-world NBA application demonstrate superior performance over SOTA methods. After the rebuttal, all reviewers recommended acceptance. Due to its strong theoretical contributions, innovative methodology, and comprehensive empirical validation, the paper is recommended for acceptance.